# Uptake of small extracellular vesicles by recipient cells is facilitated by paracrine adhesion signaling

Koichiro M. Hirosawa [1], Yusuke Sato[2], Rinshi S. Kasai [3], Eriko Yamaguchi[1], Naoko Komura[1], Hiromune Ando [1,4,5], Ayuko Hoshino[6,7], Yasunari Yokota[8] & Kenichi G. N. Suzuki [1,3,4,5]

Small extracellular vesicles (sEVs) play crucial roles in intercellular communication. However, the internalization of individual sEVs by recipient cells has not been directly observed. Here, we examined these mechanisms using state-of-the-art imaging techniques. Single-molecule imaging shows that tumor-derived sEVs can be classified into several subtypes. Simultaneous single-sEV particle tracking and observation of super-resolution movies of membrane invaginations in living cells reveal that all sEV subtypes are internalized via clathrin-independent endocytosis mediated by galectin-3 and lysosome-associated membrane protein-2C, while some subtypes that recruited raft markers are internalized through caveolae. Integrin β1 and talin-1 accumulate in recipient cell plasma membranes beneath all sEV subtypes. Paracrine, but not autocrine, sEV binding triggers $Ca^{2+}$ mobilization induced by the activation of Src family kinases and phospholipase Cγ. Subsequent $Ca^{2+}$-induced activation of calcineurin–dynamin promotes sEV internalization, leading to the recycling pathway. Thus, we clarified the detailed mechanisms of sEV internalization driven by paracrine adhesion signaling.

Most cells secrete extracellular vesicles (EVs). For more than a decade, small EVs (sEVs) have attracted great attention as mediators of intercellular communication, because sEVs transfer bioactive molecules, such as proteins, lipids, and nucleic acids, to recipient cells, altering their physiology and function[1]. Despite their importance, how sEVs are internalized by recipient cells has not been fully understood. Several studies have suggested that sEV uptake is complex and heterogeneous and can occur through various mechanisms, such as direct fusion between sEVs and recipient cell plasma membranes (PMs), endocytosis, and phagocytosis[2]. However, the pathways and factors controlling sEV uptake have not been directly observed. Because not all EVs

are created equal, and EV subtypes have distinct functions and compositions[3–7], the behavior of individual sEV subtypes should be investigated. The uptake mechanism may vary depending on the sEV subtype or recipient cell type; however, this possibility has not been investigated[8]. Clarifying the mechanisms of sEV uptake is critical for developing effective strategies for sEV-based therapies and diagnostics[9,10]. Moreover, elucidating factors that regulate sEV uptake can provide insights into the biological functions of sEVs and their roles in various physiological and pathological processes[5].

Here, we show that sEVs derived from two distinct tumor cell lines are categorized into discrete subtypes through single-particle imaging

[1]Institute for Glyco-core Research (iGCORE), Gifu University, Gifu 501-1193, Japan. [2]Department of Chemistry, Graduate School of Science, Tohoku University, Sendai 980-8578, Japan. [3]Division of Advanced Bioimaging, National Cancer Center Research Institute (NCCRI), Tokyo 104-0045, Japan. [4]Institute for Integrated Cell-Material Sciences, Kyoto University, Kyoto 606-8501, Japan. [5]Innovation Research Center for Quantum Medicine. Graduate School of Medicine, Gifu University, Gifu 501-1193, Japan. [6]Research Center for Advanced Science and Technology, University of Tokyo, Tokyo 153-8904, Japan. [7]Inamori Research Institute for Science, Inamori Foundation, Kyoto 600-8411, Japan. [8]Department of Electrical, Electronics and Computer Engineering, Faculty of Engineering, Gifu University, Gifu 501-1193, Japan. ✉e-mail: suzuki.kenichi.b7@f.gifu-u.ac.jp

with single-molecule detection sensitivity. Our simultaneous high-speed super-resolution imaging using photoactivated localization microscopy (PALM) and single-particle tracking of sEVs in living cells elucidates their internalization pathways in recipient cells. Furthermore, our findings reveal the detailed mechanisms of paracrine sEV adhesion signaling in recipient cells, which facilitates the sEV uptake.

## Results

### Validation of the presence of sEV subtypes

Several studies have suggested that sEVs, even when released from a single cell type, exhibit a biased molecular composition, suggesting the presence of various sEV subtypes[3,11,12]. Therefore, as a starting point for analyzing the interaction of sEVs with their recipient cells, we validated the presence of sEV subtypes originating from two distinct tumor cell lines. When two fluorescently labeled antibodies are used simultaneously to label two sEV marker proteins, each antibody may prevent the other antibody from binding to the marker proteins[4]. In addition, because the number of dye molecules per antibody molecule follows a Poisson distribution and the dye-to-antibody ratio is not always 1, we cannot determine the presence of marker proteins in an sEV particle. To overcome this challenge, we used sEVs containing tetraspanin marker proteins, such as CD63, CD9, and CD81 fused with monomeric GFP (mGFP) or Halo7-tag labeled with TMR at very high efficiency (Supplementary Fig. 1a). Under this condition, the marker proteins are labeled with fluorophores at a dye to protein ratio of 1. Then, individual sEV particles were visualized by total internal reflection fluorescence microscopy (TIRFM), with single-molecule detection sensitivity (Fig. 1a and Supplementary Fig. 1b, c).

Therefore, we expressed both mGFP-tagged and Halo7-tagged marker proteins in donor cells and collected sEVs from the cell culture medium. To ascertain the presence of marker proteins in sEVs, it was necessary to isolate stable cell lines that express both mGFP and Halo7-tagged marker proteins, but do not overexpress them to avoid detrimental effects. First, stable cell lines were isolated by a cell sorter (Fig. 1b), and western blot analysis was performed with the obtained sEVs (Fig. 1c). The results indicated that the copy numbers of exogenous tetraspanin marker proteins, CD9 and CD81, tagged with mGFP or Halo7, were approximately 0.2-fold lower than those of the endogenous proteins (Supplementary Fig. 2), and the copy number of the tagged CD63 was 0.5-fold lower than that of the endogenous protein (Fig. 1c). By fitting the fluorescence intensity histograms of tetraspanin-Halo7-TMR with log-normal function curves[13] (Supplementary Fig. 1d, e), the average number of tetraspanin-Halo7 per sEV was estimated to be approximately 4.2–4.8 (Fig. 1d). Assuming that the numbers in each sEV particle follow a Poisson distribution, 98.5–99.2% of sEVs should contain at least one labeled tetraspanin molecule, indicating that nearly all sEVs containing the specific tetraspanins were detected. Furthermore, because the amounts of tetraspanin-mGFP were similar to those of the tetraspanin-Halo7 (Fig. 1b, c), almost all sEVs containing the specific tetraspanin-mGFP should be detected. According to simultaneous two-color observations of sEVs, a large fraction of sEVs contained both CD9-mGFP and CD9-Halo7-TMR, which were used as positive controls. Only a small fraction of the sEVs contained both CD9-mGFP and CD63-Halo7-TMR (Fig. 1e). A similar analysis of all combinations of tetraspanin marker proteins is shown in Fig. 1f. As illustrated in the Venn diagram, two sEV subtypes are derived from PC-3 cells—one containing only CD63 and the other containing CD9, CD63, and CD81. This distribution significantly deviated from the random variation predicted by a Poisson distribution. As previously noted, each sEV contains an average of 4.2–4.8 fluorescent marker molecules. If the marker molecules followed a single distribution, it should conform to a Poisson distribution, in which case 96.5–98.4% of the fluorescent marker molecules would be expected to reside within the same sEV. However, the distribution diverged from this expectation, indicating a pronounced bias in the distribution pattern of marker

molecules among sEVs. Two sEV subtypes are derived from 4175-LuT cells, one containing CD9 and CD63 and the other containing CD9, CD63, and CD81.

### sEV subtypes exhibit similar diameters but different membrane packing characteristics

The physical properties of each sEV subtype were characterized. Transmission electron microscopy (TEM) images obtained by negative staining (Fig. 2a) revealed that sEVs isolated by ultracentrifugation without subtype fractionation were 79 ± 18 nm (mean ± SD) in diameter, which is consistent with the value (71 ± 7 nm) measured by the electrical resistance nanopulse method (qNano) (Fig. 2b). Subsequently, we estimated the diameters of the sEV subtypes using super-resolution microscopy (dSTORM). SaraFluor™ 650B-GM3 (SF650B-GM3)-containing liposomes (Supplementary Fig. 3) with 111 ± 21-nm diameter, measured using the nanopulse method, were used as reference. Segmentation of the super-resolution images from the localization coordinates of SF650B was executed employing a Voronoi diagram using SR-Tesseler[14,15], and the diameters of the sEV subtypes were estimated as described in the Methods section (Fig. 2b, c). Analysis of liposomes with >100 localizations of SF650B revealed that liposome diameter was 110 ± 25 nm (Fig. 2b and Supplementary Fig 4). This value was highly consistent with that measured by the nanopulse method (Fig. 2b). The diameters of the sEVs, which included CD9-, CD63-, and CD81-Halo7 labeled with the SF650B-halo ligand, were 81 ± 24, 71 ± 22, and 71 ± 27 nm, respectively (Fig. 2b), indicating that diameters of different sEV subtypes are comparable.

To further characterize sEV subtypes derived from PC-3 cells, the membrane packing of each sEV subtype was quantified using a probe of the C-terminal region of apolipoprotein A-I labeled with 5-carboxytetramethylrhodamine (ApoC-TAMRA) that recognizes membrane defects (Supplementary Fig. 5)[16]. The amount of probe bound remains constant, regardless of the alterations in membrane charge, including variations in phosphatidylserine (PS) levels[16]. Average fluorescence intensity was significantly lower for ApoC-TAMRA bound to sEVs containing CD63-Halo7-SaraFluor™650T (SF650T) than those bound to sEVs containing CD81-Halo7-SF650T or CD9-Halo7-SF650T, indicating that membrane packing was greater in sEVs containing CD63-Halo7 (Fig. 2d) isolated by ultracentrifugation. Subsequently, the same analysis was performed using sEVs isolated by the PS binding protein, Tim-4 beads affinity method, a purification approach that does not involve size fractionation[17]. CD63-Halo7-containing sEVs isolated using Tim-4 affinity beads had a diameter similar to that of CD81-Halo7- or CD9-Halo7-containing sEVs (Supplementary Fig. 6a), but with greater membrane packing, as revealed by the fluorescence intensities of ApoC-TAMRA (Supplementary Fig. 6b) and another membrane defect-sensing probe—TAMRA-conjugated amphipathic helix (peptide 2-23) of α-synuclein bound to sEVs[18] (Supplementary Fig. 6c). Because membrane packing is closely correlated with membrane fluidity[19,20], the membrane fluidity of CD63-containing sEVs derived from PC-3 cells may be lower than that of CD9- and CD81-containing sEVs. These results demonstrate that the behaviors of sEVs must be investigated by subtype.

### Endocytosis of sEVs is controlled by the dynamin-related uptake machinery

Using advanced microscopy techniques, we examined the mechanisms of sEV endocytosis by recipient cells (Fig. 3a). First, sEVs containing CD63-Halo7-SaraFluor650T (SF650T) ($5 \times 10^{10}$ particles/mL at final concentration) were added to recipient PZ-HPV-7 cells ($1.0 \times 10^{6}$ cells), and the dynamic behavior of sEVs on the PM was observed using a 3D single fluorescent-molecule observation system with a cylindrical lens (Fig. 3a-c, Supplementary Fig. 7 and Supplementary Movie 1). The sEV particles moved linearly in the plane of the PM (Fig. 3b), while they were internalized by changing the z-coordinate

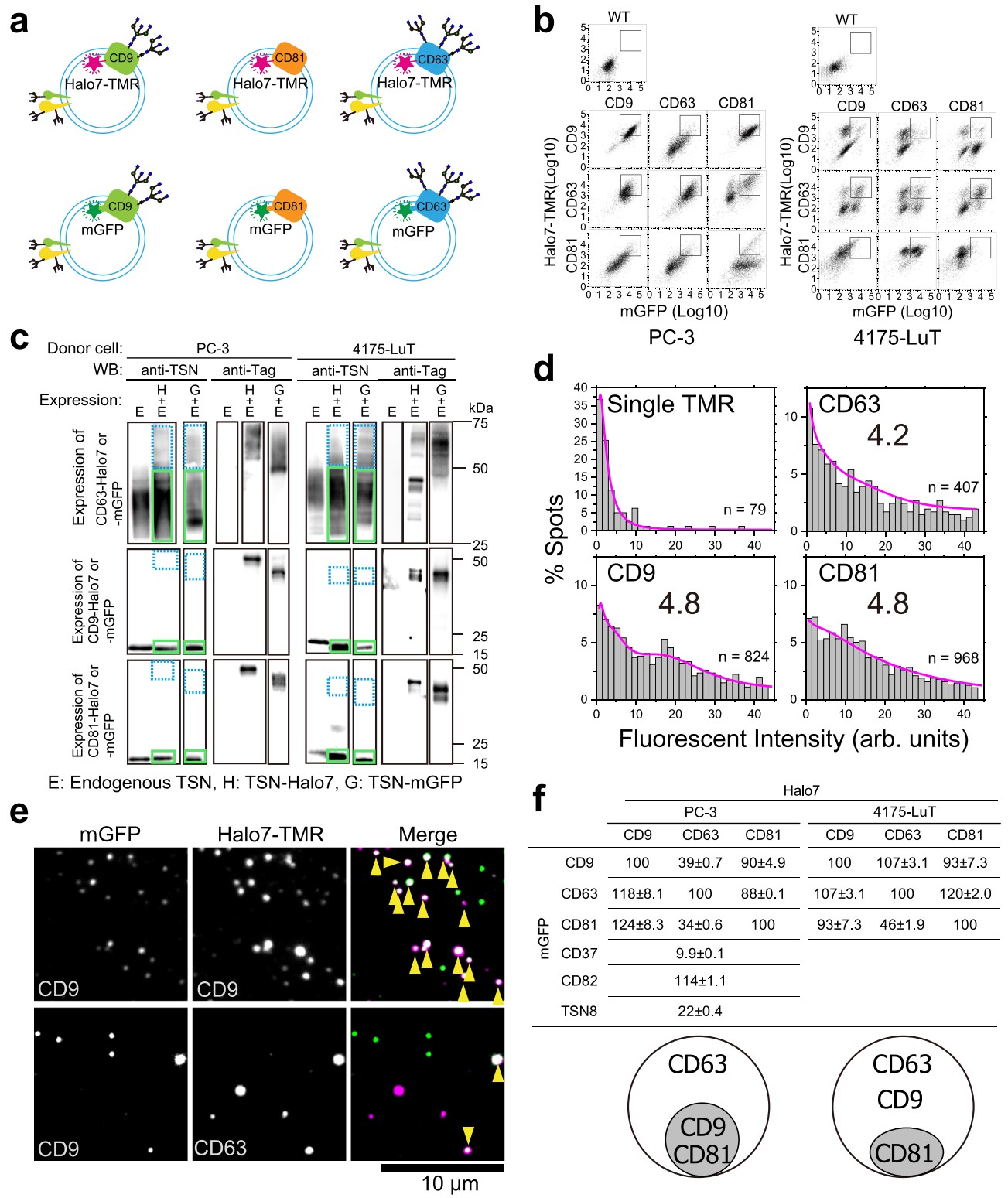

E: Endogenous TSN, H: TSN-Halo7, G: TSN-mGFP

| | | Halo7 | | | | | |
|---|---|---|---|---|---|---|---|
| | | PC-3 | | | 4175-LuT | | |
| | | CD9 | CD63 | CD81 | CD9 | CD63 | CD81 |
| mGFP | CD9 | 100 | 39±0.7 | 90±4.9 | 100 | 107±3.1 | 93±7.3 |
| | CD63 | 118±8.1 | 100 | 88±0.1 | 107±3.1 | 100 | 120±2.0 |
| | CD81 | 124±8.3 | 34±0.6 | 100 | 93±7.3 | 46±1.9 | 100 |
| | CD37 | | 9.9±0.1 | | | | |
| | CD82 | | 114±1.1 | | | | |
| | TSN8 | | 22±0.4 | | | | |

at a rate of 20 nm/s with repeated fluctuations (Fig. 3c). Second, the uptake rate of each sEV subtype was assessed by confocal microscopy, and no significant difference among the subtypes was observed for ≥20 min after sEV addition (Fig. 3d). Simultaneous analysis of sEVs and fluorescent dextran revealed that they strongly colocalize within 48 min of dextran addition (Fig. 3e). These results suggest that most sEVs are internalized by cells through endocytosis rather than through fusion with the PM as reported previously[2]. No occurrence of fusion was further corroborated by our single-molecule observations.

molecule observations. Namely, we observed that no mGFP-GPI molecules were transferred to the recipient cell membrane or underwent lateral diffusion in the recipient cell PM, even 120 min after sEVs containing mGFP-GPI were attached to the PM (Fig. 3f). Importantly, sEV uptake was greatly decreased in the recipient cell overexpressing the dominant negative mutant of dynamin (Fig. 3g, h, see also data with a dynamin inhibitor in Fig. 8a–c), indicating that dynamin-mediated endocytosis[21,22], plays a critical role in the uptake of PC-3 cell-derived sEVs by the recipient PZ-HPV-7 cell.

**Fig. 1 | Heterogenicity of small extracellular vesicles (sEVs). a** sEVs containing tetraspanin marker proteins. Three types of tetraspanin (TSN) (CD9, CD63, and CD81) were fused with mGFP or Halo7-tag. **b** Isolation of PC-3 or 4175-LuT cells expressing mGFP- and Halo7-labeled marker proteins using a cell sorter. Cells within a rectangle on the plot were collected. **c** Western blot analysis of sEVs derived from PC-3 or 4175-LuT cells that express endogenous tetraspanin (TSN) and TSN-Halo7 or TSN-mGFP at the same density levels as those in the rectangle in (**b**) (vertical axis for Halo7, the horizontal axis for mGFP). The cells were collected using a cell sorter. Western blotting was performed using antibodies against TSN, Halo7, or GFP. The ratio of the band intensities of tagged marker molecules in the dotted light blue box to those of endogenous marker molecules in the green box was calculated. Representative images from three individual blots are shown for each sample. **d** Distribution of fluorescence intensities of TSN-Halo7 labeled with tetramethylrhodamine (TMR) in individual sEV particles. Numbers in the graph indicate the average number of TMR molecules per sEV particle, which was estimated by curve fitting with the sum of the log-normal function. "*n*" indicates the number of examined sEV particles. **e** Typical dual-color fluorescence images of single sEVs on glass (green: mGFP, magenta: TMR) acquired by total internal reflection fluorescence (TIRF) microscopy. Double-stained sEV particles are indicated by yellow arrowheads. **f** (top) Table summarizing the results obtained from colocalization analysis of TSNs in sEVs. Numbers indicate the percentages of sEV particles with TSN-Halo7-TMR (top row) containing TSN-mGFP (left column). The percentages of tetraspanin-Halo7 in sEVs that colocalized with the same tetraspanin-mGFP were normalized to 100%. Occasionally, the percentages exceed 100%, likely attributable to minor variations in tetraspanin-mGFP content in sEVs. (bottom) Venn diagram of the inclusion relationship of TSNs in sEVs.

## Routes of sEV endocytosis in recipient cells are dependent on sEV subtypes

We aimed to directly determine how tumor PC-3 and 4175-LuT cell-derived sEVs are transported into normal recipient PZ-HPV-7 and WI-38 cells, respectively, through membrane invagination. Because PC-3 and PZ-HPV-7 cells are human epithelial prostate cancer and normal cells, respectively, we considered the combination as a model of sEV transfer between neighboring cells in the same tissue. sEVs derived from 4175-LuT human breast cancer cells preferentially bind to WI-38 normal human lung fibroblast cells[23]. This combination is used as a model of sEV-mediated organotropic metastasis. Therefore, we simultaneously observed single sEV particles and membrane invaginations (Fig. 4a, b). For PALM observation, mEos4b-fused AP2α and caveolin-1 (CAV1), which are marker proteins for clathrin-coated pits[24] and caveolae[25], respectively, were expressed in PZ-HPV-7 or WI-38 cells. Furthermore, we investigated the role of clathrin-independent endocytosis. A previous study demonstrated that the β-galactoside-binding lectin, galectin-3, is internalized via a clathrin-independent endocytic pathway[26]. Galectin-3 has also been abundantly found in sEVs[27]. Based on these findings, we investigated the presence of galectin-3 on the sEV membrane surface using immunostaining. Our TIRFM observations revealed that galectin-3 colocalized with nearly all sEVs bound to the recipient cell PM (Fig. 4c). Moreover, quantitative analyses showed that treatment with a membrane-impermeable galectin-3 inhibitor almost completely eliminated the colocalization of galectin-3 with sEVs bound to the PM (Fig. 4c). These results demonstrate that galectin-3 is localized on the surface of sEVs bound to recipient cells. Lysosome-associated membrane protein-2 (LAMP-2), a known galectin-3 binding partner on the cell surface[26,28,29], has been identified as a marker for clathrin-independent endocytosis[26,28–30]. Notably, previous studies demonstrated that LAMP-2C, but not LAMP-2A and -2B, localizes to the cell surface[31], is internalized in a dynamin-dependent manner[32], and does not interact with the clathrin machinery[31]. Therefore, we attempted to use LAMP-2C, which is expressed almost ubiquitously[33], as a marker for clathrin-independent endocytosis. However, LAMP2 is also predominantly expressed in lysosomes[34]. Therefore, we first investigated whether LAMP-2C, which is localized at the basal PM, could be visualized by TIRFM without detecting intracellular components. SiR-lysosome, a lysosome marker, was not observed even at the single-molecule level by TIRFM, however, it was visible using oblique-angle illumination (Supplementary Fig. 8). Similarly, marker molecules for the endoplasmic reticulum (STING) and the Golgi apparatus (Rab6a) were undetectable by TIRFM, however, both were observed under oblique-angle illumination (Supplementary Fig. 8). In contrast, numerous fluorescent spots of LAMP-2C-mEos4b molecules were detected by both TIRFM and oblique-angle illumination. Furthermore, most LAMP-2C-mEos4b molecules exhibited rapid lateral diffusion across wide areas, similar to other transmembrane proteins in the PM. These findings demonstrate that LAMP-2C-mEos4b molecules visualized by TIRFM are located on and near the cell PM, rather than in

lysosomes, and thus serve as a marker for clathrin-independent endocytosis.

We simultaneously performed PALM data acquisition by single-molecule observation of the mEos4b spots at a high speed (5 ms/frame) and single-particle observation of sEVs labeled with SF650T. To obtain one PALM image, data acquisition was performed for 1002 frames, which was then repeated by shifting the initial frames backward by six frames (30 ms). A total of 669 PALM images were obtained. By connecting the PALM image sequences, we generated a pseudo-real-time PALM movie (Fig. 4b). A parameter-free image resolution estimation method based on correlation analysis[35] revealed that spatial resolution in each frame of the PALM movie was 21 ± 0.4 nm (Supplementary Fig. 9a, b). Therefore, in the PMs of living cells, caveolae can be distinguished as two distinct structures, which cannot be resolved by conventional methods (Supplementary Fig. 9a). PALM image sequences acquired every 5 s showed that caveolae formed and disappeared on this timescale (Supplementary Fig. 9a), indicating that PALM movies should be generated to follow changes in membrane invagination. Images of single sEV particles were simultaneously recorded at 5 ms/frame, and the average images for six frames were connected to create a movie. Pseudo-real-time movies of membrane invaginations and single sEV particles were superimposed.

We found that PC-3-derived sEV particles containing CD63-Halo7-SF650T frequently colocalized with caveolae and clathrin-independent endocytic regions in the PM of PZ-HPV-7 cells (Fig. 4d–f and Supplementary Movies 2 and 3). In addition, we conducted super-resolution microscopy at a significantly lower temporal resolution (1 frame/s) over 5 min. The image sequences reveal that the fluorescence intensity of CD63 in an sEV particle, along with caveolin-1 or LAMP-2C molecules, synchronously diminished, with both disappearing within several minutes (Fig. 4g–j). Before this process, LAMP-2C molecules rapidly accumulated beneath the sEV particle within 1 min (Fig. 4h, j). These findings explicitly indicate that sEVs are internalized into these domains within 1–2 min (Fig. 4g–j).

It has been previously established that the degree of colocalization between membrane molecules and the endocytic machinery strongly correlated with subsequent internalization[36–39]. Therefore, we quantitatively analyzed the colocalization of these proteins and sEVs in Fig. 4d (Fig. 5a). The contours of membrane invaginations in PALM images were defined by binarization using kernel density estimation (KDE), which is the most useful method for analyzing protein clusters[40]. Subsequently, the distance from the contour to the center of sEVs on the PM was measured, and the number densities of pairwise distances were plotted at a given distance as previously reported[41–44]. The number densities at each distance were normalized to those of randomly generated points in silico; therefore, *y*-axis values >1 in the negative *x*-axis region (color-highlighted area in Fig. 5b) indicate that sEV particles are enriched within the membrane invagination. As a positive control, SF650T-labeled transferrin, a well-known protein internalized through clathrin-mediated endocytosis[39], showed strong

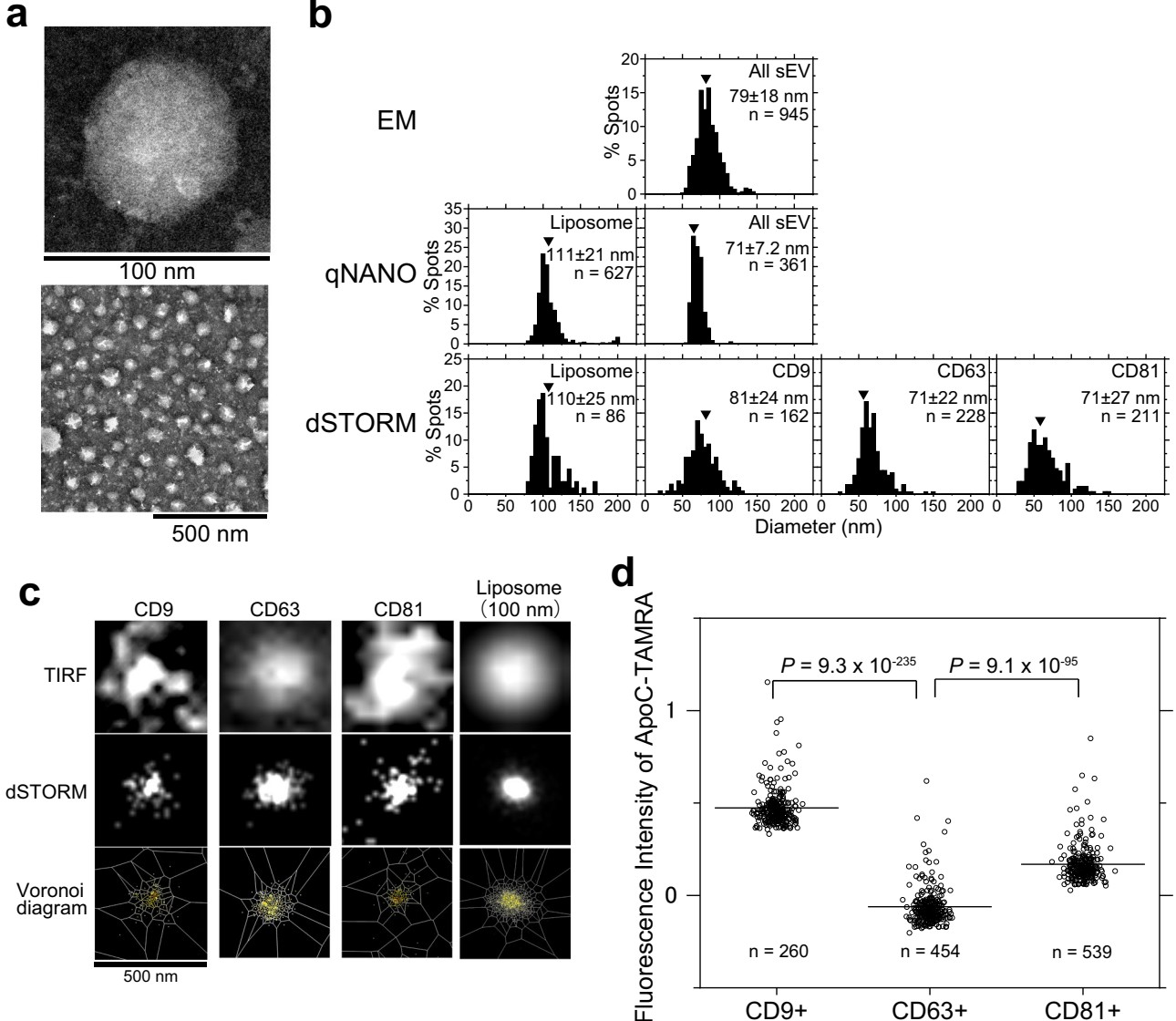

**Fig. 2 | sEV subtypes have similar diameters but different membrane defects.**
**a** Representative magnified view (top) and wide-field view (bottom) of PC-3 cell-derived sEV particles visualized with the negative-stain transmission electron microscope (TEM), selected from 24 independent TEM fields. **b** sEV diameters were determined using TEM (top), qNANO (middle), and direct stochastic optical reconstruction microscopy (dSTORM) (bottom). Numbers indicate means ± SD. "*n*" indicates the number of examined sEV particles derived from PC-3 cells. **c** Size analysis using dSTORM. Typical images of an sEV containing CD9, CD63, and CD81 stained with SaraFluor™650B (SF650B) via Halo7 and a DMPC liposome

containing GM3 conjugated with SF650B at the terminal sialic acid observed by TIRF microscopy. dSTORM images were determined using the Voronoi-based segmentation method (see Methods for details). **d** Distribution of fluorescence intensities of the C-terminal region of apolipoprotein A-I labeled with 5-carboxytetramethylrhodamine (ApoC-TAMRA) on individual PC-3 cell-derived sEVs containing CD9, CD63, and CD81 labeled with SaraFluor™650T (SF650T) via Halo7 isolated by ultracentrifugation. "*n*" indicates the number of examined sEV particles. Statistical analyses were conducted as described in "Statistics and Reproducibility" in the Methods section.

colocalization with AP2α-labeled clathrin-coated pits in PZ-HPV-7 cells (Fig. 5b, c). Additionally, we observed that LAMP-2C molecules were absent from these clathrin-coated pits (Fig. 5b, c), providing additional evidence that LAMP-2C functions as a marker for clathrin-independent endocytosis. None of the sEV subtypes derived from PC-3 or 4175-LuT cells were concentrated in the AP2α-labeled clathrin-coated pits in PZ-HPV-7 or WI-38 cells, respectively (Fig. 5b, c). In contrast, all sEV subtypes derived from both cell lines were heavily concentrated in LAMP-2C-labeled clathrin-independent endocytic regions in both recipient cell types. Notably, this extensive colocalization was almost completely eliminated after treatment with a membrane-impermeable galectin-3 inhibitor (Fig. 5b, c), suggesting that the colocalization occurs through the interaction between galectin-3 on the sEV surface and LAMP-2C in the PM. Interestingly, only the CD63-Halo7-containing

sEV subtype derived from PC-3 cells colocalized with CAV1-labeled caveolae on PZ-HPV-7 cells (Fig. 4d, middle; Fig. 5b, c, top-middle). Furthermore, sEVs containing CD81-Halo7-SF650T, namely, the sEV subtype containing all three marker proteins derived from 4175-LuT cells (shaded circles in the Venn diagram in Fig. 1f) was enriched in caveolae on WI-38 cells (Fig. 5b, c, bottom-middle). The enrichment of sEVs in the caveolae of recipient cells was sEV subtype-dependent (Fig. 1f, 4d, 5b, c). The frequencies of the colocalization events (≥1 frame) between sEVs and clathrin-independent endocytic regions or caveolae were 2.12 ± 0.88 events/μm²/min (mean ± SE) and 0.15 ± 0.12 events/μm²/min, respectively. Moreover, the number densities of sEVs, which showed colocalization with clathrin-independent endocytic regions or caveolae, were 0.090 ± 0.022 particles/μm²/min and 0.031 ± 0.022 particles/μm²/min, respectively. These results indicate

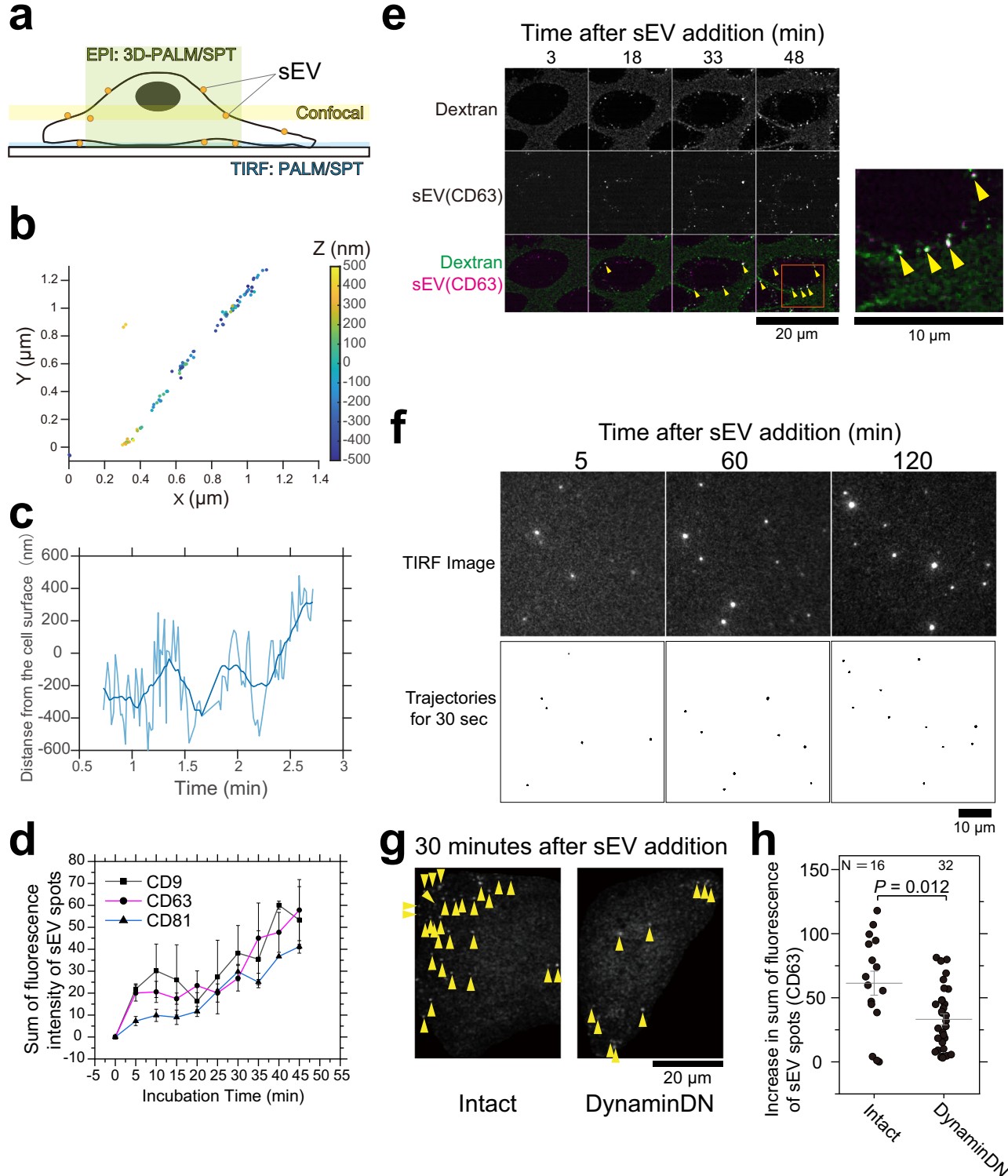

that sEVs colocalize with clathrin-independent endocytic regions more frequently than with caveolae.

Simultaneous two-color 3D super-resolution images of caveolae and PC-3-derived sEVs containing CD63-Halo7-SF650T revealed that the sEV particles colocalized with caveolae on the PZ-HPV-7 cell PM, parallel and perpendicular to the PM (Fig. 5d). Our dSTORM observations revealed that 91% of CD63-containing sEVs derived from PC-3 cells had a diameter of ≤90 nm (Fig. 2b). Given that caveolae have an average diameter of approximately 90 nm[45], sEVs with a diameter of ≥90 nm are unlikely to undergo caveolae-mediated endocytosis and

may be preferentially internalized via clathrin-independent endocytosis.

Next, we investigated the fate of intracellular sEVs after they were internalized. Two-color 2D simultaneous observation of single sEV particles and super-resolution dSTORM imaging of the recycling endosome marker mEos4b-Rab11 suggested that sEVs colocalized with recycling endosomes (Fig. 5e, top). Furthermore, 3D observation (Fig. 5e, bottom) quantified that 20–40% of all sEV subtypes colocalized with recycling endosomes 3–6 h after the addition of sEVs to PZ-HPV-7 cells (Fig. 5f) although whether sEVs fused with recycled

**Fig. 3 | Endocytosis of sEVs occurs through dynamin-regulated machinery. a** Intracellular regions observed by microscopy. **b** 3D trajectory of a single sEV particle (CD63-Halo7-SF650T derived from PC-3 cells) near the PZ-HPV-7 cell PM under oblique-angle illumination. Z-coordinates are shown using a color bar on the right. **c** Time course of the z-coordinate for the trajectory in (**b**). **d** Time course for the sum of fluorescence intensities of individual sEV particles containing CD9-Halo7, CD63-Halo7, and CD81-Halo7-TMR in PZ-HPV-7 cells. Images were acquired at approximately -1.5 μm above the glass by confocal microscopy. Number of cells: 7 (CD9), 3 (CD63), 10 (CD81). **e** Representative confocal microscopy image sequences of pHrodo-Dextran and sEV-CD63-Halo7-TMR at -1.5 μm above the glass (left). After the internalization of pHrodo-Dextran, the fluorescence intensity of pHrodo is increased due to the decrease of pH in the surrounding environment, and fluorescent puncta appeared in cells with a dark background. Because pHrodo-Dextran was always present during the observation, pHrodo-Dextran emitted low-intensity fluorescence outside the cells. A magnified image of the orange rectangle

in the merged image at 48 min after dextran or sEV was added (right). Colocalization between the puncta of pHrodo-Dextran and sEV-CD63-Halo7-TMR is indicated by yellow arrowheads. **f** Time series of single-molecule sensitivity TIRF images of sEVs containing mGFP-GPI attached to the recipient cell PM (upper) and trajectories of mGFP-GPI molecules for 30 s (bottom). **g** Representative single-particle images of sEV-CD63-Halo7-TMR attached to intact PZ-HPV-7 cells or cells overexpressing a dominant negative mutant of dynamin2 (DynaminDN). Yellow arrowheads indicate sEV particles. Images were acquired at approximately -1.5 μm above the glass by confocal microscopy, as shown in (**e**). Due to low sensitivity, individual sEV particles attached to the glass were undetectable under the given observation conditions, while only internalized, aggregated EVs were observed. **h** Increase in the sum of fluorescence intensity of individual sEV-CD63-Halo7-TMR particles in intact or DynaminDN-expressing PZ-HPV-7 cells 30 min after sEV addition. In Fig. 3, data were presented as means ± SE. Statistical analysis was performed using Welch's two-sided *t*-test. "*N*" indicates the number of examined cells.

---

endosomal membranes was unclear. Because the number of fluorescently labeled Rab11 molecules per recycling endosome was estimated at $17.6 \pm 3.7$ by the previously reported method, almost all recycling endosomes were visualized. The degree to which different sEV subtypes colocalized with the recycling endosome marker was indistinguishable (Fig. 5f). Our recent study revealed that in tumor cell-derived sEVs, integrin heterodimers specifically bind to laminin but scarcely bind to fibronectin on recipient cell PMs[46]. Consistently, internalized integrin α6, a component of the laminin receptor (α6β1 and α6β4), in cells that adhere to laminin is readily transported to recycling endosomes, while integrin α5, a component of the fibronectin receptor (α5β1), is not transported[47]. These prior findings, along with our present investigation and our recent work[46], imply that some fractions of sEV particles bound to laminin on cell PMs are internalized and subsequently transported to the recycling pathway, either in an intact state or as membrane-fused entities.

## Binding of sEVs to recipient cell PM induces an integrin-based signaling platform

Next, we aimed to elucidate the mechanisms responsible for the specific colocalization of sEVs containing CD63 derived from PC-3 cells with caveolae. We investigated whether molecular assemblies are formed in the PM of PZ-HPV-7 cells just underneath the sEV binding site to facilitate colocalization of the sEVs with caveolae. Single-molecule imaging is a powerful tool for detecting transient and/or rare events in living cells[37,39,48-53]. Therefore, we simultaneously observed single particles of sEV subtypes and single membrane molecules in the PM and quantified the colocalization by cross-correlation-based method (Supplementary Fig. 10). The lipid composition of caveolae is similar to that of lipid rafts[54], and GPI-anchored proteins, a representative raft marker, are concentrated in caveolae upon crosslinking (Supplementary Fig. 11) as previously reported[55]. Dual-color single-molecule imaging revealed that single Halo-GPI molecules were enriched only in the PM area underneath CD63-Halo7-SF650T-containing sEVs, but not in the PM area underneath sEVs containing CD81- or CD9-Halo7-SF650T (Fig. 6a, b and Supplementary Movie 4). These results suggest that the PM underneath CD63-sEVs exhibits a stronger affinity for raft markers than the PM underneath sEVs containing CD81 or CD9, leading to the enrichment of CD63-sEVs in caveolae.

Caveolin-1-dependent endocytosis of integrin β1 has been reported[56]; therefore, we aimed to examine integrin β1 recruitment to the membrane underneath different sEV subtypes. Our results revealed that integrin β1 was frequently and transiently recruited to and enriched in the PM underneath sEVs of all the subtypes (Fig. 6a, b and Supplementary Movie 5), showing that recruitment of integrin β1 is not the reason for the enrichment of sEVs in caveolae. Simultaneous observation of PALM movies and single sEV particles revealed that talin-1, an inside–out signaling molecule that triggers integrin activation[57], was also enriched in the PM underneath CD63-Halo7-

SF650T-containing sEVs (Fig. 6c and Supplementary Movie 6). These results suggest that tiny integrin β1 clusters are formed in the PM underneath sEVs, which induces tiny cluster formation of talin-1, leading to integrin activation. This quantitative single-molecule analysis revealed that integrin and talin-1 molecules were locally enriched in immediate proximity to sEVs. The tiny and transient nature of integrin and talin-1 clusters formed in the PM beneath the sEVs is further suggested by the lack of significant colocalization of integrin and talin-1 with sEVs, as observed through dual-color conventional immunocytochemical analysis (Supplementary Fig. 12). Recently, we demonstrated that integrin heterodimers in all sEV subtypes containing CD63, CD81, or CD9 bind to laminin on recipient cell PMs[46]. Consequently, integrin β1, which is recruited to the PM underneath sEVs, likely interacts with laminin, suggesting that a symmetrical integrin structure forms through laminin between sEVs and the recipient cell PM (Fig. 6d). Importantly, this symmetrical structure appears to be consistent across all examined sEV subtypes.

## Paracrine sEV binding triggers the Ca²⁺ response in recipient cells

Integrin clustering triggers the recruitment of structural and signaling proteins, which leads to the formation of an adhesion complex[58,59]. This process triggers intracellular signaling, such as the intracellular $Ca^{2+}$ response[60,61]. Therefore, we investigated whether intracellular signaling is induced by the accumulation of integrin underneath sEVs in the recipient cell PM. Initially, we monitored the change in intracellular $Ca^{2+}$ concentration using the Fluo-8H indicator after the addition of sEVs isolated by ultracentrifugation. The intracellular $Ca^{2+}$ concentration transiently increased within a few minutes after the addition of sEVs ($5.0 \times 10^9$ particles/mL at a final concentration lower than the concentration in human blood plasma[62]) (Fig. 7a–c, Supplementary Fig. 13a, and Supplementary Movie 7). In addition, the intracellular $Ca^{2+}$ response was triggered by the addition of sEVs isolated from Tim-4 affinity beads but not by the addition of exomeres[63], which were isolated by treating the supernatant with Tim-4 beads after ultracentrifugation to remove sEVs completely (Fig. 7c). Furthermore, the temporal delay in $Ca^{2+}$ spike initiation inversely correlated with sEV concentration; notably, a reduction in sEV concentration resulted in a prolonged delay (Supplementary Fig. 13b–d). Thus, the intracellular $Ca^{2+}$ response is triggered by the binding of sEVs to the PM and not by contaminating factors. The increase in intracellular $Ca^{2+}$ concentration observed within a few minutes after sEV addition (Fig. 7a, b) indicates that sEV binding induced prompt signaling at the PM, rather than a secondary response after sEV internalization[64,65].

Intracellular $Ca^{2+}$ concentrations in recipient cells were quantified after sEVs derived from three donor cell types were incubated with three recipient cell types (nine combinations) (Fig. 7d). Remarkably, when sEVs were added to the cells that secreted them, no intracellular $Ca^{2+}$ response was detected (Fig. 7d) even though sEVs were bound to

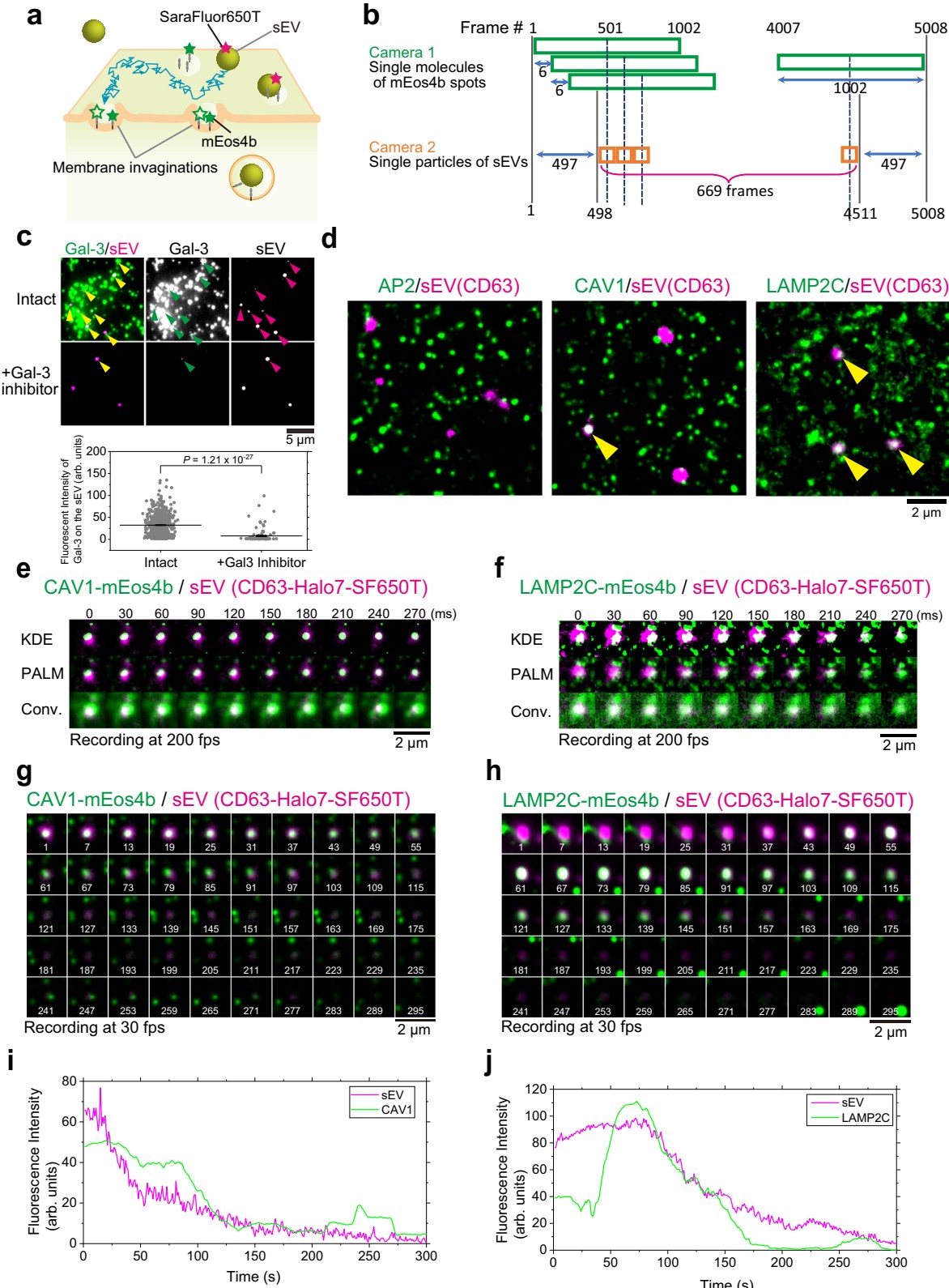

the cell PM (Supplementary Fig. 14a, b, open symbols). By contrast, sEVs derived from two cancer cell lines, PC-3 and 4175-LuT, as well as primary mesenchymal stem cells (MSCs) bound to different recipient cell PMs, induced intracellular Ca²⁺ responses in the recipient cells (Fig. 7d). When sEVs did not bind to recipient cells, no Ca²⁺ response was observed (4175-LuT-derived sEVs vs. PZ-HPV-7 cells) (Fig. 7d and Supplementary Fig. 14). Therefore, intracellular Ca²⁺ responses in

recipient cells were induced exclusively by the binding of sEVs derived from cells distinct from the recipient cells, a process called "paracrine sEV binding" (here, "paracrine" refers to "secretion exchanges between different cells", irrespective of the distance between them). In contrast, autocrine sEV binding (where sEVs originate from the same cell line as the recipient) did not elicit an intracellular Ca²⁺ response. As noted above, our previous study revealed that integrin composition in sEVs

**Fig. 4 | Endocytosis of sEVs occurs on specific membrane invaginations.**
**a** Schematic diagram showing the simultaneous real-time observation of single sEV particles and super-resolution microscopy images of membrane invaginations on living cell PMs. **b** The method used to create photoactivated localization microscopy (PALM) movies of membrane invaginations superimposed with single-particle movies of sEVs. Data acquisition was performed by observing single-fluorescent molecules of mEos4b at 200 frames/s (5008 frames). The first PALM image was reconstructed using frames 1–1002, which was then repeated by shifting the initial frames backward by 6 frames (30 ms), thus obtaining 669 PALM images. Still, PALM images were connected to form a "PALM movie." The movies of single sEV particles were rolling-averaged for 6 frames and synchronously merged with the PALM movie. **c** (Top) Representative TIRFM images showing immunostained galectin-3 (Gal-3) (green) and sEVs (magenta) on the PZ-HPV-7 cell surface before and after treatment with a membrane-impermeable galectin-3 inhibitor. Arrowheads indicate colocalization. (Bottom) Quantification of the fluorescence intensity of Gal-3 colocalized with sEVs bound to the cell surface before and after inhibitor

treatment. $N = 30$ (Intact), 10 (+Gal-3 inhibitor). **d** Snapshots of PALM movies of membrane invaginations (green) merged with movies of single sEV particles (magenta). Colocalization events are indicated by yellow arrowheads.
**e, f** Representative image sequences of caveolin-1 (CAV1)-mEos4b/sEVs-CD63-Halo7-SF650T (**e**) and lysosome-associated membrane protein-2C (LAMP-2C)-mEos4b/sEVs-CD63-Halo7-SF650T (**f**). Single sEV particle images (magenta) merged with (top) binarized images of caveolae by the kernel density estimation (KDE) method (green), (middle) PALM images, and (bottom) conventional fluorescence images (recorded at 200 Hz). **g, h** Long-term observation of sEV internalization. Snapshots from PALM movies (green) of CAV1-mEOS4b (**g**) and LAMP-2C-mEOS4b (**h**) merged with single sEV particles (magenta). Images were recorded at 30 Hz and are displayed at 6-s intervals. The white numbers at the bottom indicate the time in seconds. **i, j** Time series plots of fluorescence intensities of the sEV particles and CAV1-mEOS4b (**i**) or LAMP-2C-mEOS4b (**j**), shown at 1-s intervals. In Fig. 4, data were presented as the mean ± SEM. Statistical analysis was performed using Welch's two-sided $t$-test.

was critical for their binding to the cell surface[46]. To investigate whether the lack of $Ca^{2+}$ response during autocrine binding was due to low integrin levels in the sEVs, sEVs derived from PC-3 cells overexpressing integrin β1 by a factor of $5.4 \pm 0.5$ ($n = 3$) were applied to wild-type PC-3 cells. However, no $Ca^{2+}$ response was observed (Supplementary Fig. 15). This result indicates that the absence of $Ca^{2+}$ response in autocrine binding is not caused by the low integrin β1 content in the sEVs.

To validate whether the intracellular $Ca^{2+}$ response is triggered by adhesion signaling induced by sEVs on the PM, we examined the $Ca^{2+}$ response in recipient cells overexpressing the dominant negative mutant of talin-1 (TalinDN). The intracellular $Ca^{2+}$ response was notably suppressed, indicating that sEV-mediated adhesion signaling induced the $Ca^{2+}$ response (Fig. 7e).

Because the Src family kinases, phospholipase Cγ (PLCγ) and focal adhesion kinase (FAK) have been reported to be involved in integrin adhesion signaling[61,66,67], we examined whether these signaling molecules trigger an intracellular $Ca^{2+}$ response after integrin is engaged through sEV binding. Our results indicate that sEV-induced intracellular $Ca^{2+}$ responses are markedly abrogated by inhibiting Src family kinase and PLCγ activity with PP2 and U73122, respectively, but not by their control analogs (Fig. 7e). To confirm whether Src family kinase in the recipient cell is activated after the addition of sEVs, the phosphorylation of FAK at tyrosine residue 861 (Tyr-861), a substrate of Src family kinase was assessed by biochemical assay. Our results revealed a significant increase in FAK-Y861 phosphorylation within minutes of sEV addition, whereas PP2 treatment reduced the phosphorylation more substantially than PP3 treatment (Supplementary Fig. 16). These results indicate that Src family kinase in the recipient cell is activated upon sEV addition. Furthermore, single-molecule imaging of PLCγ2 showed that the number of PLCγ2 molecules recruited to the PM significantly increased after sEV addition (within 1.5 min), but the increase was notably suppressed after treatment with PP2 but not PP3 (Fig. 7f, g). The recruitment of PLCγ2 to the PM after sEV addition was undetectable by confocal microscopy (Supplementary Fig. 17), suggesting that only a small fraction of cytosolic PLCγ2 was recruited to the PM. Therefore, these results indicate that sEV binding activates Src family kinases and subsequently induces PLCγ2 recruitment to the PM, which may trigger the $Ca^{2+}$ response. To further validate whether PLCγ2 recruitment to the PM triggers intracellular $Ca^{2+}$ signaling, simultaneous imaging of $Ca^{2+}$ concentration was performed at 0.3 s/frame, a temporal resolution 60-fold higher than in other observations, and single molecules of PLCγ2 at a video rate. The time course of $Ca^{2+}$ concentration (Supplementary Fig. 18, top) and the cumulative number of PLCγ2 spots recruited to the PM (Supplementary Fig. 18, bottom) revealed that the $Ca^{2+}$ mobilization occurred around the time when the cumulative number of the recruited PLCγ2 reached a plateau in all three cases (Supplementary Fig. 18). Together with experiments

with a PLC inhibitor (U73122), these results indicate that PLCγ2 initiated the calcium response after sEV stimulation.

According to the previous study[68], the FAK autophosphorylation site Tyr-397 mediates binding with the C-terminal SH2 domain of PLCγ1. Building upon our results, we hypothesized that Src-activated PLCγ molecules are recruited to FAK underneath sEVs bound to the PM. However, single molecules of FAK, PLCγ1, and PLCγ2 were not enriched in the PM underneath sEVs (Fig. 7h), suggesting that the integrin and talin-1 clusters underneath sEVs do not serve as platforms for FAK and PLCγ signaling. Conversely, PALM movie analysis revealed that single PLCγ2 molecules were frequently recruited to FAK clusters (Fig. 7i, j), suggesting that integrin adhesion signaling propagates from the sEV binding site to FAK clusters that are formed irrespective of sEV binding.

The $Ca^{2+}$ response was attenuated by an inhibitor of the inositol 1,4,5-triphosphate ($IP_3$) receptor ($IP_3R$) (Fig. 7e), demonstrating that in the ER, $IP_3R$ mediates the $Ca^{2+}$ response. These results suggest that PLCγ generates $IP_3$ from phosphatidylinositol bis(4,5)phosphate [$PI(4,5)P_2$], and the increase in cytosolic $IP_3$ concentration triggers $Ca^{2+}$ release from the ER through $IP_3Rs$[48,50,69].

## sEV-induced $Ca^{2+}$ response facilitates sEV uptake

We examined the biological significance of the intracellular signals induced by sEV binding. Using the same experimental strategy in Fig. 3e, we found that sEV uptake into cells was considerably diminished after overexpression of TalinDN or inhibition of dynamin, galectin-3, PLCγ, Src family kinases, and $IP_3R$ activity (Fig. 8a, b). The same treatments of the cells reduced dextran uptake (Fig. 8c), while treatment with the galectin-3 inhibitor only moderately decreased dextran uptake, likely because of alternative internalization pathways for components other than EVs. Therefore, the uptake of both sEVs and dextran is facilitated by the intracellular $Ca^{2+}$ response induced by sEV binding to the PM (Fig. 7). Although Figs. 3g, h, 8a–c (data acquired using Dynasore) clearly show that sEVs are internalized in a dynamin activity-dependent manner, $Ca^{2+}$ does not directly regulate dynamin activity because dynamin does not contain a $Ca^{2+}$-binding site. Instead, a ubiquitously expressed $Ca^{2+}$-dependent phosphatase, calcineurin[70], is known to dephosphorylate dynamin, leading to enhanced GTPase activity of dynamin in various cell types[71,72]. Therefore, we used the calcineurin inhibitor FK506 to test whether calcineurin activates dynamin and facilitates the uptake of sEVs and dextran upon $Ca^{2+}$ mobilization. Although sEV binding induced a normal $Ca^{2+}$ response even after cells were treated with FK506 (Fig. 8d), the uptake of sEVs and dextran was largely reduced (Fig. 8b, c). These results show that the $Ca^{2+}$ response induced by sEVs increases the activities of calcineurin and dynamin, facilitating sEV uptake.

The internalization activity of caveolae can also be directly regulated by Src family kinases[73]. To assess whether caveolae

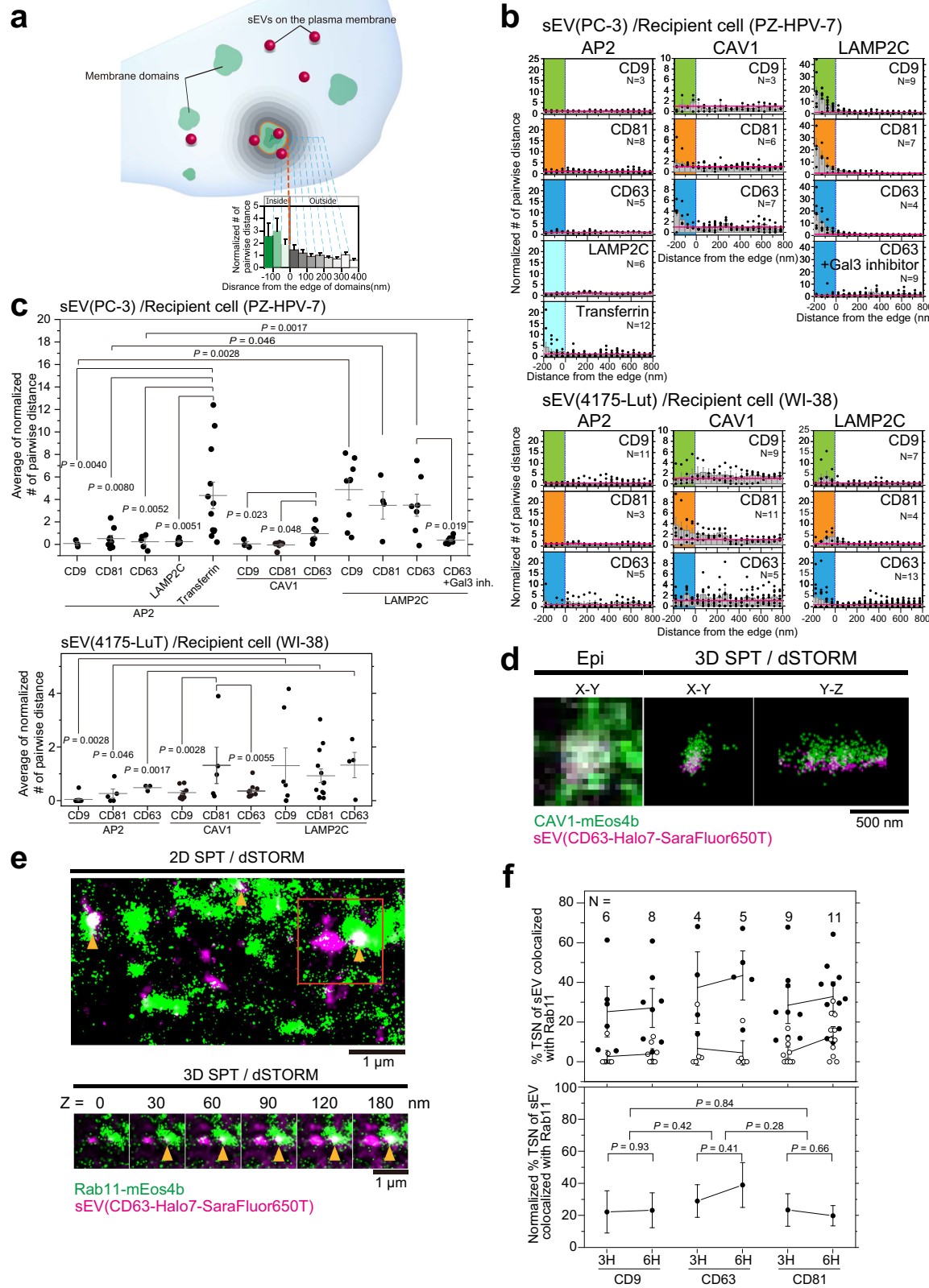

internalization is also directly regulated by Src family kinase, simultaneous single-particle imaging of CD63-Halo7-SF650T containing sEVs and PALM movie observation of caveolae was performed in cells treated with the inhibitor (PP2). Our results revealed that the colocalization of sEVs with caveolae was slightly reduced in cells treated with PP2 compared with PP3. These results suggest that, in addition to the activation of calcineurin-dynamin by the elevation of intracellular $Ca^{2+}$

concentration, Src family kinase activity may also partially promote sEV uptake via caveolae (Supplementary Fig. 19).

## Discussion

The mechanisms governing the uptake of tumor-derived sEVs by recipient cells have not been directly observed[5]. To address this issue, we developed a cutting-edge super-resolution movie technique to

**Fig. 5 | Colocalization analysis of sEVs and membrane invaginations. a** Method used to analyze colocalization. Normalized relative frequency was defined as the ratio of the average value of the pair correlation function of actual images to that of randomly distributed spots generated by a computer. Zero on the *x*-axis indicates the contour of the membrane area in PALM images determined by the KDE method. When sEVs are enriched in the membrane area, the normalized number of pairwise distances is >1. **b** Probability density analysis of sEVs, individual LAMP-2C or transferrin molecules, and membrane invaginations. Colored areas indicate regions within membrane invaginations. All plots were calculated from more than 1000 colocalization events. **c** Enrichment of sEVs in membrane invaginations. The average value of the normalized number of pairwise distances from −200 to 0 nm in Fig. 5b was calculated for each cell and plotted in the graphs. Data were presented as the means ± SE. The number of cells is indicated in Fig. 5b. **d** Representative 3D-

dSTORM of CAV1-mEos4b (green) and single-particle images of PC-3 cell-derived sEV-CD63-Halo7-SF650T (magenta) in PZ-HPV-7 cells. **e** 2D/3D-dSTORM of Rab11-mEos4b (green) and single-particle images of PC-3 cell-derived sEVs-CD63-Halo7-SF650T (magenta) in PZ-HPV-7 cells. Cropped z-section images (bottom) of the orange rectangle in the 2D projection image (top). Colocalization of sEV particles with Rab11-labeled recycling endosomes is indicated by yellow arrowheads. **f** Degree to which sEV-tetraspanin (CD9, CD63, and CD81)-Halo7-SF650T colocalized with Rab11-mEos4b, a recycling endosome marker. Closed and open circles indicate raw and randomized data, respectively (top). Net values were estimated by subtracting randomized data from raw data (bottom). In Fig. 3, "*N*" indicates the number of examined cells. Data were presented as the means ± SE. Statistical analyses were conducted as described in "Statistics and reproducibility" in the Methods section.

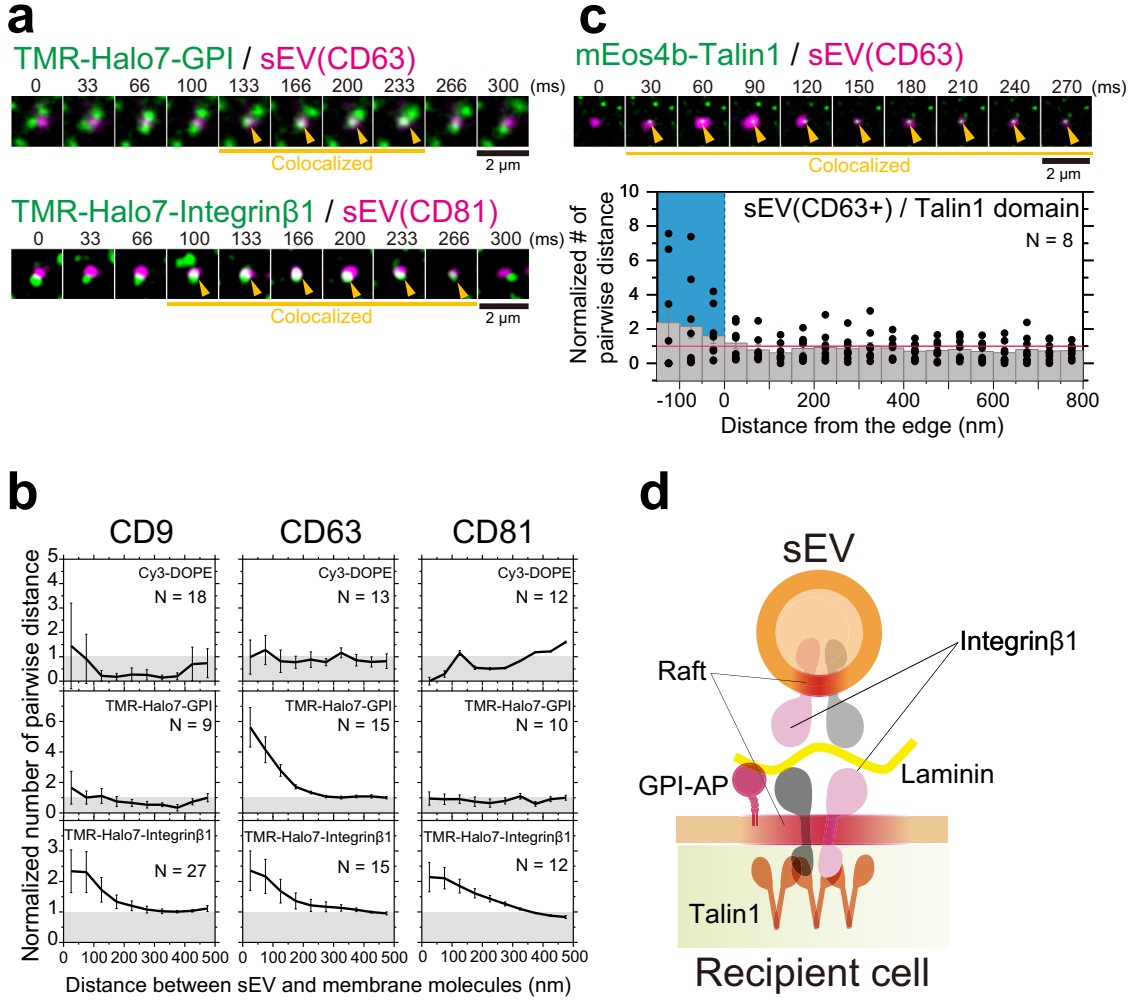

**Fig. 6 | Signaling platform induced by sEV binding to the PM of recipient cells. a** Typical image sequence (every 33 ms) of single molecules of TMR-Halo7-GPI (top) or TMR-Halo7-integrin β1 (bottom) and a single particle of sEV-CD63-Halo7-SF650T or sEV-CD81-Halo7-SF650T on the living PZ-HPV-7 cell PM, which were obtained by TIRFM. Yellow arrowheads indicate colocalization events. **b** Colocalization analysis of single sEV particles with single molecules of Cy3-DOPE, TMR-Halo7-GPI, or TMR-Halo7-integrin β1 in the cell PM. "*N*" indicates the number of examined cells. Data at each distance are presented as the means ± SE. **c** Talin1-enriched domains were induced underneath the sEVs. Image sequence (every 30 ms) of the PALM image of mEos4b-talin 1 and a single sEV particle (top) and density analysis (bottom). "*N*" indicates the number of examined cells. Data at each distance are presented as the means ± SE. **d** Schematic diagram showing that integrin β1, talin-1, and GPI-anchored protein (GPI-AP) are enriched in the recipient cell PM underneath the laminin-bound sEV.

visualize membrane invaginations in living cells, with high spatial and temporal resolution, and simultaneously, performed single sEV particle imaging. First, we demonstrated the presence of distinct subtypes of tumor-derived sEVs (Fig. 1). Second, we discovered that all sEV subtypes derived from two distinct tumor cell types were internalized via clathrin-independent endocytosis mediated by galectin-3 and LAMP-2C, whereas some sEV subtypes were internalized through caveolae (Figs. 4, 5). Intriguingly, sEVs internalized through caveolae exhibited smaller membrane defects than other sEV subtypes (Fig. 2) and induced raftophilic domains underneath them in the cell PMs

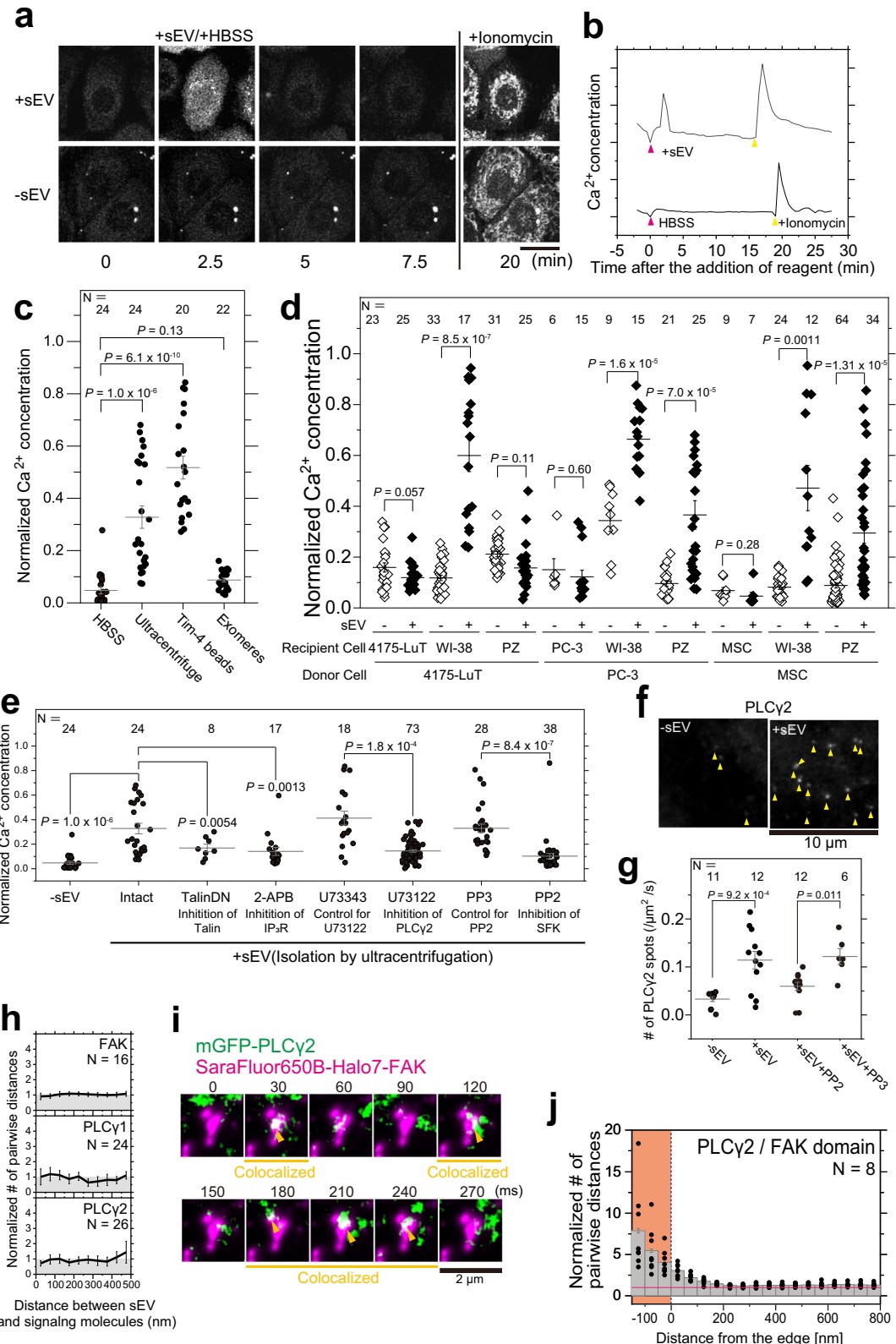

(Fig. 6). Third, we found that paracrine sEV binding, rather than autocrine binding, triggered intracellular $Ca^{2+}$ signaling (Fig. 7). This process was induced by adhesion signaling after integrin β1 and talin-1 clusters formed underneath the sEVs (Figs. 6, 8e). The cluster formation induced the activation of Src family kinases and recruited PLCγ2 to FAK clusters, which facilitated the digestion of $PI(4,5)P_2$ to produce $IP_3$ and led to $Ca^{2+}$ mobilization through $IP_3R$ in the ER (Figs. 7, 8e). An

increase in intracellular $Ca^{2+}$ enhanced dynamin activity through calcineurin, which facilitated the internalization of sEVs (Figs. 3, 8e). These stochastic molecular events could only be elucidated with our imaging technique.

Several pathways for the internalization of sEVs, including clathrin-coated pits, caveolae, and clathrin-independent endocytosis, have been reported[8]. However, a consensus has not been reached

**Fig. 7 | sEV binding triggered intracellular Ca²⁺ signaling. a** Typical confocal microscopy images of Fluo-8H-loaded PZ-HPV-7 cells. sEVs (top) or HBSS (bottom) were added to the cells 2.5 min after the initial observation. Ionomycin was added at 20 min. Scale bar: 20 μm. **b** Time course of the fluorescence intensity of Fluo-8H in (a). **c** Normalized Ca²⁺ concentration after the addition of HBSS, sEVs isolated by ultracentrifugation, sEVs isolated using Tim-4 beads, and sEV-depleted culture medium (exomeres). **d** Normalized Ca²⁺ concentration after the addition of sEVs (5.0 × 10⁹ particles/mL at final concentration) (closed diamond) derived from three donor cell types to three recipient cell types (9 combinations). sEV(−) indicates the results obtained after the exomeres were added (open diamond). **e** Normalized Ca²⁺ concentration of PZ-HPV-7 cells in which talin-1 activity was inhibited by expressing a dominant negative mutant (TalinDN) or treating the cells with inhibitors of IP₃ receptor (IP₃R) (2-APB), phospholipase C (PLC) (U73122), Src family kinase (SFK) (PP2), a PLC control reagent (U73343), or an SFK control reagent (PP3) after the addition of PC-3 cell-derived sEVs. sEV(-) indicates the results obtained after the

exomeres were added. **f** Typical single molecules of mGFP-PLCγ2 (indicated by yellow arrowheads) were frequently recruited to the PZ-HPV-7 cell PM within 90 s after the addition of PC-3 cell-derived sEVs. The images were obtained by TIRF microscopy. **g** Number of mGFP-PLCγ2 spots recruited to the PM before and after the sEVs were added in the absence or presence of PP2/PP3. **h** Colocalization analysis of single sEV particles with single molecules of membrane-associated focal adhesion kinase (FAK), PLCγ1, and PLCγ2 in the cell PM. **i** Typical image sequence (every 30 ms) of dSTORM images of SF650B-Halo7-FAK domains (magenta) and single molecules of mGFP-PLCγ2 (green). Yellow arrowheads indicate colocalization events. **j** PLCγ2 molecules were frequently recruited to FAK-enriched domains. Density analysis of single molecules of mGFP-PLCγ2 on the SF650B-Halo7-FAK domains. In Fig. 7, "*N*" indicates the number of examined cells; data were presented as the means ± SE. Statistical analyses were conducted as described in "Statistics and reproducibility" in the Methods section.

regarding the predominant pathways involved, possibly due to difficulties in experimental methods[2,5]. For example, the bypass pathway can be overemphasized, as observed in knockout experiments in which sEV uptake was enhanced in CAV1-KO cells[74]. A confocal microscopy-based study reported that within minutes of addition, sEVs are internalized as single particles without accumulation from a specific site on the PM through unclear mechanisms[75,76]. Therefore, it was necessary to directly observe and quantify sEV uptake by live recipient cells. Our state-of-the-art microscopy techniques with high temporal and spatial resolution enabled accurate determination of the internalization pathway of distinct sEV subtypes in living recipient cells (Figs. 4, 5). Although a recent review article proposed that sEV molecules are redistributed to recycling endosomes in recipient cells[5], no direct evidence has been presented. Our results using 3D-dSTORM and single-particle imaging indicated that 20–40% of internalized sEVs were transported into the recycling pathway (Fig. 5e, f), suggesting the following potential scenarios–either the internalized sEVs are secreted again from the recipient cell surface or their membranes fuse, leading to the release of sEV cargos into the cytoplasm. In addition, membrane molecules, including integrins, are recycled on the recipient cell PM (Fig. 8e).

Galectin-3 binds to N-acetyllactosamine (LacNAc) and associates with various glycoproteins, such as integrins[77], and glycolipids, including the ganglioside GM1[78]. Both integrin heterodimers and GM1 are present in cancer cell-derived sEVs, with GM1 density in these sEVs being approximately 30 times higher than in the cancer cell PM[46]. Furthermore, experiments using de novo glycan displays on cell surfaces demonstrated that galectin-3 strongly binds to highly branched, complex-type N-linked glycans, including LacNAc[79]. Furthermore, highly branched, complex-type N-linked glycans containing LacNAc are more abundantly expressed in cancer cell PMs than in normal cell PMs[80] and are more extensively concentrated in sEVs than in their parental cancer cells[81]. These recent findings suggest that the environment on cancer-derived sEV membranes generally promotes galectin-3 binding, and the galectin-3-mediated interaction with LAMP-2C in recipient cell PMs may be a common feature. Treatment with a membrane-impermeable galectin-3 inhibitor significantly reduced sEV internalization (Fig. 8a, b), suggesting that the interaction between galectin-3 and LAMP-2C in the recipient cell PM beneath sEVs may be a crucial initial step in sEV internalization.

Our results indicate that a GPI-anchored protein, which preferentially localizes within a raft-like domain (liquid-ordered [Lo] like domain), was enriched beneath CD63-containing sEVs characterized by lower membrane fluidity in sEVs derived from PC-3 cells. Thus, raft-like domains may symmetrically form on both the sEV membrane and cell PM directly beneath the sEVs (Fig. 6d). This symmetrical structure is plausible, as focal adhesions are highly ordered structures similar to rafts[82] and certain integrin subunits, such as α6 and β4, undergo palmitoylation and partition into rafts[83,84]. A potential mechanism for this

symmetrical structure could be the associations of raftophilic molecules between the sEV membrane and the cell PM. For example, glycosphingolipids, typical raft markers, can associate with one another between membranes[85]. Galectin-3 may mediate these interactions because it binds to numerous glycolipids and glycoproteins. However, more investigations are warranted to elucidate the exact mechanisms underlying this symmetric structure.

Integrin cluster formation in the cell PM can trigger adhesion signaling, which elicits intracellular calcium responses[61] and induces the recruitment of downstream molecules[60,86]. We observed the recruitment of PLCγ2 and initiation of the Ca²⁺ response within 1–5 min of sEV addition, indicating prompt initiation of adhesion signaling (Fig. 7 and Supplementary Fig. 18), which leads to secondary signaling responses, such as release and functional expression of miRNAs and proteins following sEV uptake. To our knowledge, this study is the first to report that sEVs induce rapid and immediate signaling. For example, tumor-derived sEVs containing PD-L1 interacted with PD-1 expressed in T-cell PMs, resulting in the inhibition of T-cell activation[7,87]. However, cellular responses, such as IL-2 secretion, were evaluated 24–48 h after sEV addition. Incubation of WI-38 cells with 4175-LuT-derived sEVs increased the expression and phosphorylation of Src family kinases in WI-38 cells[23]; however, these signals were not promptly transduced because expression level was examined in fixed cells 2 h after sEV addition. Hence, our discovery demonstrates, for the first time, that the binding of tumor-derived sEVs promptly initiates intracellular signal transduction via adhesion molecules in the recipient cell PM. Interestingly, only small numbers of sEVs (approximately 8 sEV particles/100 μm², namely 200 sEV particles/cell) were sufficient to trigger the signal transduction. Furthermore, these signaling events were not only mediated by tumor-derived sEVs, but also by primary mesenchymal stem cell-derived sEVs (Fig. 7d).

We discovered a mechanism by which paracrine sEV binding induces adhesion signals and facilitates their internalization (Fig. 8). This process involves the induction of intracellular Ca²⁺ mobilization by adhesion signaling, which enhances dynamin activity through calcineurin near the PM and promotes endocytosis[71,72] (Fig. 8e). Notably, fusion events between sEVs and recipient cell PMs were not observed (Fig. 3f). The internalization of sEVs by recipient cells is essential for incorporating bioactive components of sEVs into the cytosol and modifying the characteristics of recipient cells. Therefore, we have demonstrated the physiological significance of sEV-induced signaling. However, factors that lead to the lack of intracellular signaling induced by autocrine sEV binding (Fig. 7d) are unknown. We found that integrin heterodimers in sEVs specifically bind to laminin on the recipient cell PM[46], and integrin heterodimer clusters likely occur on the cell PM and in sEVs across laminins (Fig. 6d). However, we observed that autocrine sEV binding does not induce intracellular signal transduction, suggesting that factors other than integrin heterodimers are necessary to initiate sEV-mediated intracellular signaling.

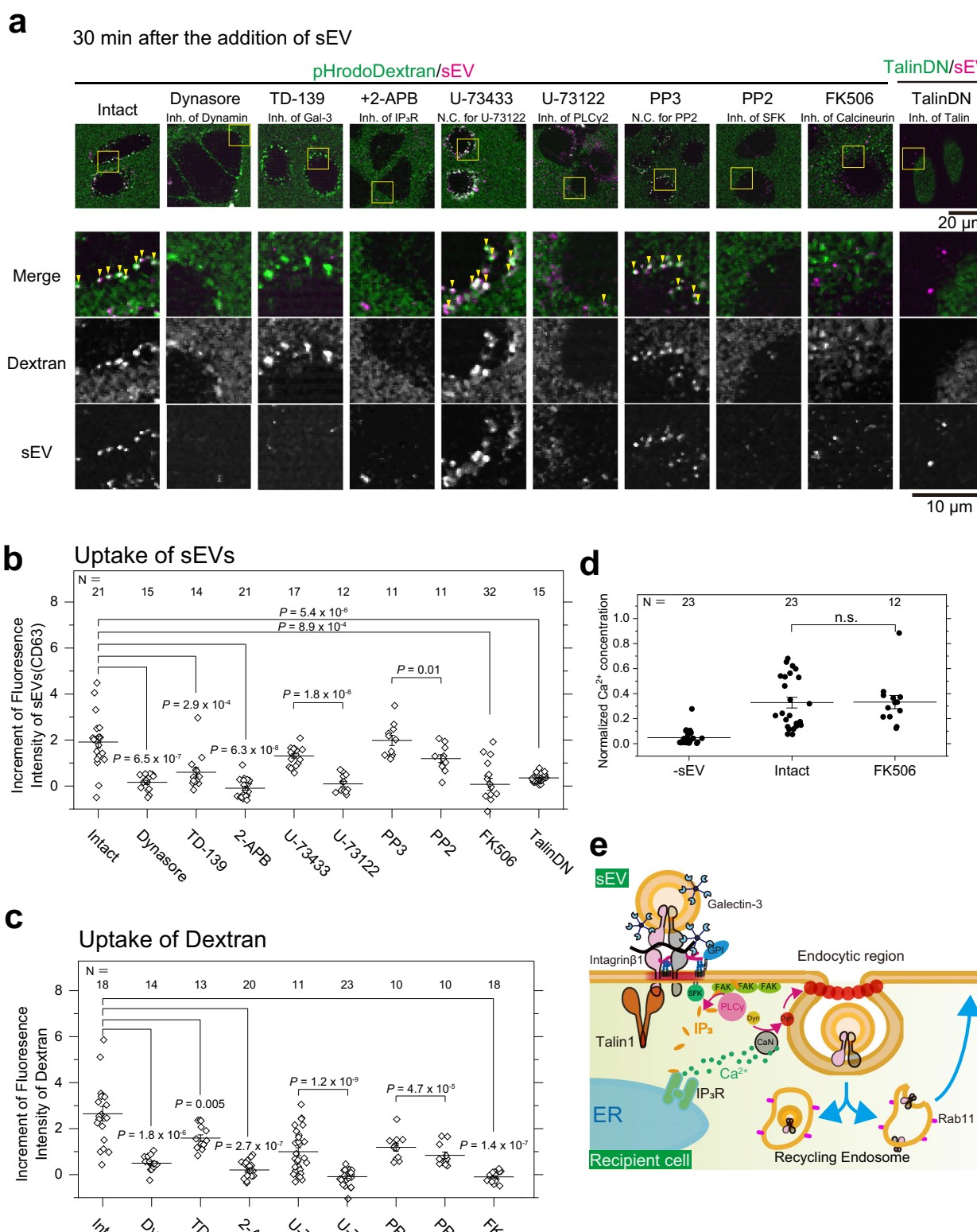

**a** 30 min after the addition of sEV

**b** Uptake of sEVs

**c** Uptake of Dextran

**d**

**e**

We elucidated the signaling pathway induced by paracrine sEV adhesion in detail. Interestingly, PLCγ2 was recruited to FAK clusters, which differ from integrin and talin-1 clusters (Fig. 7i, j). FAK is a scaffold protein that self-associates and interacts with cellular partners through its FERM domain[88]. FAK clusters, also known as nascent adhesions, have recently been identified as synergistic liquid–liquid phase separated (LLPS) regions[89]. Therefore, PLCγ2 may function

effectively in the PM by capitalizing on the LLPS structure (Fig. 8e). However, it is unclear how signals are transduced from integrin and talin-1 clusters to FAK clusters. Src family kinases are downstream of integrin and talin-1, and Src activity is necessary for PLCγ2 recruitment[61]. In B cells, FAK clusters were isolated from the BCR cluster, which serves as the signaling entry point, and Src family kinases connected these signaling hubs[42]. Our results, combined with

**Fig. 8 | sEVs induced intracellular signaling, which facilitated the endocytosis of sEVs. a** Typical confocal microscopy images of cells 30 min after the addition of pHrodo-Dextran (green) and sEV-CD63-Halo7-TMR (magenta). Cells were treated with the following compounds or expressed a dominant negative mutant: a Dynamin inhibitor (Dynasore), a galectin-3 inhibitor (TD-139), an $IP_3R$ inhibitor (2-APB), a PLC control reagent (U73343), a PLC inhibitor (U73122), an SFK control reagent (PP3), an SFK inhibitor (PP2), a calcineurin inhibitor (FK506), or a dominant negative mutant of talin-1. As described in Fig. 3e, pHrodo-Dextran was present during the observation and also emitted low-intensity fluorescence outside the cells, whereas after internalization, fluorescence intensity increased due to pH reduction and fluorescent puncta appeared in the cells with a dark background. The magnified images of the yellow squares in the top panels are shown in the lower three rows. The colocalization between the puncta of pHrodo-Dextran and sEV-

CD63-Halo7-TMR is indicated by yellow arrowheads in the merged images. **b** Increase in the sum of fluorescence intensity of individual sEV-CD63-Halo7-TMR particles in PZ-HPV-7 cells 30 min after sEVs were added. **c** Increase in the sum of fluorescence intensity of pHrodo-Dextran puncta in PZ-HPV-7 cells 30 min after sEVs were added. **d** Normalized $Ca^{2+}$ concentration in PZ-HPV-7 cells treated with or without 10 µM of the calcineurin inhibitor FK506 after the addition of HBSS (same data as in Fig. 7c) or sEVs derived from PC-3 cells. **e** Schematic diagram showing sEV endocytosis facilitated by intracellular signaling triggered by sEVs through galectin-3, integrin, talin-1, SFK, FAK cluster, PLCγ, $IP_3R$, $Ca^{2+}$, calcineurin (CaN), and dynamin (Dyn). In Fig. 8, "*N*" indicates the number of examined cells. Data were presented as the means ± SE. Statistical analyses were conducted as described in "Statistics and reproducibility" in the Methods section.

---

other reports, suggest that Src family kinases likely play pivotal roles in transmitting these signals. The mechanisms involved should be elucidated in the future. Because we used only nine combinations of cells and sEVs to investigate the uptake mechanism of cancer cell-derived sEVs by normal cells and the signaling mechanism after the sEV uptake, the general applicability should be assessed in the future using a broader range of EV-donor and recipient cell combinations. Nevertheless, our study demonstrated that paracrine sEV binding induces adhesion signaling, which promptly triggers signal transduction at the recipient cell PM underneath sEVs. Increased intracellular $Ca^{2+}$ facilitates sEV uptake by recipient cells. These findings provide insights for the development of strategies to modify recipient cells by incorporating bioactive materials from sEVs. Moreover, this approach holds promise for future therapeutic applications.

## Methods
### Cell culture
Human prostate cancer (PC-3; ATCC, CRL-1435) cells were cultured in HAM F12 medium (Sigma–Aldrich). MDA-MB-231 human breast cancer organotropic line 4175-LuT cells (kindly provided by Dr. Joan Massagué (Memorial Sloan Kettering Cancer Center (MSKCC), New York, NY, USA) were cultured in DMEM (Sigma–Aldrich). Human fibroblast (WI-38; ATCC, CCL-75) cells were cultured in MEM. The culture media for these three cell lines were supplemented with 10% FBS (Gibco), 100 U/mL penicillin, and 100 µg/mL streptomycin (Gibco). Human prostate (PZ-HPV-7; ATCC, CRL-2221) cells were cultured in keratinocyte serum-free medium (Gibco) supplemented with 0.05 mg/mL bovine pituitary extract (Gibco) and 5 ng/mL human recombinant epidermal growth factor (Gibco). Human mesenchymal stem cells (MSCs; Lonza, PT-2501; Lot 23TL142784) were cultured in MSCGM (Lonza, PT-3238) supplemented with MSCGM single-quote supplements and growth factors (Lonza; PT-4105). Cells cultured for two to four passages were used for sEV collection or observation.

### Preparation of cDNA plasmids, transfection, and incorporation of dyes
All plasmids used in this study were constructed with an NEBuilder HiFi DNA assembly system (New England Biolabs) and are listed in Supplementary Table 1. Wild-type PC-3 cells or 4175-LuT cells were transfected with cDNA encoding CD9, CD63, or CD81 tagged with mGFP or Halo7 at the C-terminus (1000 ng cDNA) by using a 4D-nucleofector (LONZA). The transfection of PC-3 cells was performed using SF buffer, DS-150, and 4175-LuT cells using SE buffer, DS-130. Cells that stably expressed proteins tagged with Halo7 or mGFP were established by using G418 (Nacalai Tesque, 0.2 mg/mL for PC-3 and 2 mg/mL for 4175-LuT) or blasticidin (Invitrogen, 2 mg/mL for PC-3 and 1 mg/mL for 4175-LuT), respectively. Cells that stably expressed appropriate amounts of the molecule of interest were selected using a flow cytometer (BD FACSMelody). For observation of a typical non-raft unsaturated phospholipid, Cy3-dioleoylphosphatidyletahnolamine (Cy3-DOPE, synthesized as previously reported[90] on the PM, PZ-HPV-7 cells were

incubated with 5 nM Cy3-DOPE in HBSS (200 µL) for 10 min at room temperature as reported previously[52]. For overexpression experiments, PC-3 cells were transfected with cDNAs encoding integrin β1, CD9-Halo7, or CD81-Halo7 along with transposase, subsequently, cell lines with stable expression were generated[91,92].

### Cell sorter
Cells expressing Halo7-tagged tetraspanins (CD63, CD9, or CD81) were stained with 100 nM TMR-Halo ligand for 30 min at 37 °C. After washing with the culture medium and PBS, cells expressing tetraspanin-mGFP and tetraspanin-Halo7-TMR were harvested using cell scrapers and centrifuged at 1400×*g* for 3 min. To detect the expression of tetraspanins fused with mGFP and Halo7, the cell pellet was analyzed using a FACS Melody cell sorter and FlowJo (BD Biosciences). The gating strategy followed a sequential approach: first, SSC vs. FSC gating was applied to exclude debris, followed by FSC-H vs. FSC-A gating to exclude doublets. Finally, EGFP vs. TMR gating was employed to measure mGFP and TMR intensities in cells, and boundaries for this final gate were established based on an intact cell control.

### Isolation of sEVs from the cell culture supernatant
PC-3 cells, 4175-LuT cells, and MSCs were cultured in two 150-mm dishes and grown to approximately 100% confluence ($1–2 × 10^7$ cells), after which the cell culture medium was replaced with 30 mL of FBS-free medium (DMEM for MSCs). After 48 h of incubation, the cell culture supernatant was collected and subjected to centrifugation at 300×*g* for 10 min at 4 °C to remove cells. Next, to pellet debris and apoptotic bodies, the supernatant was centrifuged at 2000×*g* and 4 °C for 10 min. Then, to pellet down large extracellular vesicles, the supernatant was centrifuged at 10,000×*g* and 4 °C for 30 min (himac CF16RN, T9A31 angle rotor, 50 mL Falcon tube). The supernatant was concentrated by ultrafiltration using a Centricon Plus 100 K (Millipore). Subsequently, 800 µL of the collected small EVs (sEVs) were incubated with HaloTag TMR ligand (Promega) (final concentration = 50 nM), HaloTag SF650T ligand (Goryo Chemical) (final concentration = 100 nM), or HaloTag SF650B ligand (final concentration = 100 nM) (Goryo Chemical) in PBS for 1 h at 37 °C. The membrane permeable HaloTag TMR and SF650T ligands were used for dual-color imaging with mGFP and mEos4b, respectively. The HaloTag SF650B ligand was employed for dSTORM imaging to estimate the diameters of sEV subtypes. In the pellet-down method, sEVs were ultracentrifuged at 200,000×*g* and 4 °C for 4 h (with a himac CS100FNX, an S55A2 angle rotor, and S308892A microtubes for ultracentrifugation). The sEV pellets were resuspended in 200 µl of Hank's balanced salt solution (HBSS; Nissui) and observed through a microscope ($1–5 × 10^9$ particles/mL). For western blotting, the pellet was resuspended in RIPA buffer containing Protease Inhibitor Cocktail Set III (Millipore). Alternatively, sEVs were also isolated from the cell culture supernatant using Tim-4 affinity beads which bind to PS exposed on sEV surface according to the manufacturer's protocol (Fujifilm Wako Chemical Co.). This method enables the purification of sEVs with minimal contaminants

and without size fractionation. Briefly, pretreatment before ultra-filtration was performed in the same manner as with the ultra-centrifugation method. Subsequently, Tim-4-bound magnetic beads were added to the supernatant, and the mixture was incubated for 1 h. The sEVs were washed, released by elution with a chelating solution, and diluted with PBS. To isolate exomeres, a small amount of sEVs remaining in the supernatant obtained by ultracentrifugation was removed by the Tim-4-affinity method. All sEV samples were stored at 4 °C.

### Transmission electron microscopy of sEVs after negative staining

The copper grid with the carbon-coated acetylcellulose film (EM Japan) was washed by a brief (~1 min) application of 10 μL of Milli-Q water, followed by blotting to remove excess liquid. The sEV sample (5 μL) was pipetted onto the grid and incubated for 1 min. Excess liquid was removed by blotting. Five μL of 2% phosphotungstic acid (TAAB Laboratory and Microscopy) was briefly (~45 s) placed on the grid, followed by blotting to remove excess liquid. The grid was dried in a desiccator containing silica gel desiccant at room temperature for 3 days and used for TEM observation. Images were acquired using a single TEM instrument (JEM-2100F, JEOL) at 200 kV. Images resulted from an average of 3-s acquisitions on a side-mounted CCD camera (Gatan) and image solution software (Gatan Digital Micrograph).

### Determination of the diameter of sEVs by qNano

The sEVs isolated, as described above, were dissolved in PBS. The diameter of the sEVs was measured by a tunable resistive pulse sensing instrument, qNano (Izon Science), according to the manufacturer's protocol. The data were analyzed by Izon Control Suite 3.3 (Izon Science). Measurements were performed with the following settings: an NP100 pore (particle detection range: 40–320 nm) with a stretch of 47 mm, a voltage of 1.2 V, and a pressure of 0.8 kPa. The samples were calibrated with 95 nm polystyrene calibration beads (Izon Science, CPC100).

### Western blot analysis

The levels of endogenous and Halo7- or mGFP-tagged tetraspanin marker proteins (CD9, CD63, and CD81) in sEVs from PC-3 or 4175-LuT cells, phosphorylation levels of FAK, and expression level of integrin β1 in the PZ-HPV-7 cells were determined by western blotting. sEVs were suspended in RIPA buffer containing protease inhibitor cocktail set III (Millipore). The protein concentrations of the samples were determined by a BCA protein assay kit (Thermo Fisher Scientific) and adjusted to 1 mg/mL for SDS–PAGE. Then, 5x concentrated Laemmle's SDS sample buffer was added to the sample. After the mixture was incubated for 10 min at 70 °C in a block incubator, 20 μL of the mixture was loaded in the lanes of a precast 4–12% gradient polyacrylamide gel (Bolt Bis-Tris Plus Gels, Thermo Fisher Scientific). Molecular weights were determined using Precision Plus Protein Prestained Standards (Bio-Rad Laboratories). After electrophoresis, the proteins were transferred to a 0.45-μm polyvinylidene difluoride membrane (Millipore). The membranes were incubated in Blocking One solution (Nacalai Tesque) for 10 min at 4 °C and then incubated with antibodies against CD63 (1:1000, SHI-EXO-M02, Cosmo Bio), CD81 (1:1000, sc-166029, Santa Cruz), CD9 (1:1000, ab263019, Abcam), GFP (1:000, SAB4301138, Sigma-Aldrich), Halo (1:500, G9281, Promega), or β-actin (1:5000, MA1-140, Thermo Fisher Scientific) in Blocking One solution overnight. After washing with Tris-buffered saline (20 mM Tris and 150 mM NaCl, pH = 7.4) containing 0.1% Tween 20 (TBS-T), the membranes were incubated in Blocking One solution containing horse-radish peroxidase (HRP)-conjugated goat anti-mouse IgG (1:5000, 12-349, Millipore), HRP-conjugated goat anti-rabbit IgG (1:10,000, A0545, Sigma-Aldrich) or HRP-conjugated donkey anti-rabbit IgG (1:4000, NA934, Cytiva) for 1 h at room temperature. After washing with TBS-T,

the membranes were treated with ECL prime reagent (GE Healthcare) following the manufacturer's recommendations. The chemilumines-cent images of the membranes were obtained by FUSION-SOLO.7S (Vilber-Loumat) and analyzed by ImageJ. To estimate the phosphor-ylation levels of FAK, PZ-HPV-7 cells ($1.0 \times 10^6$ cells) were collected at 1, 5, and 10 min after the addition of PC-3 cell-derived sEVs ($1.0 \times 10^9$ particles/mL) and suspended in RIPA buffer containing a Protease Inhibitor Cocktail Set III (Millipore) and phosphatase inhibitor Phos-STOP (Sigma-Aldrich). Western blot analysis was performed using primary antibodies against FAK (1:1000, ab40794, Abcam), pY861FAK (1:1000, 44-626 G, Invitrogen), and actin (1:5000, MA1-140, Thermo Fisher Scientific). For integrin β1 expression analysis, anti-CD29 (1:500, 610467, BD Biosciences) was used as the primary antibody.

### Immunoprecipitation

The initial step involved treating 0.6 mg of protein G-conjugated Dynabeads (Invitrogen) with 200 μL of 15 μg/mL rabbit anti-GFP anti-body (SAB4301138, Sigma-Aldrich) in 0.02% Tween 20-containing PBS. The mixture was incubated with rotation for 10 min at room tem-perature, and the supernatant was then removed. The beads were washed once with 0.02% Tween 20-containing PBS and twice with 20 mM sodium phosphate (pH 7.0) and 150 mM NaCl using a magnetic rack. Subsequently, the supernatant was removed, and the beads were washed once with 0.02% Tween 20-containing PBS and twice with PBS using a magnetic rack. The antibody-conjugated beads were then mixed with 300 μL of lysate of cells expressing CD9 or CD81-mGFP (protein concentration 0.1 mg/mL) in PBS. After incubating the mix-ture by rotation overnight at 4 °C, the supernatant was removed, and the beads were washed three times with a lysis buffer. Finally, the beads were resuspended in Laemmle's SDS sample buffer and incu-bated at 70 °C for 10 min. The supernatant was collected using a magnetic rack and analyzed by western blotting

### Immunofluorescence microscopy

Immunofluorescence staining of the active form of integrin and talin-1 was performed as follows: cells were incubated with sEV containing CD63-Halo7-SF650T ($5 \times 10^9$ particles/mL) for 1 h at 37 °C, fixed with 4% paraformaldehyde for 90 min at room temperature, and permeabi-lized with 0.01% Triton X-100 (MP Biochemicals) in PBS for 5 min. After blocking with 1% BSA solution for 30 min, the cells were treated with 4 μg/mL mouse anti-active integrin β1 antibody (HUTS-4, Millipore) or 4 μg/mL mouse anti-talin 1 antibody (GeneTEX 97H6) for 2 h at room temperature. Then, the cells were washed three times with PBS and stained with 2 μg/mL Alexa488-labeled secondary antibody for mouse IgG (ab150077, Abcam) for 30 min at room temperature. CD63-Halo7-SF650T in sEV particles were observed by TIRFM with single-molecule sensitivity, whereas the active forms of integrin β1 and talin-1 labeled with Alexa488- conjugated secondary antibody were visualized by TIRFM at much lower sensitivity (laser intensity of 0.3 μW/μm²) to perform the ensemble-averaged imaging.

Immunofluorescence colocalization of large GPI clusters, formed by the successive addition of primary and secondary antibodies, with the immunofluorescent spots of caveolae was performed as follows: PZ-HPV-7 cells expressing Halo-GPI were washed twice with HBSS and incubated with 10 μg/mL rabbit polyclonal anti-Halo antibody (G928A, Promega) in the same solution for 10 min at 37 °C. After washing with HBSS, a 10 μg/mL solution of rhodamine-labeled secondary antibody for rabbit IgG (55666, Cappel) was added and incubated with the cells for 10 min at 37 °C. The cells were then fixed with 4% paraformalde-hyde for 90 min at room temperature, permeabilized with 0.01% Tri-ton X-100 in PBS for 5 min, and blocked with 1% BSA for 30 min. After washing with PBS, the cells were incubated with 0.5 μg/mL mouse caveolin-1 antibody (Clone 2297, BD Biosciences) and 2 μg/mL Alexa488 labeled secondary antibody for mouse IgG (ab150077, Abcam).

Immunofluorescence staining of galectin-3 on live cells was performed as follows: Cells were incubated with sEV containing CD63-Halo7-TMR ($5 \times 10^9$ particles/mL) for 60 min at 37 °C. Cells were then washed twice with HBSS and incubated with 5 µg/mL anti-galectin-3 monoclonal antibody (eBioM3/38, eBioscience) in the same solution for 60 min at 37 °C. After washing with HBSS, a 2 µg/mL solution of Alexa Fluor 488-labeled secondary antibody against rat IgG (A21208, Thermo Fisher Scientific) was added and incubated with the cells for 60 min at 37 °C. After washing again with HBSS, fluorescent images were acquired by TIRFM using a custom-built, objective-lens-type microscope (based on an Olympus IX-83) equipped with a high-speed, gated image intensifier (C9016-02MLG; Hamamatsu Photonics) coupled to an sCMOS camera (ORCA-Flash4.0 V2; Hamamatsu Photonics).

## Cell treatments with various reagents

Src family kinase (SFK) activity was inhibited by treating the cells with 10 µM PP2 (529573, Tokyo Kasei Kogyo) for 10 min at 37 °C, and its analog, PP3 (5334-30-5, Tokyo Kasei Kogyo), was used as a control[93]. The IP$_3$R antagonist 2-APB (Wako, Japan)[94] was incubated at 10 µM for 10 min at 37 °C to inhibit store-operated release. PLCγ activity was inhibited by treatment with 6 µM U73122 (Cayman Chemical) for 15 min, and the analog U73343 (Cayman Chemical) was used as a control[95]. The GTPase activity of calcineurin was inhibited by treating the cells with 10 µM FK506 (06306071, Wako)[72]. Dynamin activity was inhibited by incubating cells with 70 µM Dynasore (Cell Signaling Technology, 46240) for 30 min before and during observation[96]. A membrane-impermeable galectin-3 inhibitor, TD-139 (Selleck), was administrated to cells at 10 µM for 30 min before and throughout the observation[97].

## Determination of the number of sEVs bound to cell plasma membranes

PC-3 and 4175-LuT cell-derived sEVs containing CD63-Halo7 labeled with SF650T (sEV-CD63-Halo7-SF650T) were incubated with PZ-HPV-7 and WI-38 cells cultured on a glass-based dish at 37 °C. The concentration of the fluorescently labeled sEVs was adjusted according to methods described in our recent report[46]. Briefly, glass windows of glass-based dishes were coated with 100 µL of 10 µg/mL anti-CD63 IgG antibody (8A12; Cosmo bio) in HBSS and then incubated for 2 h at 37 °C. Then, the antibody solution was removed, and the glass window was coated with 100 µL of 50 µg/mL casein (Sigma-Aldrich) in HBSS by incubation for 1 h at 37 °C to prevent nonspecific binding of the sEVs to the glass surface. One hundred µL of sEV-CD63-Halo7-SF650T was placed in a glass window coated with an anti-CD63 IgG antibody and incubated for 1 h at 37 °C. After the solution was removed and washed twice with HBSS, the single fluorescent particles of sEV-CD63-Halo7-SF650T were observed with single-molecule detection sensitivity by TIRFM using a custom-built objective-lens-type microscope (based on an Olympus IX-83) equipped with a high-speed gated image intensifier (C9016-02MLG; Hamamatsu Photonics) coupled to an sCMOS camera (ORCA-Flash4.0 V2; Hamamatsu Photonics), and the number of sEVs on glass was measured. By observing the sEV particles at three different concentrations, we obtained a calibration curve, which enabled us to prepare identical concentrations of sEV solutions. The ExoSparkler Exosome Membrane Labeling Kit-Deep Red (Dojindo) was also used as a fluorescent probe for sEVs derived from PC-3 cells, 4175-LuT cells, and MSCs. sEVs were successfully labeled with Mem Dye-Deep Red following the Exosparkler Technical Manual. Subsequently, the fluorescently labeled sEVs were purified by the Tim-4 affinity method (Fujifilm Wako Chemical Co.).

## mGPI-GPI diffusion assay

PC-3 cells ($1 \times 10^6$ cells) were transfected with 2 µg of mGFP-GPI cDNA using a 4D-nucleofector (LONZA) and cultured in two 150-mm dishes until reaching approximately 100% confluence ($2 \times 10^7$ cells). Then, sEVs

were purified by ultracentrifugation, following the method described in "Isolation of sEVs from the cell culture supernatant". The isolated sEVs were resuspended in HBSS (approximately $1 \times 10^9$ particles/mL), and $1 \times 10^8$ particles were added to PZ-HPV-7 cells on glass-bottom dishes. TIRF microscopy with single-molecule sensitivity was performed on the basal membrane of PZ-HPV-7 cells for 30 s each at 5, 60, and 120 min after sEV addition. The centroid positions of the observed fluorescent spots were tracked for 30 s, and their trajectories were plotted.

## Quantification of the number of fluorescent molecules in a single sEV

The mean (±SE) number of tetraspanins (CD9, CD63, and CD81) tagged with Halo7-TMR in single sEVs was determined from the distributions of signal intensities for individual fluorescent spots, which were observed by TIRFM using an in-house-built Olympus IX-83-based inverted microscope (100x 1.49 NA oil objective) equipped with two high-speed gated image intensifiers (C9016-02MLG; Hamamatsu Photonics) coupled to two sCMOS cameras (ORCA-Flash4.0 V2; Hamamatsu Photonics)[49,53,98,99]. The photon count of each TMR was approximately $164.2 \pm 4.4$, following our previously established method[100]. The distributions of the fluorescence intensities of individual spots of TMR attached to the glass surface were fitted with a single log-normal function[52,101], representing TMR monomers. We have previously confirmed the linearity between the number of molecules in clusters and fluorescence intensity in our observation system[52,99,100,102]. The distributions of signal intensities of tetraspanins tagged with Halo7-TMR in individual sEV particles were subsequently fitted with the sum of the log-normal function for monomers, dimers, and greater oligomers, based on the monomer's log-normal function (10 mer was assumed to constitute the upper limit in this study)[13]. The percentages of each component were calculated and averaged. For the binding assay of TAMRA-conjugated amphipathic helix peptides (ApoC-TAMRA and α-synuclein(p2-23)-TAMRA), the fluorescence intensities of TAMRA on individual sEV particles containing CD9, CD63, and CD81-Halo7-SF650T were quantitatively analyzed via TIRFM using a Nikon Ti inverted microscope (100x 1.49 NA oil objective) equipped with two qCMOS cameras (ORCA Quest C15550-20UP; Hamamatsu Photonics).

## Colocalization analysis of marker proteins in single sEVs

To examine whether tetraspanin marker proteins (CD9, CD63, and CD81) tagged with mGFP or Halo7-TMR are colocalized in the same sEV particle, we performed simultaneous dual-color observation of single sEV particles bound to the glass surface; these observations were performed by TIRFM with single-molecule detection sensitivity with a custom-built Olympus IX-83-based inverted microscope (100x 1.49 NA oil objective) equipped with two high-speed gated image intensifiers (C9016-02MLG; Hamamatsu Photonics) coupled to two sCMOS cameras (ORCA-Flash4.0 V2; Hamamatsu Photonics), as described previously[49,53,98,99]. mGFP was excited using a 488 nm laser (Excelsior-488C-50-CRW, 50 mW, Spectra-Physics) at 1.9 µW/µm², and TMR was excited using 561 nm laser (Excelsior-561-100-CDRW, 100 mW, Spectra-Physics) at 2.8 µW/µm². The obtained sEV images were binarized by the "Li methods" function of ImageJ[103] for each channel, and the number of sEV particles in the green channel (G) of mGFP, red channel (R) of Halo7-TMR, and merge of the two channels (M) (only colocalized spots exist in M) was counted by using the "Analyze Particles" function of ImageJ. The ratios of M to R (M/R) were quantitatively analyzed to eliminate effects caused by differences in the labeling efficiency of Halo7 with TMR.

## Simultaneous dual-color observation of PALM movies of membrane invaginations and single-fluorescent particles of sEVs on living cell plasma membranes

PZ-HPV-7 or WI-38 cells expressing AP2α-mEos4b, CAV1-mEos4b, or LAMP-2C-mEos4b were cultured on glass-based dishes for 2 days.

Because the intensity of single-fluorescent spots of mEos4b is high[104] and mEos4b does not form dimers[105], mEos4b was used to acquire PALM images. After washing with HBSS three times, sEVs containing CD9, CD63, or CD81-Halo7-SF650T were incubated with the cells for 60 min at 37 °C. For the reference experiment using SF650T-conjugated transferrin, we proceeded as follows: Six µL of 500 µM SaraFluor650T (SF650T; Goryo Chemical) NHS ester was incubated with 100 µL of 1.0 µM human transferrin (HZ-1317; Protein tech) in 0.1 M NaHCO$_3$ for 60 min at room temperature. SF650T-conjugated human transferrin was subsequently isolated using a NAP-5 column (Cytiva). The dye-to-protein ratio was 2.2. Single-fluorescent particles of the sEVs and single molecules of mEos4b on the basal surface of the cells at 512 × 512 pixels (25.6 µm × 25.6 µm) were observed at 5 ms resolution (200 frames/s) for 5008 frames by TIRF illumination using a custom-built Olympus IX-83-based inverted microscope (100x 1.49 NA oil objective) equipped with two high-speed gated image intensifiers (C9016-02MLG; Hamamatsu Photonics) coupled to two sCMOS cameras (ORCA-Flash4.0 V2 Hamamatsu Photonics), as described previously[14,49,53,98,99]. The final magnification was 133×, yielding pixel sizes of 47.1 nm (square pixels). Because approximately 50–100 particles of sEVs bound to the cell basal PM could be observed, the gap between the glass and the cell basal PM is broad enough for sEVs of 70 nm in diameter (Fig. 2b) to transverse. The superimposition of images in different colors obtained by two separate cameras was performed as reported previously[48,50,52,53,99,100]. mEos4b was activated and excited using a 405 nm laser (Excelsior-405C-50-CDRH, 50 mW, Spectra-Physics) and a 561 nm laser (Excelsior-561-100-CDRW, 100 mW, Spectra-Physics) at 20 nW/µm$^2$ and 14 µW/µm$^2$, respectively, and SF650T was excited using a 647 nm laser (140 mW, Omicron Laserage Laserprodukte) at 3.2 µW/µm$^2$. The average photon count of mEos4b was approximately 100 ± 9.3 per frame, according to our previously established method[100]. The single-molecule localization precision of mEos4b was 20 ± 1 nm. The lifetimes, bleaching, blinking, and characteristics of mEos4b have been detailed previously[14,49,100]. Single-molecule tracking of mEos4b and single-particle tracking of sEVs were performed by in-house computer software as previously reported[49,53,98–100,106].

The reconstructions of PALM images with a pixel size of 10 nm were performed using the ThunderSTORM plugin[107] for ImageJ installed in the Fiji package with "wavelet filtering" (B-spline order = 6 and B-spline scale = 6.0) and "least squares" (fitting radius = 3 pixels, initial sigma = 1.6 pixels). After spot detection, the postprocessing steps "Remove duplications" (distance threshold = uncertainty) and "Drift correction" (cross-correlation with 5 bins) were further performed. Subsequently, ThunderSTORM was used to output the frame, x and y coordinates of each spot as csv data, and these csv data were imported to a custom MATLAB program to create a super-resolution movie. In addition, we use the uncertainty, which represents the spatial precision of the spot. For each spot, a Gaussian distribution with the x and y coordinates as the center and a standard deviation (SD) equal to A ( = 6) times the uncertainty of the spot is generated as an existence probability distribution of the spot. The existence probability distribution for all structures at time $t$ is obtained as a sum of all distributions for spots that appeared in [6$t$ + 1  6$t$ + 1002] frames. The PALM super-resolution movie data are generated by repeating this process for $t$ = 0, 1, 2, ... (Code is available at https://github.com/kgnsuzukilab). To match the temporal resolution of a single-particle movie with that of a PALM movie and to preserve synchronicity, six consecutive frames in the single-particle movie were averaged, and the temporal resolution was converted to 33.3 frames/s. Subsequently, PALM images and six-frame averaged single-particle images were superimposed (Fig. 4b). Parameter-free image resolution estimation based on decorrelation analysis was performed using the Fiji plugin "Image decorrelation analysis"[35]. This software employs a Fourier-ring-correlation (FRC) method based on the calculation of the cutoff frequency of the correlation factor obtained for the original image and low-pass filtered images. This approach is used to estimate the resolution based on an individual image without further requirements or a priori knowledge[35].

For long-term simultaneous observation (Fig. 4g–j), single fluorescent particles of the sEVs and single molecules of caveolin-1 or LAMP-2C-mEos4b were imaged at 33 ms resolution (video rate) for 10,000 frames (~5.5 min) using TIRFM. To create one PALM image, the existence probability distribution of spots was integrated for 1000 frames. Thirty consecutive frames from the single-particle movie were averaged, converting the temporal resolution to 1 frame/s. The PALM images and 30-frame averaged single-particle images were then superimposed (Fig. 4g, h). The fluorescence intensity in the merged images was analyzed using ImageJ.

## Confocal microscopy

Confocal laser scanning fluorescence microscopy (CLSM) was performed at 37 °C with an FV1000-D microscope (Olympus IX81) equipped with an Ar laser (488 nm) and LED laser (559 nm). A 60× (numerical aperture (NA) = 1.46) oil objective was used to obtain images (typically, 1024 × 1024 pixels). The images were quantified and analyzed by the acquisition software FV10-ASW4.2 and ImageJ.

## Intracellular Ca$^{2+}$ imaging

Intracellular Ca$^{2+}$ mobilization in living recipient cells was monitored by confocal microscopy using Fluo8-H AM (AAT Bioquest) as a probe. The recipient cells in a glass-based dish were incubated with 5 µM Fluo-8H AM in HBSS (100 µL) in the dark for 30 min at 37 °C, after which the cells were washed three times with HBSS. Two hundred µL of sEV solution in HBSS was added to the cells. Images were acquired every 20 s using CLSM, and observations were made for 10–20 cells in each case. To measure the saturation level of the fluorescence signal intensity at high Ca$^{2+}$ concentrations, 1 µM (final concentration) ionomycin (Fujifilm Wako Chemical Co.) was added (to increase the intracellular Ca$^{2+}$ concentration to 1.3 mM of the extracellular Ca$^{2+}$ concentration). The image sequences were analyzed using ImageJ software; time-dependent changes in the Fluo-8H signal intensity in the region of interest (ROI) (20-pixel diameter circle) were examined. Intracellular Ca$^{2+}$ concentration induced by the addition of sEVs was normalized by that induced by 1 µM ionomycin (Figs. 7c–e, 8d).

For simultaneous observation of Ca$^{2+}$ and single molecules of PLCγ2 at high temporal resolution, Fluo-8H and single molecules of SF650T-Halo7-PLCγ2 were imaged using a setup similar to single-molecule microscopy. Fluo-8H was continuously observed via oblique-angle illumination with a 488 nm laser (0.3 µW/µm$^2$), while single molecules of SF650T-Halo7-PLCγ2 were observed for 21 frames (7 s) at 39 frames (13 s) intervals using TIRFM with a 647 nm laser (3.0 µW/µm$^2$) to prevent SF650T photobleaching. Images were acquired at a rate of 3 Hz.

For the observation of Ca$^{2+}$ levels after the addition of serum-free cell culture supernatant to recipient cells, Fluo-8H in the recipient cells was monitored using the same microscopy setup as described earlier. PC-3 and PZ-HPV-7 cells were cultured in 150-mm dishes and grown to approximately 100% confluence. Thereafter, the cell culture medium was replaced with 30 mL of FBS-free DMEM. After 48 h of incubation, the cell culture supernatant was collected and centrifuged at 300×$g$ for 10 min to remove cells. Subsequently, 500 µL of the supernatant was added to PC-3 or PZ-HPV-7 cells loaded with Fluo-8H in glass-base dishes. The fluorescence of Fluo-8H in the cells was observed by confocal microscopy.

## Single-particle tracking and 3D super-resolution microscopy

To determine the 3D positions of sEVs containing tetraspanins labeled with SF650T via Halo7 and Cav1-mEos4b or Rab11-mEos4b in the cell PM or cytoplasm, single molecules/particles of these fluorescent

probes in cells were observed by oblique-angle illumination (647 nm, 3.0 μW/μm² for observation at 33 ms/frame) using a Nikon Ti inverted microscope (100x 1.49 NA oil objective) equipped with two qCMOS cameras (ORCA Quest C15550-20UP; Hamamatsu Photonics). A cylindrical lens (MED54301, Nikon) was set in each imaging path to create astigmatism in two directions, allowing the x and y coordinates of the probe to be determined from the center of the image, as well as the z-coordinate from the ellipticity[108]. To calibrate the Z position, we observed single particles of multicolor fluorescent beads (TetraSpeck Microspheres 100 nm in diameter; Thermo Fisher Scientific) fixed on a glass-based dish and continuously moved (1 nm/ms) using a Z-piezo stage (Nano Z100, Mad City Labs) at 5 ms/frame. The obtained movies were analyzed by the "Cylindrical lens calibration" function of the ThunderSTORM plugin. The obtained xyz coordinates were plotted with MATLAB software. The typical localization precision values of the sEVs are shown in Supplementary Fig. 7a. To quantitatively analyze the degree of colocalization between sEVs and Rab11, we first measured the center coordinates of the sEVs using the "Find Maxima" function of ImageJ. Next, we quantified the number density of Rab11-mEos4b within a 100 nm radius around the center coordinates of the sEVs by ImageJ. To assess the degree of colocalization, we counted the number of sEVs that colocalized with Rab11-mEos4b with a number density more than two times higher than the randomized background signal. The number of Rab11-mEos4b molecules in a recycling endosome was $17.6 \pm 3.7$ (mean $\pm$ SE), which was estimated by dividing the number of localizations of Rab11-mEos4b in a recycling endosome for 5002 frames by the number of localizations per mEos4b molecule, as previously reported[14]. According to the Poisson distribution, all recycling endosomes contained more than one Rab11-mEos4b molecule and were detectable by PALM.

## Colocalization analysis between single molecules on the recipient cell PM and single sEV particles

To quantitatively analyze the colocalization of single molecules in the recipient PZ-HPV-7 cell PMs and single sEV particles, they were tracked with in-house computer software as previously reported[49,53,98–100,106]. The precision of the position determinations for single stationary fluorescent probes in the observation at 5 ms/frame was estimated using the standard deviations obtained for the determined coordinates of the probes fixed on coverslips. For mGFP, TMR, and SF650T, the localization precisions of the single molecules were $26 \pm 1.0$, $24 \pm 0.5$, and $26 \pm 2.4$ nm, respectively. We subsequently performed a pair cross-correlation analysis as follows. (1) An ROI was selected. (2) The distances between all pairs of green and red spots in the ROI in a video frame were measured, and this process was repeated for all the video frames in an image sequence (Supplementary Fig. 10a). (3) The number densities of pairwise distances R(r) were plotted against the distance (Supplementary Fig. 10b, top). (4) The cross-correlation function, C(r), was estimated by dividing R(r) by the number density of pairwise distances N(r) in the control image, which was obtained by superimposing the magenta image with the 180-degree rotated green image (Supplementary Fig. 10b, bottom). The method yields robust, quantitative results by employing a model-independent approach that obviates the need for assumptions about molecular behavior or distribution patterns[41,42].

## Colocalization analysis between membrane invaginations in PALM movies and single sEV particles

The distances between the centroid of the sEVs and the contour of the membrane invaginations were quantified using single-particle tracking data of the sEVs and binarized images of the membrane invaginations in the PALM movies. To create binarized images of the membrane invaginations in the PALM movies, we used the kernel density estimation (KDE) method. KDE is a traditional image segmentation method that is used to rapidly determine the criterion for segmenting

PALM images, as reported previously[49,109] and described below. Furthermore, a recent study demonstrated that KDE is among the most appropriate methods for segmenting images of clearly outlined structures, such as protein clusters[40]. KDE provides a way to interpolate object boundaries without bias by randomly fluctuating activated boundary fluorophores.

ThunderSTORM datasets were imported into in-house KDE software (MATLAB) for image segmentation and colocalization analysis. We used localization coordinates as well as uncertainty, which reflects the spatial detection precision of the spots. Let $x_i, y_i$ and $u_i, i = 1, 2, \cdots, N$ be the horizontal and vertical localization coordinates and the uncertainty of spot $i$, respectively. $N$ represents the total number of spots. Considering the uncertainty of the location measurement of spot $i$, the existence probability $p_i(x, y)$ of the spot $i$ spreads to the Gaussian distribution with the center coordinates and the standard deviation (SD) that is proportional to the uncertainty $u_i$ as follows:

$$p_i(x,y) = \frac{1}{2\pi(Au_i)^2} e^{-\frac{(x-x_i)^2 + (y-y_i)^2}{2(Au_i)^2}}$$

in which $A$ is the proportional coefficient of the SD to the uncertainty $u_i$; this coefficient was estimated as $A \approx 6$ in some preliminary experiments. The Gaussian distribution for constructing each existence probability $p_i(x, y)$ is called the Gaussian kernel. The existence probability distribution for all spots, i.e., the PALM image, is obtained by the average of the existence probability distribution $p_i(x, y)$ as follows:

$$p(x,y) = \alpha \frac{1}{N} \sum_{i=1}^{N} p_i(x,y) + (1-\alpha) p_{bg}(x,y)$$

excluding spots with physically implausible small or large uncertainties $u_i$ for single fluorophores because these spots are likely to be noise (e.g., electrical shot noise from a camera). $p_{bg}(x, y)$ represents the background noise generated by the probability density distribution $f(v) = \frac{1}{2\pi\sigma_{bg}^2} e^{-\frac{(v-\mu_{bg})^2}{2\sigma_{bg}^2}}$, independent of $x, y$. $\alpha$ is a weight determined by the respective occurrence probabilities of the spots and background noise.

The structures can be considered to exist when the PALM image, which represents the probability of a structure existing, is above a certain threshold value $\theta$ because the spots appear uniformly inside the structures with a certain probability. The optimal threshold value $\theta$ must be determined for each obtained PALM image because the optimal value differs depending on the target molecule and the experimental environment. In general, the histogram of $p(x, y)$ values for all $x, y$ forms a bimodal shape that contains larger values, which are created by dense spots in structures, and smaller values created by sporadic noise. Therefore, the threshold value $\theta$ should be determined to separate these two structures. Otsu's method is well known as a method for determining these threshold values[110]. Let $S_{in}$ and $S_{out}$ be sets of coordinates $(x, y)$ where $p(x, y) \geq \theta$ and $p(x, y) < \theta$, respectively. The sets $S_{in}$ and $S_{out}$ indicate the inside and outside of the clusters, respectively. The numbers of elements in sets $S_{in}$ and $S_{out}$ are represented by $N_{in}$ and $N_{out}$, respectively. The intraclass variances of the sets $S_{in}$ and $S_{out}$ are calculated as follows:

$$\sigma^2[S_{in}] = \frac{1}{N_{in}} \sum_{(x,y)\in S_{in}} (p(x,y) - \mu[S_{in}])^2,$$

$$\sigma^2\left[S_{out}\right] = \frac{1}{N_{out}} \sum_{(x,y)\in S_{out}} \left(p(x,y) - \mu[S_{out}]\right)^2.$$

In the above equations, $\mu\left[S_{in}\right]$ and $\mu\left[S_{out}\right]$ are the respective averages of the set $S_{in}$ and $S_{out}$ obtained by

$$\mu\left[S_{in}\right] = \frac{1}{N_{in}} \sum_{(x,y)\in S_{in}} p(x,y),$$

$$\mu\left[S_{out}\right] = \frac{1}{N_{in}} \sum_{(x,y)\in S_{out}} p(x,y).$$

The average of these intraclass variances is:

$$\sigma^2\left[S_{in}, S_{out}\right] = \frac{N_{in}\sigma^2\left[S_{in}\right] + N_{out}\sigma^2\left[S_{out}\right]}{N_{in} + N_{out}}.$$

Otsu's method determines the optimal threshold value $\hat{\theta}$ to minimize the average of the intraclass variances $\sigma^2\left[S_{in}, S_{out}\right]$ as follows:

$$\hat{\theta} = \arg\min_{\theta} \sigma^2\left[S_{in}, S_{out}\right].$$

In this study, the structures are determined using Otsu's method. Thereafter, the distance from the contour of the determined structures to the center of the sEVs on the PM was measured, and the number densities of pairwise distances at a given distance were plotted. If an sEV is contained in a structure, the distance is displayed as a negative value. Furthermore, we also measured the distances between random coordinates and the contours of the structures in silico. The corresponding distributions of relative frequency were calculated using a bin width of 50 nm. For normalization, the relative frequency of sEVs within each bin was divided by the corresponding relative frequency of the computer-generated random spots. A ratio exceeding 1 indicates a greater possibility of sEVs at the measured distance. The method yields robust, quantitative results by employing a model-independent approach that obviates the need for assumptions about molecular behavior or distribution patterns[43,44]. The code is available at https://github.com/kgnsuzukilab.

## Validation of molecule location by TIRFM and oblique-angle illumination

To examine the location of LAMP-2C molecules, markers for lysosomes, Golgi, and ER were observed by both TIRFM and oblique-angle illumination[14]. For LAMP-2C, Golgi, and ER observations, cells were transfected with 1 µg of LAMP-2C-mEos4b, mEos4b-Rab6a, and mEos4b-STING, respectively, using a 4D-nucleofector (LONZA). The transfected cells were seeded onto glass-bottom dishes and cultured for 36 h before microscopy. LAMP-2C-mEos4b, mEos4b-Rab6a, and mEos4b-STING in cells were imaged by TIRFM or oblique-angle illumination using 405 nm (20 nW/µm²) and 561 nm (14 µW/µm²) lasers. For lysosome observations, cells were incubated with a 100 nM SiR-Lysosome (Spirochrome) reagent[111] for 1 h at 37 °C, and then washed three times with HBSS. These cells were then observed by TIRFM or oblique-angle illumination using a 647 nm laser (3.2 µW/µm²).

## Estimation of sEV diameter by dSTORM

To estimate the sEV diameter, we acquired dSTORM images of sEVs that contained tetraspanins (CD9, CD63, and CD81) tagged with Halo7 SF650B attached to the glass. These images were attained by observing single molecules of SF650B for 10,000 frames by TIRF illumination (647 nm, 16 µW/µm² for observation at 4 ms/frame) using a Nikon Ti inverted microscope (100x 1.49 NA oil objective) equipped with a qCMOS camera (ORCA Quest C15550-20UP; Hamamatsu Photonics).

dSTORM images with a pixel size of 10 nm were reconstructed using the ThunderSTORM plugin in ImageJ with "wavelet filtering" (B-spline order = 6 and B-spline scale = 6.0) and "least squares" (fitting radius = 3 pixels, initial sigma = 1.6 pixels). The obtained x and y coordinates for each spot in the csv data were subsequently imported into SR-Tesseler software[15] for cluster analysis based on Voronoi polygon density methods. To generate Voronoi polygons without correcting for blinking and multi-ON frame detection, the "Detection cleaner" function was used (within the entire ROI). The sEVs were identified using the "Objects" function with a Voronoi polygon density factor of 5. To ensure that the estimated sEV diameters were accurate, only sEVs with a minimum of 100 localizations were analyzed (Supplementary Fig. 4). Finally, the SR-Tesseler output data were plotted and subjected to statistical analysis using ORIGIN 2018b software.

## Preparation of fluorescently labeled small unilamellar vesicles (SUVs)

As a size reference for sEVs, 100-nm-diameter fluorescently labeled SUVs were prepared according to the following protocol: 100 µL of 10 mM dimyristoylphosphatidylcholine (DMPC, Avanti Polar Lipids, Inc.) and 100 µL of 1 µM SF650B-GM3 synthesized by us (see "Synthesis of SaraFluor650B-GM3") were mixed in a chloroform solution in a brown vial and dried under nitrogen flow for 15 min, generating a thin lipid film on the bottom of the vial. The dried lipids were hydrated overnight in 1 mL of PBS (the final lipid concentration in the PBS suspension was 1 mM). The resulting multilamellar vesicles were freeze-thawed five times using liquid nitrogen and warm water at 45 °C. The vesicle suspension was extruded 10 times through a polycarbonate filter with 100 nm pores using a LiposoFast extruder (Avestin, Ottawa, Canada) immediately before use. The extruded vesicle suspension was diluted 100-fold with HBSS and attached nonspecifically to the glass surface.

## Synthesis of ApoC-TAMRA and α-synuclein(p2-23)-TAMRA

Fmoc-protected amino acids were purchased from Watanabe Chemical Industries (Hiroshima, Japan) or FUJIFILM Wako Pure Chemical Co. (Tokyo, Japan). 5-TAMRA was purchased from Chemodex (Gallen, Switzerland). Probes were dissolved in DMSO to generate a 100 µM stock solution. The stock solution was stored in a freezer (−20 °C) in the dark until further experiments. The final DMSO concentration in the ApoC-TAMRA-treated samples was less than 0.1% (v/v). ApoC-TAMRA (Supplementary Fig 5a-A) and α-synuclein(p2-23)-TAMRA (Supplementary Fig 5a-B) were synthesized using a Biotage Initiator and microwave peptide synthesizer (Biotage, Uppsala, Sweden) based on Fmoc solid-phase peptide chemistry on Rink-Amide-ChemMatrix resin (Biotage). The 2-(6-chloro-1-H-benzotriazole-1-yl)-1,1,3,3-tetra-methylaminium hexafluorophosphate (HCTU)/diisopropylethylamine (DIEA) system was employed for the coupling reaction. After all the amino acid residues were elongated, the AEEA spacer was introduced at the N-terminus, followed by coupling with TAMRA at the N-terminus. Deprotection and cleavage from the resin were conducted using trifluoroacetic acid/triisopropylsilane/water (95/2.5/2.5). The solution was dropped into cold diethyl ether to precipitate the crude peptide probe. The crude product was purified by a reverse-phase HPLC system (pump, PU-2086 Plus × 2; mixer, MX 2080-32; column oven, CO-1565; detector, UV-2070 plus and UV-1570 M (Japan Spectroscopic Co. Ltd.)) equipped with a C18 column (Inertsil ODS3; GL Sciences Inc., Tokyo, Japan) using a gradient of water/acetonitrile containing 0.1% TFA (Supplementary Fig. 5b-A and b-B). The probe was verified by MALDI-TOF-MS (Bruker Daltonics Autoflex Speed-S1): MALDI-TOF-MS (ApoC-TAMRA) 3097.28 [calcd. for $(M + H)^+$: 3097.61], (p2-23-TAMRA): 2845.89 [calcd. for $(M + H)^+$: 2845.54] (Supplementary Fig. 5c). The concentration of the probes was determined based on the molar absorption coefficient of TAMRA at 555 nm in DMSO ($\varepsilon = 90,000$ cm⁻¹M⁻¹).

## Synthesis of SaraFluor650B-GM3

Detailed information on the synthesis of SaraFluor650B-GM3 is provided in Supplementary Fig. 3. In brief, the prepared SaraFluor650B-linkerN$_3$ (S3) (Supplementary Fig. 3b) and BCN-GM3 (S5) products (Supplementary Fig. 3a) were dissolved in MeCN/H$_2$O (200/100 μL) at room temperature. After the mixture was stirred for 19 h at room temperature, the reaction was monitored by TLC (CHCl$_3$/MeOH/5% CaCl$_2$ aq. = 5:3:0.2), and the reaction mixture was concentrated. The residue was purified by PTLC using CHCl$_3$/MeOH/H$_2$O (5:3:0.2) to generate SaraFluor650B-GM3 (S6).

## Statistics and reproducibility

Statistical analysis was performed using OriginPro software, with a *p* value of less than 0.05 considered statistically significant. For datasets that followed a normal distribution, two-tailed Welch's *t*-tests were used for comparisons between two conditions. In case requiring multiple statistical tests, the significance level was corrected by the Holm–Sidak method. We performed pilot experiments using smaller sample numbers, and after performing Welch's *t*-test, we determined the expected minimal sample size to prove or disprove the hypothesis. We generally did more experiments than this minimal number of experiments. No data were excluded. All presented micrographs (Figs. 1c, e; 2a; 3e, g; 4e, f; 6a; 7a, f, i, 8a and Supplementary Figs. S1b, S2, S7b, S8, S9, S11, S12, S14a, c, S15a, S16a, S17) are representative of findings from at least three independent experiments and were consistently reproducible. This standard of reproducibility applies to all other data included in this publication.

## Reporting summary

Further information on research design is available in the Nature Portfolio Reporting Summary linked to this article.

# Data availability

The datasets generated and analyzed in the current study are available in the Supplementary Information file and source data. All other data that support the findings of this study are available from the corresponding author upon request. Source data are provided with this paper.

# Code availability

Custom-written computer codes for data collection and analysis are available at https://github.com/kgnsuzukilab.

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

## Acknowledgements

We thank Dr. Tetsuya Hirata and Prof. Yasuhiko Kizuka (iGCORE, Gifu University) for supporting the cell sorter experiment. We also thank Yumi Matsuno for supporting the preparation of extracellular vesicles from donor cells; Shinobu Kawaguchi for constructing various cDNAs; Yuji O. Kamatari for TEM observation; Takahiro Fujiwara and Akihiro Kusumi for developing the analysis software for single-molecule imaging; Joan Massagué for providing 4175-LuT cells. This work was supported in part by Japan Science and Technology Agency (JST) grants from the Core Research for Evolutional Science and Technology

(CREST) program in the field of "Extracellular Fine Particles" to K.G.N.S., K.M.H., H.A., and Y.S. (JPMJCR18H2), the CREST program in the field of "Cell Control" to K.G.N.S. (JPMJCR24B3), the JST PRESTO program in the field of "Extracellular Fine Particles" to Y.S. (JPMJPR19H4) and A.H. (JPMJPR18H9), the JST FOREST program to A.H. (JPMJFR216B) and Grants-in-Aid for Scientific Research from the Japan Society for the Promotion of Science (JSPS) (Kiban B to K.G.N.S. [21H02424, 18H02401]), a Grant-in-Aid for Challenging Research (Exploratory) from JSPS to K.G.N.S. (20K21387), a Grant for Core-to-Core program from JSPS to K.G.N.S. and H.A. (JPJSCCA202000007), a Grant-in-Aid for Innovative Areas from the Ministry of Education, Culture, Sports, Science and Technology of Japan (MEXT) to K.G.N.S. (18H04671), the National Cancer Center Research and Development Fund to K.G.N.S. (2023-A-03), the Japan Agency for Medical Research and Development (AMED) to K.G.N.S. (23tk0124003h001, 1268282), the Takeda Science Foundation to K.G.N.S. and H.A., and the Uehara Memorial Foundation to K.G.N.S.

## Author contributions

K.M.H. performed the single fluorescent-molecule tracking experiments and biochemical experiments. K.M.H., R.S.K., and K.G.N.S. developed and constructed the single-molecule imaging station. Y.Y. and K.M.H. developed the analysis software. Y.S. synthesized the TAMRA-labeled AH peptide probes. E.Y., N.K., and H.A. synthesized the SaraFluor650B-labeled GM3 probes. A.H. provided the 4175-LuT cells. K.M.H., R.S.K., Y.S., A.H., and K.G.N.S. discussed the results and experimental plans extensively during the entire course of this research. K.M.H. and K.G.N.S. conceived and formulated the project and evaluated and discussed the data. K.M.H. and K.G.N.S. wrote the manuscript, and all authors participated in revising the manuscript.

## Competing interests

The authors declare no competing interests.
