## [Transparent Peer Review file · Nature Communications]

Uptake of small extracellular vesicles by recipient cells is facilitated by paracrine adhesion signaling

Corresponding Author: Professor Kenichi Suzuki

Version 0:

Reviewer comments:

Reviewer #1

(Remarks to the Author)

In the paper entitled: "Uptake of small extracellular vesicles by recipient cells is facilitated by paracrine adhesion signaling". Dr Suzuki and colleagues characterized the EVs released by Prostate cancer cells and then studied their interaction with acceptor cells using high resolution microscopy approaches.

The authors have a very interesting observation that is the capacity of certain EVs to signal with specific acceptor cells. I would suggest developing their work around this discovery. Instead, although the authors have developed sophisticated imaging techniques the conclusion on EV endocytosis are not novel and have problematic data interpretation that rise serious concerns.

The major concern I have are the following:

1. The authors use of Lamp2 as a marker of phagocytosis. Lamp-2 is a marker of lysosomes and does not label the formation of the phagocytic cup. Indeed, as clearly stated in their reference 23 (Marion S et al., Traffic 2011), lamp2 colocalization is detected after 2 hours of administration of beads used to detect the phagosome, or in other words define the fusion of the phagosome with lysosomes (phagocytic maturation). Therefore, all these data are mis-interpreted.
2. The misinterpretation of these data rise serious concern regarding the capacity of the microscopy pipeline to be able to discern events that happen at the cell interface and events that take place within the cell. In addition, it is not clear the time that is needed to observe "true" internalizations.
3. Indeed, the authors observe events in a very short time scale. At this time scale many endocytic events cannot take place. For instance, the average life of a clathrin coated vesicle is of ~1 min (e.g. Ehrlich M et al., Cell 2004; Aguet et al., Dev Cell 2013). And similar considerations can be done for phagocytosis (e.g. Vorselen D et al, eLife 2021).
4. True internalizations should consist of: a. the particle (e.g. EV) is entrapped in a forming cargo (e.g. caveolae); b. the particle colocalize for a certain amount of time, necessary to finish the cargo or being pinched by dynamin (~20 s, see Cocucci et al., MBOC 2014) and then, c. move in the cytosol either through direct movement or randomly diffusing. The data presented here do not clearly characterize the dynamics of these events, probably for the short time of observation (see points 1 and 3).
5. The authors suggest an interesting mechanism: EVs can stimulate the cell to induce their uptake. It is also interesting that the event seems to be specific, or in other words that the signal is not autocrine. It would be useful to overexpress the integrins that are considered important for signaling in donor cells and see whether this can result in stimulation by their same EVs. In addition, it is well-known that there are two types of caveolae: short lived that do a kiss-and-run fusion and long lived (see Pelkman and Zerial Nature 2005 and Boucrot et al JCS 2011). Therefore, the short lived caveolae that appear on the plasma membrane are the first type. In the work of Pelkman it has been demonstrated that kinases can control the dynamics of caveolae. It would strength the paper if the authors could demonstrate a similar dynamic on caveolae formation rather than concentrating on dynamin, which is probably a downstream step.
6. The authors have strong claims on the dynamics of EVs generalizing their data. They should dump their conclusion because this system of signaling and internalization might be valid only in this model and not in others.
7. The fact that EVs can be internalized by clathrin endocytosis or caveolae has already been demonstrated. Therefore, I would focus on the signaling mechanism which is novel and interesting even if related only to this cellular model.

Minor comments:

1. Western blot in figure 1 is unclear. The antibody against tetraspanins should detect both the wild type and chimera protein but this is not the case. This western should be improved.
2. In figure 1 they show single molecule calibration. However, it is necessary to show the single step photobleaching events

to ensure that the single TMR are single molecules. Are the imaging parameters used to acquire the single molecules the same as the one used to acquire the EV signal? It is important to show linearity of the detectors and of the laser power to ensure a correct calibration. These data are missing in the actual manuscript but should be shown.

3. I do not understand what the authors mean with "membrane packaging". In addition an easy explanation for the higher fluorescence of certain EV in comparison to other might be just dependent on their lipid composition. I would suggest to substitute "membrane packaging" with "membrane composition". A possibility is that some EVs have different curvature, however the data presented here do not support this possibility since the EVs have all similar diameter. Conversely, the amount of PS may drive the binding of this peptide. I would also suggest citing: Pranke IM et al, JCB 2011.

4. It is widely accepted that EVs are internalized and do not fuse directly with the plasma membrane. It would be interesting to have a citation in this regard.

Reviewer #2

(Remarks to the Author)

The manuscript "Uptake of small extracellular vesicles by recipient cells is facilitated by paracrine adhesion signaling" by Hirosawa et al. tackles an interesting topic of extracellular vesicle internalization. The authors use various high-end microscopy methods and raise interesting observations. However, I feel more efforts are required to increase the robustness of the findings.

First of all, the authors make an unjust statement that small EVs equal to exosomes. Current view in the field is that exosomes derive from multivesicular bodies' intraluminal vesicles. It is important to explain what the authors think the origin of their sEVs is and why, and why would they not be derived from other cellular origins. The tetraspanin markers, that the authors use, are not unique to exosomes. Could the source of sEVs relate to the cell type selectivity the authors report?

The microscopic approaches in the manuscript is very difficult to follow. The authors have used various different methods, including single molecule localizations microscopies. It remains unclear at many instances if the data is based on true single molecule localizations, how this is verified (include raw data, photon counts, life times, accuracy, bleaching, blinking etc characteristics, localization maps), and when just bulk fluorescence is taken into account. For example Movie 1: is it single molecule (does it bleach in one frame?) or is it single vesicle with several single molecules? The details of the usage of dSTORM and PALM are not explained nor the need for or the validation of single molecule resolution. 3D-dSTORM is not a trivial method, and would need careful validation. Also, why do the authors use single molecule detection to follow protein recruitment to the plasma membrane. Typically ensemble TIRF imaging should better address this. In general, the authors should report their single molecule data in much more open and explainable manner, and also compare the data to the ensemble imaging when possible. For analysing colocalization, single molecule data is rarely a method of choice. The article also contains a medley of different fluorophores, but it is not explained why they are chosen and the details of the each imaging set up are missing. Each figure legend should clearly state how the imaging was done and what is shown.

Figure 1c: the western blot does not show any bands of the CD9 or CD81 fusion proteins at the same level as the membranes probed with the anti-tag antibodies. How is this explained? How is the 0.1-0.5 times reduced expression calculated? The quantification should be added.

Fig 3g-h shows less sEVs on the cell membrane in cells with inactive dynamin. Would the conclusion then not rather be that there is more internalization? The necessary details on how the data is obtained are missing to allow interpretation of the data.

In general the sum of EV fluorescence intensity does not appear like a right measure of EV uptake, as also the EVs trapped on cell surface and failing to internalize can give high signals. This analysis been used also in other figures. Authors should explain this and find more specific tools to examine internalized vesicles as opposed to cell-surface trapped vesicles. How long can the sEV be visualized inside the cells and can they be distinguished from the cell surface-bound ones?

Fig 4. The authors should be able to give some quantitative analysis data on the relative percentages of different internalization modes on this data. Why single molecule imaging? How does this look when imaged in ensemble? Other methods and good controls for colocalization analysis should be sought.

Fig 5. Same concerns than in Fig 4. The colocalization of sEVs with Rab11 seems very weak or even absent. Proper controls should be included here to ascertain that the colocalization is not only random. A simple way to do this would be to turn one channel 90° and compare the observed colocalization to that. Also, in the main text, the following speculation here regarding the previous data on integrins, is confusing with percentages, the derivation of which is not really explained, and should be better reasoned.

Fig 6. The colocalization with integrins or Talin should also be analysed by alternative methods, like immunofluorescence.

Fig 7. Again, alternative methods for colocalization should be included.

Fig 8. Internalization vs surface-binding. Include higher magnification images in single channels.

In the end, it remains very confusing how the authors suggest the paracrine sEV uptake would work. Mediated by integrins or not (if not, what then)? What is known about the adhesion receptors in these cell lines? Could authors do proteomic

analysis on eth sEVs to find out the potential candidates of differential targeting?

minor comments:

The authors should give brief explanations and justifications to the techniques they have used. E.g. The rationale of Tim-4 affinity purification of the EVs
mGFP-GPI diffusion assay

Figure 2d. The p values reported in the image are in the order of 10 to the ⁻²³⁵. These incredibly low -values are often obtained from data with very high n, as here is the case too. Reporting then in such manner does not feel meaningful, but instead <0.0001 could be used.

The materials and methods seem to be lacking a lot of details.

Reviewer #3

(Remarks to the Author)

This paper reports some important and novel finds. There are essentially three parts to the paper. Firstly, there is the introduction of novel techniques for investigating interactions and identities of individual sEV vesicles level. By expressing fluorescently-tagged membrane proteins in the parent cell, the membrane content and characteristics of individual EV 'off-spring' were imaged with exquisite specificity using novel imaging techniques. This showed that sEV of similar size, had different lipid 'fluidities'. Secondly, using an extension of this imaging approach the authors showed that the uptake of individual sEVs occurred without membrane fusion (as has often been supposed) as the fluor-tagged markers did not translocate to the recipient cell. Instead the authors give compelling evidence for internalisation of the sEV and showed a dependence on dynamin and that the internalisation site correlation with the loci of LAMP. They conclude that the sEVs were internalised by a process similar to conventional phagocytosis but with a dependence of caveolae. Given that conventional phagocytosis is usually triggered by larger targets ($c > 1 \mu m$ rather than the sEVs measured here at 70nm), this is a novel and thought-provoking finding. The question of how such small particles can trigger the cell to respond by quasi-phagocytosis is tackled in the third part of the paper. The authors show that intracellular signalling events, especially an elevation of cytosolic Ca^{2+} , are associated with the uptake of sEVs, and that the cytosolic Ca^{2+} signal probably activates calcineurin. The latter section is probably the weakest and the authors may consider the following points.

1. On page 2 14 and 19, it is stated that there was an "immediate increase in intracellular concentration of Ca^{2+} after sEV addition". The data seems to show long time delays between adding the sEVs and the Ca^{2+} spike (see fig 7b) and long and variable delays in the accompanying movie (Suppl movie 7).

Similarly, it was reported that "the intracellular Ca^{2+} response is triggered by the binding of sEVs to the PM". If there is evidence for this simple sequence it should be stressed. In suppl movie 7, some cell have complex Ca^{2+} signalling (with multiple Ca^{2+} peaks etc).

2. Figure 7a shows that the Ca^{2+} signal was global, ie through the whole cell. Was there any evidence that the sEV sedimented on to the cells uniformly and within this time scale.

3. Clearly these questions arise because the time resolution was slower (1/20sec) and spatial resolution was poorer for detecting the Ca^{2+} event(s) than the other imaging techniques reported here. This obviously makes it difficult to correlate the two events

Reviewer #4

(Remarks to the Author)

In this study, the authors utilized superresolution microscopy to investigate the fate of various "subtype" single extracellular vesicles (EVs) after internalization by recipient cells. They confirmed that all types of EVs are internalized through fluid-phase endocytosis or alternative pathways, including caveolin/dynamin-mediated mechanisms, as previously described (e.g., Costa Verdera et al., 2017; de Jong et al., 2020). The fact that both CD63 and CD9 vesicles are internalized is not surprising since those molecules seem to not be involved in the uptake process (Tognoli et al., 2023). Additionally, they observed and proposed a mechanism for EV adhesion that facilitates internalization through a calcium-dependent process. The superresolution imaging employed is impressive and of the highest quality, documenting the internalization of EVs at an unprecedented level. This includes the quantification of EVs recycled through recycling endosomes. Co-localization with molecules such as integrin and talin, which may facilitate EV docking, is also thoroughly documented. These observations confirm many previous studies that established a correlation between EV docking on the cell surface and the presence of integrin and related machinery (e.g., Altei et al., 2020).

However, the core observations and proposed mechanistic depictions lack novelty. It is well-established that most EVs are internalized through fluid-phase endocytosis, and the calcium regulation of dextran (a fluid-phase marker) was observed long ago (e.g., Sagi-Eisenberg et al., 1983). Similarly, the connection between internalized EVs and recycling endosomes, which the authors claim as novel, has already been described both morphologically and phenotypically (e.g., Walsh et al., 2021).

Furthermore, the proposed mechanism is demonstrated using drugs that lack specificity and through the overexpression of dominant mutants. Deciphering the mechanism at the molecular level would require knockout models and at least one independent method (such as biochemistry or non-imaging cell biology techniques) to establish the phenotype definitively. Overall, this study constitutes a high-standard documentation of EV internalization characterization and imaging but falls short in providing novel mechanistic insights.

Reviewer #5

(Remarks to the Author)

In this paper by Hirose et al, endocytosis and singling of different classes of extracellular vesicles is explored. First the authors use an interesting fluorescence method to show that vesicles have heterogeneous amounts of different tetraspanin markers. Thus, they classify these vesicles into different sub-types according to their cargo. When these vesicles were added to cells, they co-localized with specific membrane marker such as caveolae, integrins, and talin, and were internalized in a dynamin-dependent manner. The binding of EVs from other types of cells induced a cytosolic calcium increase that lead to an activation of Src and PLC. This enzyme/ion cascade activated calcineurin and promoted EV uptake into recycling organelles. From these studies, the authors develop a global model of paracrine (cell-to-cell) EV adhesion and uptake. In general, I find the paper interesting. I have several specific comments that the authors might consider to improve the manuscript.

1. The concept of autocrine and paracrine in this manuscript is a bit complex and I feel maybe not appropriate. While the authors show that many of the effects are limited to those that occur between different cell types, I am not sure if it is appropriate to present this work as a difference between “autocrine” and “paracrine” types of interactions in an organism. In general, I would add a discussion to temper this type of language and its interpretations.
2. In figure 1 it would be helpful to discuss or present how a random distribution of 4 proteins that were sorted into a single population of EVs would be distributed and how this distribution is different from the one the authors measure with their labeling methods assuming the number of copies the authors quantitate in their images applies. To what degree does the measured data diverge from stochasticity?
2. Please discuss how membrane packing defects and fluidity would affect the behavior of EVs in the discussion.
3. Figure 3 has a very nice experimental method but EV endocytosis has already been shown to be influenced by dynamin. Please expand on how these data add to the existing literature.
4. Again, figure 4 is experimentally nice but how often were these behaviors observed? The authors should present some statistical analysis of the frequency or generality of these behaviors in the figure and text with errors.
5. It would be very helpful to show larger images of the entire cell in super-resolution such that the images and overlap between the probes could be evaluated in Figure 5 by the reader. This is also true for Figure 6, where sEVs are co-localized with talin and integrins. While I appreciate the quantitative presentation, pairing with the actual whole-cell imaging data in the figure would be appropriate.
6. “The lipid composition of caveolae is similar to that of lipid rafts, and GPI-anchored proteins, a representative raft marker, are concentrated in caveolae upon cross-linking.” As a control I believe it is important to show that the GPI-marker is a good and specific marker for caveolae in this system. Please include this control.
7. The role of dynamin in caveolae is under dispute and complex (Parton et al. Nature Review MCB 2024). Please discuss and evaluate how this role for dynamin could impact the author’s data and conclusions.
8. Caveolae are on-average around 90 nm in diameter. An sEV is around the same size. How do the authors propose that caveolae can endocytosis an object of the same size or larger? Please discuss.

Version 1:

Reviewer comments:

Reviewer #1

(Remarks to the Author)

In this revised version, Dr. Suzuki and colleagues have addressed some of my concerns. However, critical issues remain unresolved:

1. General Contribution to the Field of EVs. The authors claim that single-molecule detection is crucial to define the uptake of EVs and that previous studies were unable to address this at the single-object level. While I agree with the authors’ statement, their work does not significantly advance the field.
 - o Dynamin Inhibition Experiment: The experiment is inconclusive, likely because dynamin inhibition is incomplete. Despite a statistical difference, significant uptake remains even with dynamin inhibition (Fig. 3H). The statistical difference suggests that a fraction of EVs is dynamin-dependent, but others are likely internalized via dynamin-independent processes. This experiment could be improved by using specific inhibitors such as Dynasore or Dyngo (Macia et al., Dev Cell 2006; McCluskey et al., Traffic 2013).
 - o Response to Point 7: The authors state, “We found that EVs containing only CD63 have more raft-like domains and were colocalized with caveolae more preferentially than sEVs containing CD81 and CD9.” This is an overstatement since Fig. 5C clearly shows this is not true for sEV(4175-LuT)/recipient (WI-38). This suggests that additional molecules present in EVs might influence their trafficking.

- o Single EV Uptake: While I appreciate their effort to demonstrate caveolin or Lamp2 uptake of single EVs (a single event shown for each), this does not substantially alter the existing understanding in the field.
 - o Reliance on Statistical Inference: The work relies heavily on statistical inference from short co-localization events, which is an indirect approach.
2. Calibration and Estimation of Tetraspanins in EVs. The authors have added single-molecule calibration, but serious concerns remain regarding the calibration and estimation of tetraspanins in EVs:
- o Western Blot Calibration: The calibration for CD63 remains incomprehensible (Fig. 1C). The approach for CD9 and CD81 is also unclear. The straight forward method involves running EVs directly and blotting for tetraspanins; the ratio between the two bands could provide the substitution (first lane of the authors' blot). Currently, the authors compare signals from EVs with those from immunoprecipitated cell lysates, which is not relevant as the efficiency of protein sorting into EVs is unknown.
 - o Improved Methodology: To assess substitution, the authors should immunoprecipitate EVs or load more EV material to detect the labeled band and compare it with the wild-type protein. Additionally, it is unclear why CD81 wild type can still be detectable when an anti halo antibody was used to immunoprecipitate CD81-Halo from the cell lysate.
 - o Unclear Calibration: The calibration process remains unclear and should be revised for better accuracy and transparency.
3. Quantification of CD63, CD9, and CD81 by Imaging The imaging results suggest that EVs contain variable fractions of the three tetraspanins, raising concerns about the conclusions in Fig. 5C. Specifically:
- o It is unclear how the frequency of events in Fig. 1F does not sum up to 100% of events. In some cases the frequency is higher than 100% (e.g. compare CD9 with CD63 and CD81 mGFP).
 - o The data suggest that the three populations behave similarly, with the only consistent result being increased colocalization with Lamp2 compared to CAV1 or AP2.
 - o The statistical significance of these events is limited due to the wide distribution of average pairwise distances. Transient low distances do not unambiguously indicate internalization.
4. Classification of LAMP2 as a Phagocytosis Marker The claim that LAMP2 is a marker of phagocytosis and that the observed process qualifies as phagocytosis is problematic:
- o Phagocytosis Specificity: Phagocytosis is a highly specific process, primarily observed in antigen-presenting cells, involving membrane zippering around a particle. LAMP2, as a type I membrane protein containing a GYxxΦ motif (Yamaguchi et al., J Biochem 2024), is directed to endosomes or lysosomes via clathrin-coated vesicles. It is surprising that the authors do not observe colocalization of LAMP2 single molecules with clathrin-coated vesicles, which would align with its known trafficking pathway.
 - o Alternative Explanation: The observed process might involve clathrin-coated vesicle formation rather than phagocytosis. To rule out clathrin-mediated endocytosis, the authors should demonstrate transferrin colocalization with clathrin-coated vesicles as a positive control.
 - o Discrepancy with Cited Literature: The paper by Leone et al. 2017, cited by the authors, does not classify the endocytic process involving LAMP2 as phagocytosis or macropinocytosis but suggests it is clathrin-independent. The authors should address this discrepancy and avoid referring to the process as phagocytosis without sufficient evidence.
5. Calcium Stimulation by EVs The Ca⁺⁺ stimulation induced by EVs in the authors' system is intriguing but unclear. Specifically:
- o It remains uncertain whether EV binding alone is sufficient to induce the calcium spike.
 - o This could be clarified by using inhibitors like Dynasore or Dyngo to block dynamin, which would help determine whether internalization is necessary for the calcium response.

The authors demonstrated a continuous distribution of tetraspanin molecules in EVs, ranging from 1 to 50 (Fig1D). Although they employ single-molecule detection, their localization relies on the PALM/STORM approach, which requires multiple localizations. This raises the possibility that their observations are biased toward larger EVs containing more tetraspanins. Smaller EVs that truly undergo internalization through caveolin or Clathrin might remain untracked.

Overall, while the authors have made some improvements, the manuscript still requires substantial revisions to address the unresolved issues outlined above.

(Remarks on code availability)

Reviewer #2

(Remarks to the Author)

The authors have added important clarity to the manuscript, by adding more data as well as more important information required to follow the experiments.

(Remarks on code availability)

Reviewer #3

(Remarks to the Author)

The authors have addressed all the specific points which were raised (by me). Their responses show that they appreciate the importance of some points and have also done additional work. The most notable new work (shown in Suppl fig 18)

clearly demonstrates the increases in PLC signal and the subsequent Ca²⁺ signal in individual cells. There was a variable 'latency' as is often seen (eg uncaging IP3 in individual cells can give a slightly later and variable Ca²⁺ signals.) It is a pity that the authors did not show (or comment on) the spatial data for this effect as it may provide evidence for spatially restricted signalling by individual vesicles. However these are minor points compared to the major successes in the paper. I feel that in its present form this paper makes an important contribution to the field of extracellular vesicle physiology and is of the high quality expected of papers published in Nat Commun.

(Remarks on code availability)

Reviewer #4

(Remarks to the Author)

My concerns have been addressed, thank you for the response in the rebuttal letter.

Significant work has consolidated the internalization model, and super-resolution nanoscopy, the real strength of the study, is outstanding.

(Remarks on code availability)

Reviewer #5

(Remarks to the Author)

The authors have addressed my specific concerns and questions from the first round of review with additional text and figures.

(Remarks on code availability)

Version 2:

Reviewer comments:

Reviewer #1

(Remarks to the Author)

With the additional experiments and modifications to the text, the authors have successfully addressed my remaining concerns. I appreciate the effort put into the revisions and the clear explanations provided in the rebuttal letter. Thank you for your thorough and thoughtful work.

(Remarks on code availability)

REVIEWER COMMENTS

Reviewer #1 (Remarks to the Author):

In the paper entitled: “Uptake of small extracellular vesicles by recipient cells is facilitated by paracrine adhesion signaling”. Dr Suzuki and colleagues characterized the EVs released by Prostate cancer cells and then studied their interaction with acceptor cells using high resolution microscopy approaches.

The authors have a very interesting observation that is the capacity of certain EVs to signal with specific acceptor cells. I would suggest developing their work around this discovery. Instead, although the authors have developed sophisticated imaging techniques the conclusion on EV endocytosis are not novel and have problematic data interpretation that rise serious concerns.

We appreciate the reviewer's thoughtful and constructive comments. In response, we performed additional experiments and revised the manuscript in accordance with all the reviewer's suggestions. We believe that the revisions have significantly strengthened the manuscript.

The major concern I have are the following:

1. The authors use of Lamp2 as a marker of phagocytosis. Lamp-2 is a marker of lysosomes and does not label the formation of the phagocytic cup. Indeed, as clearly stated in their reference 23 (Marion S et al., Traffic 2011), lamp2 colocalization is detected after 2 hours of administration of beads used to detect the phagosome, or in other words define the fusion of the phagosome with lysosomes (phagocytic maturation). Therefore, all these data are mis-interpreted.

Thank you for your thoughtful and critical comments on the localization of the Lamp-2 molecules. Indeed, Lamp2 is a molecule predominantly found in lysosomes, and is required for fusion of lysosomes with phagosomes. However, we observed LAMP2-mEos4b using TIRF microscopy (TIRFM), which only captures fluorescence from molecules within approximately 200 nm of the glass surface. This means that we observed only LAMP2-mEos4b on the cell plasma membrane and possibly lysosomes adjacent to the cell plasma membrane. Then, we tried to observe SiR-lysosome, a

lysosomal marker by TIRFM to examine whether lysosomes adjacent to the cell plasma membrane are visible by TIRFM. While we observed bright fluorescent signals of SiR-lysosome under oblique angle illumination (which illuminates deeper into the cell interior), no fluorescent signals were observed by TIRFM (Supplementary Fig. 8 in the revised version). We obtained similar results with other organelle markers such as STING (ER) and Rab6 (Golgi). In contrast, we observed numerous fluorescent spots of LAMP2-mEos4b molecules by TIRFM as well as by oblique angle illumination. Furthermore, most of the LAMP2-mEos4b molecules visualized by TIRFM underwent lateral diffusion across wide areas (Supplementary Fig. 8 in the revised version). These additional results explicitly demonstrated that LAMP2-mEos4b molecules visualized by TIRFM are present on and near the cell plasma membranes and are not in the lysosomal membrane. Furthermore, a recent study also reported the presence of LAMP2 in the PM, showing that LAMP2 in the host cell PM serves as a receptor for the *Trypanosoma cruzi* metacyclic trypomastigote surface molecule, g82 (Rodrigues et al., Cellular Microbiology, e13003, 2019). Another study also showed that LAMP2 is an endocytic receptor on the surface of human monocyte-derived dendritic cells (Leone et al., J. Immunol., 199, 531-546, 2017). These results suggest that the confocal microscopic images of LAMP2 colocalized with beads shown in Reference 23 represented both a phagocytic cup on the cell plasma membrane and lysosomes (we deleted Reference 23). These results explicitly indicate that the image sequence of LAMP2-mEos4b by TIRFM shows the early stage of phagocytosis on and near the cell plasma membrane. Unless we show these results, the readers will be very confused. We appreciate the reviewer's comment very much. Please also see the response to point 2, showing evidence that LAMP2 visualized by TIRFM is a phagocytosis marker.

These results are added to Supplementary Figure 8 and we made a new paragraph to show that LAMP2 is a phagocytic marker from pages 9 to 10.

2. The misinterpretation of these data rise serious concern regarding the capacity of the microscopy pipeline to be able to discern events that happen at the cell interface and events that take place within the cell. In addition, it is not clear the time that is needed to observe "true" internalizations.

As elaborated in point 1, our super-resolution observations of LAMP2-mEos4b by TIRFM captured the phagocytosis process occurring on and near the plasma membrane. This process likely represents the transition of LAMP2 from the cell plasma membrane to lysosomes. To further substantiate our findings, we conducted additional experiments involving slow super-resolution imaging of sEVs with LAMP2 and CAV1. The super-resolution microscopic images were acquired at 1 frame/second over an extended period of 5 minutes (Fig. 4f-i in the revised version). Fig. 4g and 4i show that LAMP2-mEos4b molecules gradually accumulated beneath

an sEV particle on the cell PM. Subsequently, the fluorescent intensity of CD63 in the sEV particle and LAMP2-mEos4b molecules were synchronously decreased and both disappeared within 1-2 minutes. These results further support that LAMP2 visualized by TIRFM represents the early stages of phagocytosis on and near the cell plasma membrane.

Furthermore, the fluorescent intensity of CD63 in the sEV particle and CAV1 molecules were also synchronously decreased, and both disappeared within 1-2 minutes. These results indicate that the uptake processes via both caveolae and phagocytosis occur within a relatively brief period of several minutes. We hope this clarification and the additional experimental evidence address the reviewer's concerns and strengthen our conclusions.

We added these results to Figure 4f-i and their explanation from the second paragraph on page 11 to the first paragraph on page 12.

3. Indeed, the authors observe events in a very short time scale. At this time scale many endocytic events cannot take place. For instance, the average life of a clathrin coated vesicle is of ~1 min (e.g. Ehrlich M et al., Cell 2004; Aguet et al., Dev Cell 2013). And similar considerations can be done for phagocytosis (e.g. Vorselen D et al, eLife 2021).

As described in point 2., we were able to capture the internalization phenomenon through an additional experiment involving 5-minute observation periods (Fig. 4f-i in the revised version). To more accurately capture the entry and exit of molecules, even higher-speed observations (5 ms) would be necessary. However, there is an inherent trade-off between the photobleaching of fluorescent molecules and temporal-spatial resolution. Given this constraint, the spatial precision in our study is limited to an observation time of 25 seconds. Our data (Fig. 5b, c) can be interpreted as an assembly of results from observing various stages of the internalization process occurring asynchronously on the cell PM. Despite the limitations in continuous observation time, the real-time super-resolution imaging at 21-nm spatial resolution and quantitative analysis methods developed in this study can still achieve the goal of identifying which uptake structures sEVs enter. In the time-lapse imaging in the additional experiment, the number of localization of LAMP2-mEos4b and CAV1-mEos4b in the individual membrane structure was less than 20% compared to those obtained by real-time imaging shown in Fig. 4 and Fig. 5. The small numbers of localization of mESO4b made it difficult to quantitatively estimate the number of pairwise distances shown in Fig. 5b and c. Therefore, we believe that the method employed in this study provides valuable insights into the dynamics of molecule internalization, even though we cannot follow a single event from start to finish due to the mentioned technical constraints. This approach, while not without limitations,

represents our best effort to balance the need for high-resolution data with the physical constraints of fluorescent molecules and imaging technology.

4. True internalizations should consist of: a. the particle (e.g. EV) is entrapped in a forming cargo (e.g. caveolae); b. the particle colocalize for a certain amount of time, necessary to finish the cargo or being pinched by dynamin (~20 s, see Cocucci et al., MBOC 2014) and then, c. move in the cytosol either through direct movement or randomly diffusing. The data presented here do not clearly characterize the dynamics of these events, probably for the short time of observation (see points 1 and 3).

As mentioned in point 2, we conducted additional experiments with extended 5-minute observation periods. This allowed us to successfully capture the internalization phenomenon for individual sEVs at each step of the process. By observing single sEVs over this longer timeframe, we were able to track the complete sequence of internalization events from initial cell membrane interaction to full uptake. This approach provided us with a more comprehensive understanding of the sEV internalization dynamics and the role of LAMP2 in this process.

5. The authors suggest an interesting mechanism: EVs can stimulate the cell to induce their uptake. It is also interesting that the event seems to be specific, or in other words that the signal is not autocrine. It would be useful to overexpress the integrins that are considered important for signaling in donor cells and see whether this can result in stimulation by their same EVs. In addition, it is well-known that there are two types of caveolae: short lived that do a kiss-and-run fusion and long lived (see Pelkman and Zerial Nature 2005 and Boucrot et al JCS 2011). Therefore, the short lived caveolae that appear on the plasma membrane are the first type. In the work of Pelkman it has been demonstrated that kinases can control the dynamics of caveolae. It would strength the paper if the authors could demonstrate a similar dynamic on caveolae formation rather than concentrating on dynamin, which is probably a downstream step.

Thank you for your interest in our results. We conducted experiments to observe Ca^{2+} signals when sEVs derived from PC3 cells overexpressing integrin $\beta 1$ were added to PC3 cells (Supplementary Fig. 15). We observed no Ca^{2+} response upon the addition of the sEVs, indicating that the low amount of integrin $\beta 1$ in sEVs is not the reason for

the absence of Ca²⁺ response by the autocrine binding. While this mechanism is intriguing, future research will be necessary to be fully elucidated.

We added these results to Supplementary Figure 15 and their explanation in the first paragraph on page 18.

We appreciate the reviewer's insight regarding Src kinase's role in caveolae regulation. This is indeed an interesting aspect. Pelkman et al. demonstrated that siRNA of six kinases, including SRC, KIAA0999, and DYRK3, inhibits caveolae-mediated uptake. They also showed reduced uptake with SRC inhibition (PP2 treatment). We examined whether colocalization of sEVs with caveolae is inhibited after Src inhibition with PP2 using TIRF microscopy and the same analysis method. We found that colocalization of sEVs with caveolae after PP2 treatment was slightly smaller than after PP3 treatment (Supplementary Fig. 19). These results suggest that, in addition to the dynamin activity, Src activity also partially may promote the uptake of sEVs via caveolae.

We added these results to Supplementary Figure 19 and their explanation in the second paragraph on page 21.

6. The authors have strong claims on the dynamics of EVs generalizing their data. They should dump their conclusion because this system of signaling and internalization might be valid only in this model and not in others.

Thank you for the valuable comment. We conducted experiments using only nine combinations of cells and sEVs (PC3, 4175-LuT, MSC, PZ-HPV-7, WI-38 cells as shown in Fig. 7d), and we agree with the reviewer's comment, while we believe this system can serve as a model for the uptake mechanism of cancer cell-derived sEVs by normal cells.

We deleted the phrase "suggesting the generality of the prompt signal transduction after sEVs bind to recipient cell PMs." in the fourth line from the bottom on page 19 of the original version (this corresponds to the bottom line on page 24 of the revised version). Furthermore, to remind the readers that we are not generalizing the data, we acknowledged that the uptake mechanism may vary with different cell-sEV combinations in the first paragraph on page 26 of the revised version.

7. The fact that EVs can be internalized by clathrin endocytosis or caveolae has already been demonstrated. Therefore, I would focus on the signaling mechanism which is novel and interesting even if related only to this cellular model.

Thank you for the valuable comment. We agree with the reviewer's comment on the point that clathrin and caveolin are related to the internalization of EVs. Previous studies examined whether the content of internalized EVs in cells changed after

treatment of cells with the inhibitors of clathrin-mediated endocytosis (For example, Horibe et al., BMC Cancer, 18, 47, 2018) or after knockdown of uptake-related proteins. However, uptake-related proteins such as clathrin and caveolin-1 associate with a variety of molecules, and may affect uptake via multiple pathways as we described in discussion (second paragraph on page 22). The process of internalization of EVs through individual membrane structures has never been directly visualized in living cells at high spatial resolution and the quantitative analysis has never been done as far as we know. Even if they were not recognized as new results, in our study, we examined whether internalization pathways of sEVs are dependent on EV subtypes. We found that EVs containing only CD63 have more raft-like domains and were colocalized with caveolae more preferentially than sEVs containing CD81 and CD9. We also found that the localization of sEVs containing CD63 into caveolae may be induced by the formation of raft-like domains underneath the sEVs. These novel findings, along with the associated signaling mechanisms, are key highlights of our research.

To make the novelty of our research on the internalization of sEV subtypes easier to understand, we revised the following part.

The second line on page 2, Abstract

Previous version;

However, how sEVs are internalized by recipient cells **is unclear**.

Revised version;

→However, how sEVs are internalized by recipient cells **has not been directly examined**.

The 4th-5th line on page 3

Previous version;

How sEVs are internalized by recipient cells **is poorly** understood.

Revised version;

→how sEVs are internalized by recipient cells **has not been fully** understood.

The 8th line on page 3

Previous version;

pathways and factors that determine sEV uptake **are unknown**

Revised version;

→pathways and factors that determine sEV uptake **have not been directly examined**

The second line in the second paragraph of page 9

Previous version;

We aimed to determine how...

Revised version;

→We aimed to directly determine how...

The first to second line of Discussion on page 21

Previous version:

The mechanisms involved in the uptake of tumor-derived sEVs by recipient cells are poorly understood.

Revised version:

→The mechanisms involved in the uptake of tumor-derived sEVs by recipient cells have not been directly examined.

Minor comments:

1. Western blot in figure 1 is unclear. The antibody against tetraspanins should detect both the wild type and chimera protein but this is not the case. This western should be improved.

We apologize for the insufficient explanation of our western blotting data. As you noted, we attempted to simultaneously detect bands for wild-type and chimeric proteins within the same sample. However, due to the low sensitivity of anti-CD9 and anti-CD81 antibodies, we couldn't detect wildtype and tagged markers in the same sample simultaneously. In the additional experiments (Supplementary Fig. 2), we concentrated CD9 and CD81 to detectable levels using IP or overexpression. We then compared the quantity with those of endogenous and tagged marker molecules in the sEVs used in our study. The results showed that the ratio of the tagged CD9 and CD81 marker proteins to endogenous ones was approximately 0.2. The ratio of tagged CD63 to the endogenous one was about 0.5 (Fig. 1c). This additional data provides a clearer picture of the relative abundance of tagged versus endogenous markers in our sEV samples.

We added these results to Supplementary Figure 2 and their explanation in the 7-10th lines in the second paragraph on page 5.

2. In figure 1 they show single molecule calibration. However, it is necessary to show the single step photobleaching events to ensure that the single TMR are single molecules. Are the imaging parameters used to acquire the single molecules the same as the one used to acquire the EV signal? It is important to show linearity of the detectors and of the laser power to ensure a correct calibration. These data are missing in the actual manuscript but should be shown.

Thank you for your comment. We added single-step photobleaching event data (Supplementary Fig. 1c, the first line on page 5) to address this concern. Additionally, the linearity between the number of molecules in clusters and the fluorescence intensity in our observation system has been confirmed by multiple previous studies (e.g. Kasai et al., J. Cell Biol., 192, 463-480, 2011; Suzuki et al. Nat. Chem. Biol., 8, 774-783, 2012; Morise et al., Nat. Commun., 10, 5245, 2019). We cited these references in the method section (the third to the second lines from the bottom on page 37).

3. I do not understand what the authors mean with “membrane packaging”. In addition an easy explanation for the higher fluorescence of certain EV in comparison to other might be just dependent on their lipid composition. I would suggest to substitute “membrane packaging” with “membrane composition”. A possibility is that some EVs have different curvature, however the data presented here do not support this possibility since the EVs have all similar diameter. Conversely, the amount of PS may drive the binding of this peptide. I would also suggest citing: Pranke IM et al, JCB 2011.

We apologize for any confusion. In this paper, we refer to the degree of membrane defects as “membrane packing” which was already described in the second paragraph of page 7. The previous research (Sato et al. RSC Adv. 2020, 10, 38323-38327 and Supplementary Figure 5) demonstrated that the probe we used in this study shows similar binding affinity (Kd) for liposomes (both 110 nm) with 0% PS and 20% PS. Therefore, this result shows that the PS content does not significantly affect our measurement of membrane packing in the current study. This clarification helps to address potential concerns about the influence of phosphatidylserine (PS) content on our membrane packing measurements.

We added their explanation in the lines 3-4 in the second paragraph on page 7.

4. It is widely accepted that EVs are internalized and do not fuse directly with the plasma membrane. It would be interesting to have a citation in this regard.

Thank you for the valuable comments. We cited Ref.21 and incorporated the description regarding our results (from the last line on page 8 to the second line on page 9).

Reviewer #2 (Remarks to the Author):

The manuscript "Uptake of small extracellular vesicles by recipient cells is facilitated by paracrine adhesion signaling" by Hirosawa et al. tackles an interesting topic of extracellular vesicle internalization. The authors use various high-end microscopy methods and raise interesting observations. However, I feel more efforts are required to increase the robustness of the findings.

We appreciate the reviewer's thoughtful and constructive comments. In response, we performed additional experiments and revised the manuscript in accordance with all the reviewer's suggestions. We believe that the revisions have significantly strengthened the manuscript.

First of all, the authors make an unjust statement that small EVs equal to exosomes. Current view in the field is that exosomes derive from multivesicular bodies' intraluminal vesicles. It is important to explain what the authors think the origin of their sEVs is and why, and why would they not be derived from other cellular origins. The tetraspanin markers, that the authors use, are not unique to exosomes. Could the source of sEVs relate to the cell type selectivity the authors report?

We sincerely appreciate your valuable comment regarding this important point. The reviewer is indeed correct that our current study does not specifically identify the origin of the small extracellular vesicles (sEVs) we examined. In response to the reviewer's insightful comment, we have carefully revised our manuscript. Specifically, we have removed the term "exosome" from the introduction section (line 1 on page 3).

The microscopic approaches in the manuscript is very difficult to follow. The authors have used various different methods, including single molecule localizations microscopies. It remains unclear at many instances if the data is based on true single molecule localizations, how this is verified (include raw data, photon counts, life times, accuracy, bleaching, blinking etc characteristics, localization maps), and when just bulk fluorescence is taken into account. For example Movie 1: is it single

molecule (does it bleach in one frame?) or is it single vesicle with several single molecules? The details of the usage of dSTORM and PALM are not explained nor the need for or the validation of single molecule resolution. 3D-dSTORM is not a trivial method, and would need careful validation.

We appreciate your insightful comment. To make the microscope setup easier to understand, we cited the references of our microscopic setup as follows.

In the second paragraph, page 37, the second paragraph on page 38, the last line on page 39:

Komura et al., Nat. Chem. Biol., 2016; Kinoshita et al., J. Cell Biol., 2017; Morise et al., Nat. Commun., 2019; Kemmoku et al., Nat Commun., 2024

The 4th line on page 40:

Suzuki et al., J. Cell Biol., 2007; Suzuki et al., J. Cell Biol., 2007; Suzuki et al., Nat. Chem. Biol., 2012; Komura et al., Nat. Chem. Biol., 2016; Morise et al., Nat. Commun., 2019; Fujiwara et al., J Cell Biol., 2023

We added the following data and citations according to the reviewers' suggestions.

The raw data of fluorescent intensity and single-step bleaching data of TMR (Supplementary Fig. 1c)

The photon counts of single fluorescent molecules per frame were 164.2 ± 4.4 and 100 ± 9.3 for TMR and mEos4b, respectively. (Methods section, the second paragraph on page 37 and the first paragraph on page 40) We estimated the photon counts according to Fujiwara et al. (J. Cell. Biol., 222, e202110160, 2024) and cited the reference.

Lifetimes, bleaching, blinking, and characteristics of mEos4b were previously reported in detail (Mori et al., 2024, Sci. Rep.; Kemmoku et al. 2024, Nat. Commun.; Fujiwara et al. 2024, J. Cell. Biol., 222, e202110160). We cited these references in the first paragraph on page 40.

The accuracy of the position of single molecules (mGFP, TMR, and SF650T) was 24 -26 nm, (Methods section, the second paragraph on page 45)

Localization maps (Supplementary Fig. 9a, top)

We described more details of the usage of mEos4b for PALM observation and the microscope setup in the method section titled “*Simultaneous dual-color observation of PALM movies of membrane invaginations and single-fluorescent particles of sEVs on living cell plasma membranes*” on page 39-40. Furthermore, we cited other references of our studies using PALM in the first paragraph on page 40 (Ref.14, Mori et al., Sci. Rep., 2024, Ref. 83, Fujiwara et al., J. Cell Biol., 2023).

Movie 1 shows 3D-single-particle tracking of an sEV containing CD63-Halo7 labeled with SF650T. We apologize for the insufficient description. We added this in the caption of Fig. 3b. This was already described in the caption of Movie 1.

The validation of 3D-single particle tracking was performed using a 100-nm-diameter multicolor fluorescent bead immobilized on a coverslip. The results are shown in Supplementary Fig. 7b and c.

Also, why do the authors use single molecule detection to follow protein recruitment to the plasma membrane. Typically ensemble TIRF imaging should better address this. In general, the authors should report their single molecule data in much more open and explainable manner, and also compare the data to the ensemble imaging when possible. For analysing colocalization, single molecule data is rarely a method of choice.

Thank you very much for this comment. Single-molecule imaging is a powerful tool for detecting transient and/or rare events in cells. For example, we found that PLC γ molecules were transiently recruited to the plasma membrane which cannot be observed using ensemble imaging which was performed as an additional experiment (Supplementary Fig. 17). The reason would be that only a small fraction of PLC γ was transiently recruited to the plasma membrane for short periods to trigger intracellular Ca²⁺ response. On the other hand, using single-molecule imaging, we detected an increase in the number of PLC γ molecules recruited to the plasma membrane upon sEV binding, leading to intracellular Ca²⁺ response (Fig. 7g). Furthermore, using single-molecule imaging, we demonstrated that PLC γ is recruited to FAK clusters away from the sEV, not directly beneath it (Fig. 7h-j). Such observations were only possible by combining single-molecule tracking with super-resolution video imaging. So far, we have observed such transient and/or rare events that cannot be detected by conventional microscopy (recruitment of raftophilic signaling molecules to GPI-anchored protein clusters, recruitment of G proteins to GPCRs, and interaction of Ras with RAF, etc.)(Suzuki et al., Nat. Chem. Biol., 8, 774-783, 2012; Kasai et al., J. Cell Biol., 192, 463-480, 2011; Murakoshi et al., PNAS, 101, 7317-7322, 2004, etc). As shown in Fig. 6a, single-molecule imaging of TMR-Halo7-integrin β 1/TMR-Halo7-GPI

and single-particle tracking of sEVs demonstrated transient recruitment of integrin β 1 and GPI-anchored proteins to the plasma membrane just beneath sEV. However, it is very hard to find colocalization between sEVs and these molecules by the conventional immunostaining method as shown in Supplementary Figure 12. This is found only by single-molecule imaging because the recruitment is transient.

To make the readers understand better why we chose the single-molecule imaging method, we added the following sentence in the second paragraph on page 14 and cited the following reference.

Single-molecule imaging is a powerful tool for detecting transient and/or rare events in living cells.

Citation: Suzuki et al., J. Cell. Biol., 2007; Suzuki et al., J. Cell. Biol., 2007; Tanaka et al., Nat. Methods, 2010; Suzuki et al., Nat Chem Biol., 2012; Koura et al., Nat Chem. Biol., 2016; Kemmoku et al., Nat. Commun., 2024

The article also contains a medley of different fluorophores, but it is not explained why they were chosen and the details of the each imaging set up are missing. Each figure legend should clearly state how the imaging was done and what is shown.

According to the reviewer's suggestions, we indicated which microscopy method was used to obtain the data in figure legends. In these figure legends, we did not show imaging methods.

Figure 1e; TIRFM

Figure 3b; oblique angle illumination

Figure 3g; confocal microscopy

Figure 6a; TIRFM

Figure 7f; TIRFM

Supplementary Figure 1a; TIRFM

Supplementary Figure 6b, c; TIRFM

Supplementary Fig. 7; oblique angle illumination

Furthermore, to explain why we used the fluorescent molecules for the experiments, we added the following sentences in the method section.

The 5th to 3th lines from the bottom on page 29

The HaloTag TMR and SF650T ligands were used for dual-color imaging with mGFP and mEos4b, respectively. The HaloTag SF650B ligand was employed for dSTORM imaging to estimate the diameters of sEV subtypes.

The 4th to 6th line in the second paragraph on page 39

Because the intensity of single-fluorescent spots of mEos4b is high and mEos4b does not form dimers, mEos4b was used to acquire PALM images.

Figure 1c: the western blot does not show any bands of the CD9 or CD81 fusion proteins at the same level as the membranes probed with the anti-tag antibodies. How is this explained? How is the 0.1-0.5 times reduced expression calculated? The quantification should be added.

We apologize for the insufficient explanation of our data, which was also noted in Reviewer 1's minor comment 1. Due to the low sensitivity of anti-CD9 and anti-CD81 antibodies, we were unable to simultaneously detect wild-type and tagged markers in the same sample. To address this, we conducted additional experiments (Supplementary Fig. 2) where we made tagged markers detectable for CD9 and CD81 through Immunoprecipitation (IP) or overexpression. We then performed comparative quantification with endogenous and tagged marker molecules contained in the sEVs used in this study. The results confirmed that the ratio of the tagged marker proteins to endogenous marker proteins is approximately 0.2-0.5. (regarding CD63, we estimated the ratio based on the result in Fig. 1c)

We added these results to Supplementary Figure 2 and their explanation in the second paragraph on page 5.

Fig 3g-h shows less sEVs on the cell membrane in cells with inactive dynamin. Would the conclusion then not rather be that there is more internalization? The necessary details on how the data is obtained are missing to allow interpretation of the data. In general the sum of EV fluorescence intensity does not appear like a right measure of EV uptake, as also the EVs trapped on cell surface and failing to internalize can give high signals. This analysis been used also in other figures. Authors should explain this and find more specific tools to examine internalized vesicles as opposed to cell-surface trapped vesicles. How long can the sEV be visualized inside the cells and can they be distinguished from the cell surface-bound ones?

We apologize for the lack of detail in our previous explanation. In Fig. 3 g-h, we observed a middle section of the cells using confocal microscopy as indicated in Fig. 3a. This approach allows us to count the number of internalized EVs. Furthermore, Fig. 3h represents the "increase" in the total fluorescence derived from EVs, which quantifies the internalization process. Images were acquired at the mid-section of

cells (~1.5 μm above the glass) using confocal microscopy. Furthermore, individual sEV particles attached to glass cannot be detected as a single fluorescent spot by confocal microscopy under the observation condition due to the low sensitivity (In other words, an average of 4 molecules of CD63-Halo7-TMR in an sEV particle cannot be recognized as a single spot by confocal microscopy under the observation condition, whereas they can be observed by TIRFM). We detected only EVs that were internalized and aggregated inside cells. We estimated the sum of the fluorescence intensity of aggregated sEV spots.

We added this description in the caption of Fig. 3g (pages 69-70) of the revised version.

Fig 4. The authors should be able to give some quantitative analysis data on the relative percentages of different internalization modes on this data.

Thank you for the valuable comment. We estimated the frequency of the colocalization events on pages 12-13 of the revised version as follows.

The frequencies of the colocalization events (≥ 1 frame) between sEVs and phagocytic regions or caveolae were 2.12 ± 0.88 events/ $\mu\text{m}^2/\text{min}$ (mean \pm SE) and 0.15 ± 0.12 events/ $\mu\text{m}^2/\text{min}$, respectively. Moreover, the number densities of sEVs, which showed colocalization with phagocytic regions or caveolae, were 0.090 ± 0.022 particles/ $\mu\text{m}^2/\text{min}$ and 0.031 ± 0.022 particles/ $\mu\text{m}^2/\text{min}$, respectively. These results indicate that sEVs colocalize with phagocytic regions more frequently than with caveolae.

Why single molecule imaging? How does this look when imaged in ensemble? Other methods and good controls for colocalization analysis should be sought.

As shown in Fig. 5, we accurately quantified the intensities of colocalization of sEVs with individual membrane structures obtained at 21 nm spatial resolution. This quantitative analysis is impossible by ensemble imaging (spatial resolution = 250~300 nm) like the bottom image shown in Fig. 4d and e, because the contours of the membrane structures are ambiguous, and it is impossible to judge whether sEVs are located within the membrane structures. Furthermore, we can observe the molecules only on the cell plasma membranes by total internal reflection fluorescence microscopy (TIRFM). For example, LAMP-2 is located in both lysosomes and phagocytic cups, but we can only observe LAMP-2 in phagocytic cups by TIRFM.

These are the reasons why we need to perform high-speed single-molecule localization microscopy in living cells.

Please also see the response to comments regarding Fig. 6 and Fig. 7 below. We described the reason why we employed the single-molecule imaging method.

Fig 5. Same concerns than in Fig 4. The colocalization of sEVs with Rab11 seems very weak or even absent. Proper controls should be included here to ascertain that the colocalization is not only random. A simple way to do this would be to turn one channel 90° and compare the observed colocalization to that.

Thank you for the valuable comment. In Fig. 5f, we implemented a randomization control by rotating one channel 180 degrees, represented by open symbols as already described in the method section. The bottom in Fig. 5f shows net values estimated by subtracting randomized data (open symbols) from raw data (closed symbols). This method provides a baseline for chance colocalization. Our analysis demonstrates that sEV colocalization with Rab11 consistently exceeded this randomized baseline across all experimental conditions. This approach allows us to differentiate between coincidental overlap and meaningful biological associations, thereby providing robust evidence for a significant interaction between sEVs and Rab11 throughout our study.

Also, in the main text, the following speculation here regarding the previous data on integrins, is confusing with percentages, the derivation of which is not really explained, and should be better reasoned.

In page 11 of the original manuscript, we described as following. “These results, together with our recent work²⁷ suggest that 20%–40% of sEV particles bound to laminin on cell PMs are internalized and transported to the recycling pathway in an intact or membrane-fused form.” The reason was described in the same paragraph, but we agree with the reviewer’s comment that the basis of the percentage is weak. Therefore, we replaced the percentage (20-40%) of sEV particles with “some fractions of sEV particles” (the first paragraph on page 14 of the revised version). Thank you very much for this comment.

Fig 6. The colocalization with integrins or Talin should also be analysed by alternative methods, like immunofluorescence.

Thank you for the critical suggestions. We performed additional experiments involving immunostaining of the active form of integrin and talin-1 (Supplementary Fig. 12). However, we were unable to detect quantitative colocalization, presumably due to limitations in the sensitivity of the technique. Namely, immunostaining cannot detect tiny clusters and transient colocalization. Meanwhile, as mentioned above, single-molecule imaging allows us to detect transient colocalization of sEVs with integrin and talin-1 (Fig. 6). These additional experiments are very effective in informing the readers why it was necessary to observe single molecules. We thank the reviewer for this comment.

We added these results to Supplementary Figure 12 and their explanation as follows (the second line from the bottom on page 15 to the second line on page 16).

The tiny and transient nature of integrin and talin-1 clusters formed in the PM beneath the sEVs is further suggested by the lack of significant colocalization of integrin and talin-1 with sEVs, as observed through dual-color conventional immunocytochemical analysis (Supplementary Fig. 12).

Fig 7. Again, alternative methods for colocalization should be included.

Thank you very much for your helpful suggestion. We observed stimulus-dependent recruitment of mGFP-PLC γ 2 to the cell plasma membrane in live cells (Supplementary Fig. 17). However, due to sensitivity limitations, we were unable to detect the enrichment of PLC γ in the plasma membrane upon stimulation. Consequently, we could not validate the colocalization of FAK with PLC γ . These additional experiments are very effective in informing the readers why it was necessary to observe single molecules. We thank the reviewer for this comment.

We added these results to Supplementary Fig. 17 and their explanation as follows (in the first paragraph on page 19).

The recruitment of PLC γ 2 to the PM after sEV addition was undetectable by confocal microscopy (Supplementary Fig. 17), suggesting that only a small fraction of cytosolic PLC γ 2 was recruited to the PM.

This may make sense because a small number of sEV (approximately 8 sEV particles/100 μm^2 , namely 200 sEV particles/cell) was sufficient to trigger the signal transduction as described on the 3rd-4th line from the bottom on page 24.

Fig 8. Internalization vs surface-binding. Include higher magnification images in single channels.

We appreciate your valuable suggestion. According to the reviewer's suggestion, we provided magnified images in Fig. 8a to enhance the visibility of the internalization process.

In the end, it remains very confusing how the authors suggest the paracrine sEV uptake would work. Mediated by integrins or not (if not, what then)? What is known about the adhesion receptors in these cell lines? Could authors do proteomic analysis on eth sEVs to find out the potential candidates of differential targeting?

In our previous work (Isogai et al.2024), we examined the molecular mechanisms in detail by single-particle imaging of sEVs derived from integrin KO-PC3 cells. We found that PC3-derived sEVs bind to laminin on the recipient cells, and integrin $\alpha6\beta1$ and $\alpha6\beta4$ in the sEVs are responsible for the binding of the sEVs to laminin. The sEVs derived from integrin $\alpha6$, $\beta1$, or $\beta4$ KO-PC3 cells were scarcely bound to the laminin on the recipient cell PM. We cited reference 35 in the first paragraph on page 14, in the first paragraph on page 16, in the first paragraph on page 18 and in the first paragraph on 25.

minor comments:

The authors should give brief explanations and justifications to the techniques they have used. E.g.

The rationale of Tim-4 affinity purification of the EVs
mGFP-GPI diffusion assay

Thank you for your comment. We apologize for the insufficient explanation. We have added details to the Methods section and main text to clarify our experimental procedures.

Tim-4 affinity purification of the sEVs
the second paragraph on page 7;

Subsequently, the same analysis was performed using sEVs isolated by the PS binding protein, TIM-4 beads affinity method, a purification approach that does not involve size fractionation.

The 5th and 7th lines on page 30;

“which bind to PS exposed on sEV surface”

“This method enables the purification of sEVs with minimal contaminants and without size fractionation”

mGFP-GPI diffusion assay

From the last line on page 8 to the second line on page 9;

“No occurrence of fusion was further corroborated by our single-molecule observations. Namely, we observed that no mGFP-GPI molecules were transferred to the recipient cell membrane...”

From pages 36 to 37;

We made a new method section titled “mGFP-GPI diffusion assay”

Figure 2d. The p values reported in the image are in the order of 10^{-235} . These incredibly low -values are often obtained from data with very high n, as here is the case too. Reporting then in such manner does not feel meaningful, but instead <0.0001 could be used.

Upon reviewing the submission guidelines, we need to write p-values as it is (in our paper published recently (Kemoku et al., Nat. Commun., 15, 220, 2024), we were

asked to write p-values in this way). Therefore, we have not made any modifications to this aspect of our manuscript.

The materials and methods seem to be lacking a lot of details.

Thank you for the valuable comment. In addition to the above-mentioned parts, we described more details in the Methods section to clarify our experimental procedures as follows.

*Cell sorter (pages 28-29)

*Manufacturer or source: FBS(Gibco), page 27; G418(Nacalai Tesque), page28; blasticidin (Invitrogen), page 28; Cy3-DOPE (synthesized as previously reported), page 28; Triton X-100 (MP Biochemicals), page 34; DMPC (Avanti Polar Lipids, Inc), page 51.

*Laser power used for each experiment (page 38, 43, 50)

*The reason we used TMR, SF650T, and SF650B (page 29)

*mGFP-GPI diffusion assay (pages 36-37)

* The reason we used mEos4b for PALM (page 39)

We thank the reviewer for these thoughtful and constructive comments.

Reviewer #3 (Remarks to the Author):

This paper reports some important and novel finds. There are essentially three parts to the paper. Firstly, there is the introduction of novel techniques for investigating interactions and identities of individual sEV vesicles level. By expressing fluorescently-tagged membrane proteins in the parent cell, the membrane content and characteristics of individual EV ‘off-spring’ were imaged with exquisite specificity using novel imaging techniques. This showed that sEV of similar size, had different lipid ‘fluidities. Secondly, using an extension of this imaging approach the authors showed that the uptake of individual sEVs occurred without membrane fusion (as has often been supposed) as the fluor-tagged markers did not translocate to the recipient cell. Instead the authors give compelling evidence for internalisation of the sEV and

showed a dependence on dynamin and that the internalisation site correlation with the loci of LAMP. They conclude that the sEVs were internalised by a process similar to conventional phagocytosis but with a dependence of caveolae. Given that conventional phagocytosis is usually triggered by larger targets ($c > 1 \mu\text{m}$ rather than the sEVs measured here at 70nm), this is a novel and thought-provoking finding. The question of how such small particles can trigger the cell to respond by quasi-phagocytosis is tackled in the third part of the paper. The authors show that intracellular signalling events, especially an elevation of cytosolic Ca^{2+} , are associated with the uptake of sEVs, and that the cytosolic Ca^{2+} signal probably activates calcineurin. The latter section is probably the weakest and the authors may consider the following points.

We appreciate the reviewer's thoughtful and constructive comments. In response, we performed additional experiments and revised the manuscript in accordance with all the reviewer's suggestions. We believe that the revisions have significantly strengthened the manuscript.

1. On page 2 14 and 19, it is stated that there was an "immediate increase in intracellular concentration of Ca^{2+} after sEV addition". The data seems to show long time delays between adding the sEVs and the Ca^{2+} spike (see fig 7b) and long and variable delays in the accompanying movie (Suppl movie 7). Similarly, it was reported that "the intracellular Ca^{2+} response is triggered by the binding of sEVs to the PM". If there is evidence for this simple sequence it should be stressed. In suppl movie 7, some cell have complex Ca^{2+} signalling (with multiple Ca^{2+} peaks etc).

We appreciate your insightful comment. Fig. 7b shows intracellular Ca^{2+} mobilization about 2 minutes after the addition of sEVs to the recipient cell (please note that the large peak at about 20 min was caused by ionomycin). To address the issue that the reviewer raised, we performed additional experiments with lower concentrations of sEV suspension. We found that the delay in calcium spikes is dependent on sEV concentration (Supplementary Fig. 13). When we added an unconcentrated serum-free PC3 cell culture supernatant which contains lower amounts of sEVs, intracellular Ca^{2+} response in the recipient PZ-HPV-7 cells was initiated 15-30 min after the addition of the supernatant (Supplementary Fig. 13b). On the other hand, when the

serum free PC3 cell culture supernatant was added to PC3 cells, or the serum-free PZ-HPV-7 cell culture supernatant was added to PZ-HPV cells, no Ca^{2+} response was observed in the recipient cells (Supplementary Fig. 13c, d).

We described these results in the first paragraph on page 17 of the revised version as follows:

Furthermore, the temporal delay in Ca^{2+} spike initiation inversely correlated with sEV concentration; notably, a reduction in sEV concentration resulted in a prolonged delay (Supplementary Fig. 13b-d).

Furthermore, instead of “immediate increase in intracellular concentration of Ca^{2+} after sEV addition”, we used the following statement. (in the first paragraph on page 17)

"The increase in intracellular Ca^{2+} concentration observed within a few minutes after sEV addition (Fig. 7a, b)"

This revision accurately describes the observed phenomenon while accounting for the concentration-dependent variability in timing.

Please also see the response to point 3, showing the elevation in Ca^{2+} signaling occurred around the time when the cumulative number of the recruited $\text{PLC}\gamma$ reached a plateau.

2. Figure 7a shows that the Ca^{2+} signal was global, ie through the whole cell. Was there any evidence that the sEV sedimented on to the cells uniformly and within this time scale.

Ca^{2+} signaling is not confined to the sites of sEV adhesion or internalization. Rather, it results from intracellular signal transduction leading to the opening of IP_3 receptors (IP_3R) in the endoplasmic reticulum. The diffusion coefficient of Fluo-8H in the cytosol was estimated to be $100 \mu\text{m}^2/\text{sec}$ (Villarruel et al., Biophys. J., 120(18):3960-3972, 2021), indicating that Ca^{2+} propagates throughout the entire cell within 100

milliseconds. On the other hand, the temporal resolution of our Ca^{2+} imaging with Fluo8H was 0.05 frame/sec (1 frame/20 sec).

Consequently, in this study, we observed Ca^{2+} signals that had propagated throughout the entire cell, rather than localized signals.

3. Clearly these questions arise because the time resolution was slower (1/20sec) and spatial resolution was poorer for detecting the Ca^{2+} event(s) than the other imaging techniques reported here. This obviously makes it difficult to correlate the two events

We appreciate the reviewer's crucial comment. To address the issue which the reviewer raised, we examined the timing of PLC γ recruitment and Ca^{2+} signaling with a 60-fold increase in the temporal resolution (0.3 sec per frame), as shown in Supplementary Fig. 18. The figure at the top shows the time course of Ca^{2+} concentration and the figure at the bottom shows the time course of the cumulative number of PLC γ spots recruited to the plasma membrane. These results explicitly show that the elevation in Ca^{2+} signaling occurred around the time when the cumulative number of the recruited PLC γ reached a plateau. The left figure (Cell#1) showed a slow increase in the cumulative number of recruited PLC γ and Ca^{2+} response occurred 4.5 minutes after the addition of the sEVs. Meanwhile, in the middle figure case (Cell#2), the Ca^{2+} response occurred earlier (about 3.5 minutes), and in the right figure case (Cell#3), it occurred earlier than others (about 2.5 minutes). The observed delay between PLC γ recruitment and Ca^{2+} signaling supports a mechanistic link between these processes. This additional data correlates the PLC γ recruitment with Ca^{2+} signaling and strengthens our findings.

We added these results to Supplementary Fig. 18 and their explanation in the first paragraph on page 19 (To further validate whether PLC γ 2 recruitment to the PM...).

Reviewer #4 (Remarks to the Author):

We appreciate the reviewer's thoughtful and constructive comments. In response, we performed additional experiments and revised the manuscript in accordance with all the reviewer's suggestions. We believe that the revisions have significantly strengthened the manuscript.

In this study, the authors utilized superresolution microscopy to investigate the fate of various "subtype" single extracellular vesicles (EVs) after internalization by recipient cells. They confirmed that all types of EVs are internalized through fluid-phase endocytosis or alternative pathways, including caveolin/dynamin-mediated mechanisms, as previously described (e.g., Costa Verdera et al., 2017; de Jong et al., 2020).

Thank you for your critical comments. Our understanding of these papers is as follows.

Costa Verdera et al., 2017 J.Controlled Release

In this study, the experiments with the inhibitors and after the knockdown of uptake-related proteins suggested that clathrin-independent endocytosis and macropinocytosis may be involved in the uptake of A431-derived EVs by the recipient HeLa cell. The internalized EVs labeled with PKH67 were observed by confocal microscopy.

de Jong et al., 2020 Nat.Communications

In this study, the authors developed CRISPR-Cas9-based reporter system for single-cell detection of EV-mediated functional transfer of small non-coding RNA molecules. We think that this is an excellent system to quantify the amount of released cargo from EVs (donor cell: MDA-MB-231). The authors evaluated EV uptake by measuring relative reporter eGFP activation in the HEK293T reporter cell of which uptake-related proteins such as caveolin-1 were knocked down.

Both the papers by Costa Verdera et al. and by de Jong et al. examined whether the amount of internalized EVs changed after the knockdown of uptake-related proteins in the recipient cell or the treatment of the recipient cell with the inhibitors. However, uptake-related proteins such as caveolin-1 associate with a variety of molecules, and may affect uptake via multiple pathways as we described in discussion (second

paragraph on page 22). Furthermore, in both studies, the uptake pathway of individual EV subtypes was not investigated.

We believe that the process of internalization of EVs through individual membrane structures has never been directly visualized in living cells at high spatial resolution and the quantitative analysis has never been done as far as we know. Even if they were not recognized as new results, in our study, we examined whether internalization pathways of sEVs are dependent on EV subtypes. We found that EVs containing only CD63 have more raft-like domains and were colocalized with caveolae more preferentially than sEVs containing CD81 or CD9. We also found that the localization of sEVs containing CD63 into caveolae may be induced by the formation of raft-like domains underneath the sEVs. These novel findings, along with the associated signaling mechanisms, are key highlights of our research.

To make the novelty of our research on the internalization of sEV subtypes easier to understand, we revised the manuscript as described in response to point 7 of reviewer 1.

The fact that both CD63 and CD9 vesicles are internalized is not surprising since those molecular seem to not be involved in the uptake process (Tognoli et al, 2023).

Our understanding of the paper is as follows

Tognoli et al, 2023 Communications biology

In this study, EVs derived from donor cells in which CD9 or CD63 was knocked out, were isolated. The authors found using luciferase activation assay, that the uptake percentage of these EVs by the recipient HeLa cells was indistinguishable from that of EVs derived from the intact cell. Meanwhile, the uptake percentage of EVs derived from the intact cell by the intact recipient cell was similar to that by the recipient cell in which CD9 or CD63 was knocked down. These results suggest that neither CD63 nor CD9 are involved in the uptake process.

We agree with the conclusion of this study. However, even if CD63 or CD9 is depleted, EVs derived from multi-vesicular endosomes (MVEs) or plasma membranes are present, and the composition of other molecules in EVs would not change because CD63 and CD9 are just cargos and are not involved in the biogenesis of EVs. We used these tetraspanins as EV subtype markers. We found at least two EV subtypes derived from PC3 cells and 4175-LuT cells are present, and these EV subtypes with different membrane fluidity behaved differently on the recipient cell plasma membranes (CD63 and CD9 do not alter these characteristics of EVs). We believe that these findings are explicitly novel.

Additionally, they observed and proposed a mechanism for EV adhesion that facilitates internalization through a calcium-dependent process. The superresolution imaging employed is impressive and of the highest quality, documenting the internalization of EVs at an unprecedented level. This includes the quantification of EVs recycled through recycling endosomes. Co-localization with molecules such as integrin and talin, which may facilitate EV docking, is also thoroughly documented. These observations confirm many previous studies that established a correlation between EV docking on the cell surface and the presence of integrin and related machinery (e.g., Altei et al., 2020).

Thank you very much for highly evaluating our imaging techniques. It took about 7 years to establish the techniques of super-resolution movie observation and the analysis methods.

However, the core observations and proposed mechanistic depictions lack novelty. It is well-established that most EVs are internalized through fluid-phase endocytosis, and the calcium regulation of dextran (a fluid-phase marker) was observed long ago (e.g., Sagi-Eisenberg et al., 1983). Similarly, the connection between internalized EVs and recycling endosomes, which the authors claim as novel, has already been described both morphologically and phenotypically (e.g., Walsh et al., 2021).

Thank you for your critical comments. Our understanding of these papers is as follows.

Sagi-Eisenberg et al., 1983 FEBS Lett.

This study showed that dextran induced Ca^{2+} uptake by rat mast cells.

We think this study is not related to our finding that the elevation of intracellular Ca^{2+} concentration evoked by sEVs on the cell plasma membrane facilitated the uptake of EVs via the dynamin-calcineurin-dependent pathway.

Walsh et al., 2021 J. Cell Biol.

In this study, the authors found that during the biogenesis of EVs in the neuron cell, the selection of cargos into EVs is regulated in recycling endosomes marked with Rab11.

This study did not examine the location of internalized EVs derived from donor cells and we think that this is not related to our study.

Furthermore, the proposed mechanism is demonstrated using drugs that lack specificity and through the overexpression of dominant mutants. Deciphering the mechanism at the molecular level would require knockout models and at least one independent method (such as biochemistry or non-imaging cell biology techniques) to establish the phenotype definitively.

Thank you for your thoughtful and critical comment and we agree with this comment. To address this issue, we conducted additional experiments using biochemical methods to investigate signal transduction induced by sEV binding. We examined whether src family kinase is activated upon the binding of sEVs on the recipient cells by the biochemical method. Our results demonstrated an increase in FAK Y861 phosphorylation in the recipient cells (PZ-HPV-7) following sEV addition (Supplementary Fig. 16). Furthermore, we confirmed that this phosphorylation is dependent on src kinase activity. This biochemical analysis strengthens our imaging studies and provides molecular-level evidence of the signaling events triggered by sEV interaction.

We added these results to Supplementary Fig. 16 and their explanation on pages 18-19.

Overall, this study constitutes a high-standard documentation of EV internalization characterization and imaging but falls short in providing novel mechanistic insights.

As previously mentioned, the finding of this study clearly demonstrated that: 1) distinct subtypes of sEV subtypes exhibit different membrane fluidity; 2) the sEV subtype containing only CD63, induces raft-like membrane domains in the recipient cell PM directly beneath the sEV, which may lead to caveolae partitioning; 3) all sEV subtypes promote clustering of integrin and talin beneath the sEVs; 4) integrin and talin clustering rapidly triggers intracellular signaling pathways (SFK, FAK, PLC γ 2, IP $_3$ receptor); 5) the endocytic pathways of EVs are subtype-specific, with internalization enhanced by calcineurin-dynamin activation driven by elevated intracellular Ca $^{2+}$ levels; and 6) paracrine EV binding to the recipient cell PM elicits a Ca $^{2+}$ response, whereas autocrine EV binding does not. We believe that all these novel findings offer significant insights.

Reviewer #5 (Remarks to the Author):

In this paper by Hirose et al, endocytosis and signaling of different classes of extracellular vesicles is explored. First the authors use an interesting fluorescence method to show that vesicles have heterogeneous amounts of different tetraspanin markers. Thus, they classify these vesicles into different sub-types according to their cargo. When these vesicles were added to cells, they co-localized with specific membrane marker such as caveolae, integrins, and talin, and were internalized in a dynamin-dependent manner. The binding of EVs from other types of cells induced a cytosolic calcium increase that lead to an activation of Src and PLC. This enzyme/ion cascade activated calcineurin and promoted EV uptake into recycling organelles. From these studies, the authors develop a global model of paracrine (cell-to-cell) EV adhesion and uptake. In general, I find the paper interesting. I have several specific comments that the authors might consider to improve the manuscript.

We appreciate the reviewer's thoughtful and constructive comments. In response, we performed additional experiments and revised the manuscript in accordance with all the reviewer's suggestions. We believe that the revisions have significantly strengthened the manuscript.

1. The concept of autocrine and paracrine in this manuscript is a bit complex and I feel maybe not appropriate. While the authors show that many of the effects are limited to those that occur between different cell types, I am not sure if it is appropriate to present this work as a difference between "autocrine" and "paracrine" types of interactions in an organism. In general, I would add a discussion to temper this type of language and its interpretations.

We appreciate your comment. We agree that we need to add an explanation (give a definition) about autocrine and paracrine signaling. We described the meaning of "paracrine sEV binding" and "autocrine binding" in the second paragraph on page 17.

2. In figure 1 it would be helpful to discuss or present how a random distribution of 4 proteins that were sorted into a single population of EVs would be distributed and how this distribution is different from the one the authors measure with their labeling methods assuming the number of copies the authors quantitate in their images applies. To what degree does the measured data diverge from stochasticity?

We appreciate your crucial comment. The distributions of CD63, CD9, and CD81 in sEV particles significantly deviated from the random variation predicted from a Poisson distribution. Each sEV contains 4.2-4.8 fluorescent marker molecules on average. If the marker molecules have a single distribution, it should follow a Poisson distribution, in which case 96.5-98.4% of the fluorescent marker molecules would be expected to be found within the same sEV. However, our observed distribution differs largely from this expectation, indicating a marked bias in the distribution pattern of marker molecules among sEVs.

We added these sentences in the first paragraph of page 6 of the revised version.

2. Please discuss how membrane packing defects and fluidity would affect the behavior of EVs in the discussion.

Thank you very much for this comment. We agree that this is an important discussion. We found that a GPI-anchored protein, which preferentially localizes into a raft-like domain (liquid-ordered [Lo] like domain), was enriched in the PM beneath CD63-containing sEVs that have lower membrane fluidity. Therefore, raft-like domains may be symmetrically formed on the sEV membrane and the cell PM just beneath the sEVs as shown in Fig. 6d of the revised version. This symmetrical structure seems plausible because previous studies showed that focal adhesions are highly ordered structures similar to rafts (Gaus et al., 2006) and that some integrin subunits such as $\alpha 6$ and $\beta 4$, undergo palmitoylation and partition into rafts (Gangoux-Palacios et al., 2003; Yang et al., 2004). One of the mechanisms of such a symmetrical structure may be associations of raftophilic molecules between the sEV membrane and the cell PM. For example, glycosphingolipids, typical raft markers, associate with each other between membranes (Day et al., PNAS, 2015). However, further studies are necessary to elucidate the exact mechanisms of such a symmetric structure.

We added this discussion in the second paragraph on page 23 to the second line on page 24.

3. Figure 3 has a very nice experimental method but EV endocytosis has already been shown to be influenced by dynamin. Please expand on how these data add to the existing literature.

Thank you for this comment. Previous studies have reported dynamin involvement in sEV uptake in certain cell types. In our experimental system, we considered it crucial to validate the effect of dynamin activity modulation on sEV uptake through imaging. This validation served as preliminary information for selecting membrane invaginations for high-resolution imaging shown in Fig. 4 and Fig. 5.

Therefore, to emphasize the validation of the effect of dynamin activity on sEV uptake in our experimental system, we added “of PC3 cell-derived sEVs by the recipient PZ-HPV-7 cell “ in the last sentence of the first paragraph on page 9.

4. Again, figure 4 is experimentally nice but how often were these behaviors observed? The authors should present some statistical analysis of the frequency or generality of these behaviors in the figure and text with errors.

Thank you for this thoughtful comment. We agree with the reviewer's comment and described the frequency of the colocalization events from the last line on page 12 to the first paragraph on page 13 of the revised version as follows.

The frequencies of the colocalization events (≥ 1 frame) between sEVs and phagocytic regions or caveolae were 2.12 ± 0.88 events/ $\mu\text{m}^2/\text{min}$ (mean \pm SE) and 0.15 ± 0.12 events/ $\mu\text{m}^2/\text{min}$, respectively. Moreover, the number densities of sEVs, which showed colocalization with phagocytic regions or caveolae, were 0.090 ± 0.022 particles/ $\mu\text{m}^2/\text{min}$ and 0.031 ± 0.022 particles/ $\mu\text{m}^2/\text{min}$, respectively. These results indicate that sEVs colocalize with phagocytic regions more frequently than with caveolae.

5. It would be very helpful to show larger images of the entire cell in super-resolution such that the images and overlap between the probes could be evaluated in Figure 5 by the reader. This is also true for Figure 6, where sEVs are co-localized with talin and integrins. While I appreciate the quantitative presentation, pairing with the actual whole-cell imaging data in the figure would be appropriate.

We appreciate the valuable comment. Colocalization events on the cell surface occur asynchronously, and events are transient and do not frequently occur. Even for statistically significant events such as colocalization of sEVs with LAMP2 (Fig. 5b), we observe fewer than three occurrences per $100 \mu\text{m}^2$, as shown in Fig. 4. Consequently, the sporadic and transient nature of these events makes it difficult to capture the colocalization in single, static images of the entire cell. It highlights the necessity of our statistical approach, which aggregates data from multiple cells and time points to provide a more accurate representation of event frequency and significance.

6. “The lipid composition of caveolae is similar to that of lipid rafts, and GPI-anchored proteins, a representative raft marker, are concentrated in caveolae upon cross-linking.” As a control I believe it is important to show that the GPI-marker is a good and specific marker for caveolae in this system. Please include this control.

Thank you for the reviewer’s suggestion, and we agree with the comment. We performed additional experiments to validate that crosslinked GPI proteins exhibit a high degree of colocalization with caveolae in PZ-HPV-7 cells. We obtained the result as shown in Supplementary Fig. 11. We added their explanation on pages 14-15.

7. The role of dynamin in caveolae is under dispute and complex (Parton et al. Nature Review MCB 2024). Please discuss and evaluate how this role for dynamin could impact the author’s data and conclusions.

We appreciate your valuable insight. Indeed, the role of dynamin in caveolae-mediated uptake remains a subject of debate. In our study, we discovered that sEV adhesion triggers signaling that regulates dynamin activity, inducing sEV uptake throughout the cell. However, we were unable to quantify the specific contribution of caveolae to this phenomenon. Dynamin may play a more significant role in regulating phagocytosis, where we observed more pronounced colocalization with sEVs. This observation suggests a potential differential involvement of dynamin in various uptake mechanisms. Our findings highlight the complex interplay between sEV adhesion, signaling cascades, and uptake mechanisms. While we've established a link between sEV-induced signaling and dynamin activation, the exact contributions of different endocytic pathways, including caveolae-mediated uptake and phagocytosis, require further investigation.

8. Caveolae are on-average around 90 nm in diameter. An sEV is around the same size. How do the authors propose that caveolae can endocytosis an object of the same size or larger? Please discuss.

Our dSTORM observations revealed that 91% of CD63-containing sEVs derived from PC-3 cells had a diameter of ≤ 90 nm (Fig. 2b). Given that caveolae have an average diameter of approximately 90 nm³⁴, sEVs with a diameter of ≥ 90 nm are unlikely to undergo caveolae-mediated endocytosis and may be preferentially internalized via phagocytosis.

We added this discussion in the second paragraph on page 13 of the revised version.

REVIEWER COMMENTS

Reviewer #1 (Remarks to the Author):

In this revised version, Dr. Suzuki and colleagues have addressed some of my concerns. However, critical issues remain unresolved:

First of all, we sincerely appreciate the reviewer's thoughtful and constructive comments. We are particularly grateful for the suggestion regarding the use of LAMP-2 as a phagocytosis marker. After carefully reconsidering the manuscript in light of the reviewer's remarks, we recognized that employing LAMP-2 as a phagocytic marker is unwarranted, as we have no direct evidence of "membrane zippering" around a particle. Instead, based on the findings presented below, we used LAMP-2 as a marker for "clathrin-independent endocytosis", a more comprehensive uptake mechanism that encompasses phagocytosis (with phagocytosis itself being one modality of clathrin-independent endocytosis).

LAMP-2 is well established as a clathrin-independent endocytosis marker, as indicated in previous reports (Lakshminarayan et al. *Nat. Cell Biol.*, 2014; Howes et al. *J. Cell Biol.*, 2010; Leone et al. *J. Immunol.*, 2017). Moreover, inhibition of dynamin activity elevates the expression level of LAMP-2 in the plasma membrane (Janvier et al., *Mol. Biol. Cell*, 2005). In addition, our immunostaining experiments revealed that nearly all sEVs were coated with galectin-3, and that the inhibition of galectin-3 with a membrane-impermeable inhibitor almost entirely eliminated galectin-3 from the EV surface (Fig. 4c in the revised manuscript). Previous proteomic analyses have shown that LAMP-2 on the cell surface interacts with galectin-3 (Fig. 3a and Supplementary Table 1 [number 48] in Lakshminarayan et al., *Nat. Cell Biol.*, 16, 592-603, 2015), and galectin-3 is internalized via clathrin-independent endocytosis. Furthermore, Yamaguchi et al. (*J. Biochem.*, 2024) demonstrated that LAMP-2C, but not LAMP-2A, or -2B, localizes at the cell surface and does not associate with any μ -chains of AP2 in clathrin-coated pits (we confirmed this by imaging as shown below). Taken together with our new data, these findings support the use of LAMP-2C as a clathrin-independent endocytosis marker.

We stated the rationale for employing LAMP-2C as a clathrin-independent endocytosis marker from the second line from the bottom of page 10 to the eighth line from the bottom of

page 11.

Intriguingly, after treatment of recipient cells with a membrane-impermeable galectin-3 inhibitor, the colocalization of individual sEV particles with clathrin-independent endocytic membrane regions visualized by LAMP-2C was markedly diminished. Furthermore, sEV internalization by recipient cells was substantially suppressed under these conditions, whereas dextran internalization was only moderately diminished by the inhibitor treatment likely due to the involvement of alternative internalization pathways for components other than EVs. Collectively, these observations explicitly demonstrate that galectin-3 on the sEV surface associates with LAMP-2C in the recipient cell PM, thereby facilitating clathrin-independent endocytosis. We have integrated these novel insights into Fig. 5b-c and Fig. 8a-c, and have described them in the middle of page 14 and in the third paragraph of page 22.

1. General Contribution to the Field of EVs. The authors claim that single-molecule detection is crucial to define the uptake of EVs and that previous studies were unable to address this at the single-object level. While I agree with the authors' statement, their work does not significantly advance the field.

Thank you for your thoughtful comment. As previously indicated, our single-particle and super-resolution microscopy analyses revealed that all sEV subtypes are predominantly internalized via the clathrin-independent endocytosis pathway, while sEV subtypes containing only CD63 are also internalized via caveolae. Without employing single-particle tracking methods, differentiating sEVs containing only CD63 from those containing all three tetraspanin marker proteins (CD63, CD81, and CD9) is not feasible.

Additionally, our new findings demonstrated that nearly all sEV particles derived from PC3 cells are coated with galectin-3, and the interaction between galectin-3 and LAMP-2C facilitates clathrin-independent endocytosis, as previously noted and discussed in our response to point 4. Furthermore, our experiments revealed that the inhibition of galectin-3 dramatically diminished the internalization of sEVs, indicating that galectin-3-LAMP-2C association is a crucial initial step for sEV internalization. These results represent a novel discovery that significantly advances our understanding of the field.

The reduction of sEV internalization after the knockout or knockdown of molecules within the uptake machinery in previous studies unequivocally shows the involvement of these molecules in EV internalization. While prior non-imaging studies have undoubtedly

significantly advanced the research field, we assert that the novel insights in the current study represent a meaningful advancement in the field, as no previous imaging studies have directly visualized the colocalization and internalization of individual EVs by specific uptake machinery.

o Dynamin Inhibition Experiment: The experiment is inconclusive, likely because dynamin inhibition is incomplete. Despite a statistical difference, significant uptake remains even with dynamin inhibition (Fig. 3H). The statistical difference suggests that a fraction of EVs is dynamin-dependent, but others are likely internalized via dynamin-independent processes. This experiment could be improved by using specific inhibitors such as Dynasore or Dyngo (Macia et al., Dev Cell 2006; McCluskey et al., Traffic 2013).

We greatly appreciate the reviewer's thoughtful comment. This is a crucial point, as our results indicate that the sEV adhesion-induced elevation of intracellular Ca^{2+} concentration in recipient cells enhances the activity of both calcineurin and dynamin.

We investigated the internalized content of sEVs and dextran after treatment with the specific dynamin inhibitor Dynasore, using confocal fluorescence microscopy. Our results demonstrated that Dynasore treatment nearly abolished the internalization of both sEVs and dextran, thereby indicating that virtually all sEVs are internalized via a dynamin-dependent mechanism. These novel results are presented in Fig. 8a-c and described from the third paragraph of page 22 to the first paragraph of page 23.

o Response to Point 7: The authors state, "We found that EVs containing only CD63 have more raft-like domains and were colocalized with caveolae more preferentially than sEVs containing CD81 and CD9." This is an overstatement since Fig. 5C clearly shows this is not true for sEV(4175-LuT)/recipient (WI-38). This suggests that additional molecules present in EVs might influence their trafficking.

We apologize for the inaccurate description. We have added "in PC3 cells" as follows.

The first paragraph on page 9;

Because membrane packing is closely correlated with membrane fluidity, the membrane fluidity of CD63-containing sEVs derived from PC-3 cells may be lower than that of CD9- and CD81-containing sEVs.

The second paragraph on page 26;

Our results indicate that a GPI-anchored protein, which preferentially localizes within a raft-like domain (liquid-ordered [Lo] like domain), was enriched beneath CD63-containing sEVs characterized by lower membrane fluidity in sEVs derived from PC-3 cells.

o Single EV Uptake: While I appreciate their effort to demonstrate caveolin or Lamp2 uptake of single EVs (a single event shown for each), this does not substantially alter the existing understanding in the field.

The internalization of sEVs via caveolae and clathrin-independent endocytosis is directly visualized for the first time in this study. Thus far, it has been known primarily that the knockout or knockdown of uptake-related proteins influences this process. Of course, there is no doubt that previous studies have significantly advanced the research field. However, uptake-related proteins, such as caveolin-1, are associated with numerous molecules and may affect uptake via multiple pathways, as described in the Discussion. We believe that directly visualizing the internalization events via the corresponding uptake machinery provides stronger evidence of the internalization pathway, and further advances the research field.

It has been previously established that the degree of colocalization between membrane molecules and the endocytic machinery strongly correlated with subsequent internalization (Cocucci et al., 2012; Ecker et al., 2021; Ma et al., 2022; Mund et al., 2023) as described in the second paragraph of page 13. Therefore, after we showed these results in Fig. 4g-j, we performed quantitative analyses of colocalization between sEVs and uptake machinery as shown in Fig. 5b and c.

However, we found several overstatements in the original manuscript as follows.

We stated “how sEVs are internalized by recipient cells has not been directly examined” in the Abstract on page 3. Since this may be an overstatement, we revised this part “the internalization of individual sEVs by recipient cells has not yet been directly observed”

Likewise, “However, pathways and factors that determine sEV uptake have not been directly

examined” in the introduction on page 4, was changed to “However, the pathways and factors controlling sEV uptake have not been directly observed.”

Similarly, “The mechanisms involved in the uptake of tumor-derived sEVs by recipient cells have not been directly examined” in the first sentence of Discussion section (the last line on page 23) was changed to “The mechanisms governing the uptake of tumor-derived sEVs by recipient cells have not been directly observed”

o Reliance on Statistical Inference: The work relies heavily on statistical inference from short co-localization events, which is an indirect approach.

Super-resolution images acquired by single-molecule localization microscopy must always undergo rigorous statistical analysis, and the methodology employed in this study is widely regarded as the “golden standard”, as reported by numerous studies (Stone et al. 2015 *Nat. Commun.*; Shelby et al. *Nat. Chem. Biol.* 2023; Pagoon et al. *Mol. Biol. Cell.* 2016, Nicovic et al. 2017 *Nat. Prot.*).

Specifically, we employed cross-correlation-based colocalization analysis, validated for live-cell single-molecule localization microscopy by Stone et al. and Pagoon et al. This sophisticated approach uniquely facilitates the analysis of diffusing molecules within dynamic live-cell environments, where traditional colocalization methods often fail due to molecular movement during image acquisition. The method provides robust, quantitative results by employing a model-independent approach that does not rely on assumptions regarding molecular behavior or distribution patterns. Furthermore, it consistently maintains reliable measurements even under fluctuating signal densities during imaging, a common challenge in live-cell microscopy. The statistical significance of detected interactions can be directly derived from the experimental data, providing a rigorous framework for evaluating molecular colocalizations. These combined features render it particularly well-suited for analyzing rapid molecular interactions in complex cellular environments.

To more clearly describe that the analysis we employed is appropriate, we cited references (41-44) in the fourth line from the bottom of page 13 and incorporated the sentence “The method yields robust, quantitative results by employing a model-independent approach that obviates the need for assumptions about molecular behavior or distribution patterns.” from the last sentence on page 49 to the second line on page 50, and in the middle of page 53.

Furthermore, it has been previously established by numerous papers that the degree of colocalization between membrane molecules and the endocytic machinery strongly correlated with subsequent internalization. Therefore, we have stated this and cited several representative references in the second paragraph on page 13 of the revised manuscript.

2. Calibration and Estimation of Tetraspanins in EVs. The authors have added single-molecule calibration, but serious concerns remain regarding the calibration and estimation of tetraspanins in EVs:

o Western Blot Calibration: The calibration for CD63 remains incomprehensible (Fig. 1C). The approach for CD9 and CD81 is also unclear. The straight forward method involves running EVs directly and blotting for tetraspanins; the ratio between the two bands could provide the substitution (first lane of the authors' blot). Currently, the authors compare signals from EVs with those from immunoprecipitated cell lysates, which is not relevant as the efficiency of protein sorting into EVs is unknown.

For CD63 quantification, we employed a direct approach by running EV samples and performing immunoblotting with CD63 antibodies, which exhibited sufficient binding affinity. To enhance data transparency, we have explicitly marked the regions used for quantification with dotted light blue and green boxes in Fig 1c, showing the bands corresponding to tagged and endogenous proteins within the same EV samples. We stated the following sentence in the caption of Fig. 1c. "The ratio of the band intensities of tagged marker molecules in the dotted light blue box to those of endogenous marker molecules in the green box was calculated."

While marking the bands with squares, we noticed that the western blot image of CD63+CD63-Halo7 in PC-3 cells presented in Fig. 1c of the previous version was incorrect. We have prepared two stable cell lines expressing CD63-Halo7 at high and low densities. The earlier version displayed the western blot image corresponding to the high-density CD63-Halo7-expressing cell line. However, the band intensity ratio of CD63-Halo7 to endogenous CD63 was approximately 0.5 in the low-density CD63-Halo7-expressing cell line, which was also used for all other experiments. Consequently, we have replaced the previous western blot image with that of the low-density expressing cells. We sincerely apologize for this inadvertent oversight.

The same direct approach was not applicable for CD9 and CD81 due to technical constraints. The combined effects of low antibody-binding affinity and low expression levels of tagged

tetraspanins impeded reliable detection of tagged proteins in the EV samples. Although we recognize that direct comparisons within EV samples, as the reviewer recommends, would be ideal, the signal intensity remained below our detection threshold. Therefore, we employed an alternative strategy utilizing higher-affinity tag antibodies to estimate the TSN bands (Supplementary Fig. 2). To achieve detectable signals, we employed two sample preparation methods: immunoprecipitation of GFP-tagged proteins and the use of cells overexpressing Halo-tagged proteins. These samples serve exclusively as intensity references for band detection, rather than as indicators of protein sorting efficiency into EVs. Specifically, we calculated the ratio of the band intensity in the purple square to that in the yellow square, which remains independent of protein sorting efficiency into EVs. For this analysis, we performed sequential immunoblotting, initially using anti-CD9 or anti-CD81 antibodies, followed by reprobing with anti-GFP or anti-HaloTag antibodies. Although we acknowledge the inherent limitations in comparing signals derived from distinct sample preparations, this method provided a practical estimation within our technical constraints. To clarify this approach further, we have expanded the explanation in the caption of Fig. S2.

o Improved Methodology: To assess substitution, the authors should immunoprecipitate EVs or load more EV material to detect the labeled band and compare it with the wild-type protein. Additionally, it is unclear why CD81 wild type can still be detectable when a anti halo antibody was used to immunoprecipitate CD81-Halo from the cell lysate.

As described in our previous response regarding technical limitations, despite considering the reviewer's suggestion, the low expression levels of tagged tetraspanins precluded the direct detection of tagged CD9 or CD81 with their respective antibodies, even after attempts at EV immunoprecipitation. Regarding the detection of wild-type CD81, to concentrate Halo-tagged proteins, we employed cells overexpressing these proteins rather than relying on immunoprecipitation. This experimental approach accounts for the presence of endogenous wild-type CD81 in the samples. To facilitate comprehension, an additional explanation of “Overexpression” has been provided above the bottom blot image in Fig. S2.

o Unclear Calibration: The calibration process remains unclear and should be revised for better accuracy and transparency.

We appreciate the reviewer's comments regarding the calibration process. To address this concern and enhance methodological transparency, we have now included the complete original fitting data, encompassing 1-10 tetraspanin molecules, in Supplementary Figure 1e, thereby enabling readers to thoroughly evaluate our calibration approach. The curve fitting was performed according to the previously reported method (Tsunoyama et al., *Nat. Chem. Biol.*, 2018), and is described in the second paragraph on page 6.

3. Quantification of CD63, CD9, and CD81 by Imaging The imaging results suggest that EVs contain variable fractions of the three tetraspanins, raising concerns about the conclusions in Fig. 5C. Specifically:

o It is unclear how the frequency of events in Fig. 1F does not sum up to 100% of events. In some cases the frequency is higher than 100% (e.g. compare CD9 with CD63 and CD81 mGFP).

We appreciate the reviewer's insightful comment regarding the frequency distributions in Fig. 1f. The values in Fig. 1f represent the percentage of tetraspanin-Halo7- in sEVs colocalized with tetraspanin-mGFP. The percentages for the combination of the same tetraspanin were normalized to 100%. For example, the percentage of CD9-Halo7 in sEVs colocalized with CD9-mGFP was normalized to 100%.

Subsequently, we assessed the percentage of CD9-Halo7 in sEVs colocalized with CD63-mGFP or CD81-mGFP, yielding estimates of $118 \pm 8.1\%$ and $124 \pm 8.3\%$, respectively. The percentages exceeding 100% likely result from the combined effects of strong colocalization between these molecules and higher expression levels of CD63-mGFP and CD81-mGFP than CD9-mGFP. Even if the actual percentage of CD9-Halo7 in sEVs colocalized with CD9-mGFP was less than 100%, it was still normalized to 100%.

As shown in Fig. 1b, we employed cell sorting to equalize the expression levels of tagged tetraspanins in donor cells. However, at the single sEV particle level, achieving completely uniform expression levels was not feasible, leading to minor variations in the amounts of tagged proteins.

Nevertheless, the observed distribution significantly deviates from the random variation predicted by a Poisson distribution. To provide a more comprehensive explanation for why the frequency exceeds 100%, we revised the caption in Fig. 1f accordingly.

“Calibration of percentages was performed by observing the colocalization of the same TSNs

stained with different dyes.” was changed to “The percentages of tetraspanin-Halo7 in sEVs that colocalized with the same tetraspanin-mGFP were normalized to 100%. Occasionally, the percentages exceed 100%, likely attributable to minor variations in tetraspanin-mGFP content in sEVs.”

o The data suggest that the three populations behave similarly, with the only consistent result being increased colocalization with Lamp2 compared to CAV1 or AP2.

Our results indicate that sEVs containing only CD63 exhibit smaller membrane defects and reduced membrane fluidity relative to other EV subtypes. Furthermore, sEVs containing CD63 are colocalized with GPI-anchored proteins and caveolae, whereas other sEV subtypes do not display such colocalization. Thus, the three EV populations do not behave similarly. Nevertheless, all three EV subtypes are internalized via a clathrin-independent pathway.

o The statistical significance of these events is limited due to the wide distribution of average pairwise distances. Transient low distances do not unambiguously indicate internalization.

We apologize for the insufficient explanation. The current statement does not adequately clarify the correlation between colocalization frequency and internalization. We did not explicitly state that it has been previously established by numerous papers that the degree of colocalization between membrane molecules and the endocytic machinery strongly correlated with subsequent internalization. This is a very important point. Therefore, we have stated this and cited several representative references in the second paragraph on page 13 of the revised manuscript.

4. Classification of LAMP2 as a Phagocytosis Marker The claim that LAMP2 is a marker of phagocytosis and that the observed process qualifies as phagocytosis is problematic:

o Phagocytosis Specificity: Phagocytosis is a highly specific process, primarily observed in antigen-presenting cells, involving membrane zippering around a particle. LAMP2, as a type I membrane protein containing a GYxxΦ motif (Yamaguchi et al., J Biochem 2024), is directed to endosomes or lysosomes via clathrin-coated vesicles. It is surprising that the authors do not observe colocalization of LAMP2 single molecules with clathrin-coated vesicles, which would align with its known trafficking pathway.

As previously noted, we agree with the reviewer that LAMP-2 should not be employed as a phagocytic marker. Instead, we used LAMP-2 as a marker for clathrin-independent endocytosis in this study. Specifically, we used LAMP-2C, which is nearly ubiquitously expressed (Bottillo et al., *Cardiovascular Pathology*, 25, 423-431, 2016). According to Yamaguchi et al. (*J. Biochem.*, 2024), as cited by the reviewer, LAMP-2C expressed at the cell PMs does not interact with any μ -subunits of adaptor protein (AP) complexes (μ 1, μ 2, μ 3A, and μ 4 of AP-1, AP-2, AP-3, and AP-4). As the reviewer recommended, we performed simultaneous observations of single molecules of LAMP-2C and super-resolution movies of clathrin-coated pits by monitoring AP2 α . We observed that LAMP-2C diffused within the cell PMs and was not concentrated in clathrin-coated pits. This result is consistent with Yamaguchi et al. and with Fig. 5c, which shows that sEVs are extensively colocalized with LAMP-2C domains, but not with AP2 α domains. We have incorporated these new results in Fig. 5b, c and described them in the first paragraph on page 14 of the revised version.

o Alternative Explanation: The observed process might involve clathrin-coated vesicle formation rather than phagocytosis. To rule out clathrin-mediated endocytosis, the authors should demonstrate transferrin colocalization with clathrin-coated vesicles as a positive control.

Thank you for this thoughtful comment. We agreed with the reviewer and performed the positive control experiment. The transferrin receptor was labeled with SaraFluor650T-transferrin, and we performed simultaneous observations of single molecules of SaraFluor650T-transferrin and PALM movie of clathrin-coated pits. We found that transferrin receptors frequently colocalized with clathrin-coated pits. We have incorporated this new result into Fig. 5b, c and described it from the last sentence on page 13 to the first paragraph on page 14 of the revised manuscript.

o Discrepancy with Cited Literature: The paper by Leone et al. 2017, cited by the authors, does not classify the endocytic process involving LAMP2 as phagocytosis or macropinocytosis but suggests it is clathrin-independent. The authors should address this discrepancy and avoid referring to the process as phagocytosis without sufficient evidence.

As noted previously, we used LAMP-2C as a marker for clathrin-independent endocytosis, the results by Leone et al., 2017 are consistent with our results.

5. Calcium Stimulation by EVs The Ca^{++} stimulation induced by EVs in the authors' system is intriguing but unclear. Specifically:

o It remains uncertain whether EV binding alone is sufficient to induce the calcium spike.

We observed Ca^{2+} mobilization after the treatment of recipient cells with sEVs isolated either by ultracentrifugation or via Tim-4 beads (Fig. 7c). As previously reported (Nakai et al., 2016), sEV purified using Tim-4 beads exhibit notably high purity. Additionally, exomeres did not induce Ca^{2+} mobilization. Furthermore, we found that integrin $\beta 1$ and talin-1 clusters formed beneath sEV particles, and that overexpressing the dominant negative mutant of talin-1 (TalinDN) inhibited Ca^{2+} mobilization. These results unequivocally demonstrate that EV binding alone elicits Ca^{2+} mobilization.

o This could be clarified by using inhibitors like Dynasore or Dyngo to block dynamin, which would help determine whether internalization is necessary for the calcium response.

We found that intracellular Ca^{2+} responses elevated dynamin activity during EV internalization, as shown in Fig. 8b-d. However, we did not show that EV internalization is required for the Ca^{2+} response. sEV binding induced a normal Ca^{2+} response even after cells were treated with a calcineurin inhibitor, FK506 (Fig. 8d), and the uptake of sEVs and dextran was largely reduced (Fig. 8b and 8c).

The authors demonstrated a continuous distribution of tetraspanin molecules in EVs, ranging from 1 to 50 (Fig1D). Although they employ single-molecule detection, their localization relies on the PALM/STORM approach, which requires multiple localizations. This raises the possibility that their observations are biased toward larger EVs containing more tetraspanins. Smaller EVs that truly undergo internalization through caveolin or Clathrin might remain untracked.

We did not perform PALM/dSTORM observations and, therefore, did not quantify localizations to estimate the number of tetraspanin molecules in individual EV particles. The results in Fig. 1d were obtained by single-molecule imaging of tetraspanin-Halo7 labeled with TMR, which does not exhibit blinking behavior. We measured the TMR fluorescence intensity in individual

EVs and plotted these values in Fig. 1d. The horizontal axis in Fig. 1d represents fluorescence intensity in arbitrary units and does not mean the number of molecules. The estimated average number of tetraspanin molecules per EV particle ranged from approximately 4.2 to 4.8. Fitting these values with a lognormal function (Tsunoyama et al., *Nat. Chem. Biol.*, 2018) revealed that the tetraspanin count per EV particle varied from 1 to 10, as shown in Supplementary Fig. 1d. Since fluorescence intensity (I) fluctuates between $I \pm \sqrt{I}$ according to photophysics principles, the intensity of tetraspanin-Halo7-TMR in EVs ranges from $I - \sqrt{I}$ to $10I + \sqrt{10I}$. Thus, a 50-fold variability in fluorescence intensity is fully justifiable. Since we observed SF650T-labeled EVs using TIRFM at the single molecule detection sensitivity, we were able to visualize the internalization of all EVs without bias (SF650T does not exhibit blinking, while SF650B does). Notably, we detected strong colocalization between clathrin-coated pits and SF650T-labeled transferrin receptor, which has a dye-to-protein ratio of 2.2, as shown in Fig. 5b, c. The number of dye molecules conjugated with tetraspanin-Halo7 per sEV ranges from 4.2 to 4.8, significantly higher than that of the transferrin receptor.

Overall, while the authors have made some improvements, the manuscript still requires substantial revisions to address the unresolved issues outlined above.

We believe that the revisions detailed above have comprehensively addressed all of the reviewer's concerns.

Reviewer #2 (Remarks to the Author):

The authors have added important clarity to the manuscript, by adding more data as well as more important information required to follow the experiments.

Reviewer #3 (Remarks to the Author):

The authors have addressed all the specific points which were raised (by me). Their responses show that they appreciate the importance of some points and have also done additional work. The most notable new work (shown in Suppl fig 18) clearly demonstrates the increases in PLC

signal and the subsequent Ca²⁺ signal in individual cells. There was a variable 'latency' as is often seen (eg uncaging IP₃ in individual cells can give a slightly later and variable Ca²⁺ signals.) It is a pity that the authors did not show (or comment on) the spatial data for this effect as it may provide evidence for spatially restricted signalling by individual vesicles. However these are minor points compared to the major successes in the paper. I feel that in its present form this paper makes an important contribution to the field of extracellular vesicle physiology and is of the high quality expected of papers published in Nat Commun.

Thank you very much for your positive evaluation and insightful comment. Indeed, spatially restricted signaling by individual vesicles presents a highly intriguing phenomenon. As mentioned in our previous response, Ca²⁺ release from the individual IP₃ receptor is a very rapid process, and we cannot observe changes in Ca²⁺ concentration with our current methodology. Nevertheless, by developing a high-speed observation technique we are eager to explore this aspect in future investigations. We gratefully appreciate your constructive feedback once again.

Reviewer #4 (Remarks to the Author):

My concerns have been addressed, thank you for the response in the rebuttal letter. Significant work has consolidated the internalization model, and super-resolution nanoscopy, the real strength of the study, is outstanding.

Reviewer #5 (Remarks to the Author):

The authors have addressed my specific concerns and questions from the first round of review with additional text and figures.